# Self-anticoagulant sponge for whole blood auto-transfusion and its mechanism of coagulation factor inactivation

Tao Xu [1], Haifeng Ji[1,2] ✉, Lin Xu[1], Shengjun Cheng[1], Xianda Liu[1], Yupei Li[3], Rui Zhong[4], Weifeng Zhao [1] ✉, Jayachandran N. Kizhakkedathu [2,5] & Changsheng Zhao[1,6]

Clinical use of intraoperative auto-transfusion requires the removal of platelets and plasma proteins due to pump-based suction and water-soluble anticoagulant administration, which causes dilutional coagulopathy. Herein, we develop a carboxylated and sulfonated heparin-mimetic polymer-modified sponge with spontaneous blood adsorption and instantaneous anticoagulation. We find that intrinsic coagulation factors, especially XI, are inactivated by adsorption to the sponge surface, while inactivation of thrombin in the sponge-treated plasma effectively inhibits the common coagulation pathway. We show whole blood auto-transfusion in trauma-induced hemorrhage, benefiting from the multiple inhibitory effects of the sponge on coagulation enzymes and calcium depletion. We demonstrate that the transfusion of collected blood favors faster recovery of hemostasis compared to traditional heparinized blood in a rabbit model. Our work not only develops a safe and convenient approach for whole blood auto-transfusion, but also provides the mechanism of action of self-anticoagulant heparin-mimetic polymer-modified surfaces.

Major surgeries in cardiology, obstetrics, orthopedics, and organ transplantation have historically posed a high risk of hemorrhage due to the aggressive nature of procedures and pre-existing coagulation disorders[1,2]. Intraoperative blood transfusion is often necessary to enhance oxygen-carrying capacity and maintain proper coagulation function[3]. However, the intermittent scarcity of allogeneic blood products, resulting from issues such as supply disruptions, stringent donor criteria, limited availability of rare blood types, and extensive laboratory testing, presents a significant threat to patient safety[4,5]. Recipients of allogeneic blood transfusions may experience complications including immunosuppression[6], allergic reactions[7], metabolic disturbances from citrate accumulation[8], and the potential transmission of viral and bacterial infections[9]. These complications highlight the urgent need for improved intraoperative auto-transfusion techniques that can minimize or eliminate the associated risks and adverse effects[10].

The core problems facing intraoperative auto-transfusion are challenges in blood collection and anticoagulation paradigm. Blood collection requires a dedicated suction device, which introduces micro-embolic bubbles and shear stress[11], leading to hemolysis and a decrease in the number of collected red blood cells (RBCs), and compromised blood quality[12]. Without anticoagulation, the

[1]College of Polymer Science and Engineering, State Key Laboratory of Polymer Materials Engineering, Sichuan University, Chengdu 610065, People's Republic of China. [2]Department of Pathology and Lab Medicine & Centre for Blood Research & Life Science Institute, University of British Columbia, 2350 Health Sciences Mall, Life Sciences Centre, Vancouver V6T 1Z3 BC, Canada. [3]Department of Nephrology, West China Hospital, Sichuan University, Chengdu 610041, China. [4]Institute of Blood Transfusion, Chinese Academy of Medical Sciences, Peking Union Medical College, Chengdu 610052, China. [5]School of Biomedical Engineering, University of British Columbia, 2350 Health Sciences Mall, Life Sciences Centre, Vancouver V6T 1Z3 BC, Canada. [6]School of Chemical Engineering, Sichuan University, Chengdu 610065, People's Republic of China. ✉e-mail: 903699293@qq.com; zhaoscukth@163.com

extracorporeal circuits are prone to occlusion by thrombi due to the rapid initiation and amplification of the hemostatic response during blood collection[13,14]. Citrate, the standard anticoagulant for blood collection[15], fails to inhibit the downstream thrombin generation via its calcium-chelating ability (Supplementary Fig. 1). Heparin inhibits thrombin and activated factor X (FXa) by binding to antithrombin III (ATIII)[16], and finds extensive use in preventing clot formation in extracorporeal circuits[17,18]. Nevertheless, the precise dose of heparin remains challenging to determine, primarily because of inter-patient variability and the fluctuating volume of blood collection, which is influenced by varying surgical conditions. Therefore, the removal of anticoagulant through centrifugation and washing of blood is necessary to prevent excessive systemic anticoagulation after auto-transfusion, although this process results in the discarding of platelets and plasma proteins[19]. Consequently, the transfusion of large volumes of collected RBCs develops dilutional coagulopathy, exacerbating clinical outcomes for patients undergoing major surgery[20].

The need to address anticoagulant-induced hemorrhagic risk in clinical settings has prompted research into biomaterial coatings capable of directly suppressing clot formation on extracorporeal devices. Examples include the development of heparin-immobilized surfaces to achieve localized anticoagulation[21]. However, the direct immobilization of heparin onto commercial hydrophobic devices is challenging without modifications, as heparin is hydrophilic. Electrostatic self-assembly methods for immobilizing heparin often result in leaching due to low stability[22], and the biological activity of immobilized heparin is generally lower than that of soluble heparin due to structural changes or constrained orientation on the surface[23]. Consequently, there is a demand for surfaces that can deactivate coagulation factors or provide persistent anticoagulation in collected blood[24]. Heparin mimetics, with an expanding scope ranging from structures closely resembling uronic backbone to other representative features that mimic the functional groups of heparin[25–28], have been extensively developed to address concerns associated with heparins. We recently reported a heparin-mimetic polymer (HMP)-modified microspheres with a tailored proportion of carboxyl and sulfonic groups, which inactivated intrinsic coagulation factors VIII, IX and XI (FVIII, FIX and FXI) and inhibited common coagulation pathway[29]. Based on our findings, we hypothesized that the multiple inhibitory effects of carboxylated and sulfonated HMP-modified materials on the coagulation cascade could enable their use in blood auto-transfusion without additional anticoagulants, and thereby achieve the unprecedented whole blood auto-transfusion in clinical trauma-induced hemorrhage.

However, the existing HMP-modified materials do not satisfy the need for simultaneous blood collection with instantaneous anticoagulation for blood auto-transfusion. We conceived that sponge with rapid capillarity-based spontaneous imbibition and retention of liquids may be a prime candidate for modification with HMP, providing large specific surface areas for blood contact, surface induced anticoagulation, and blood retention. However, the introduction of heparin mimetics onto the material surface adds complexity which necessitates deeper understanding the mechanism of self-anticoagulant function at the interface. Specifically, important questions remain unanswered: (1) which of the coagulation factors are affected by the HMP-modified coating on the sponge, (2) how the plasma protein layer builds on sponge during the blood collection and how HMP-modified surface mediates the anticoagulation function in the collected blood, and 3) whether such self-anticoagulation behavior of the material poses any potential side effects that may limit the design and future clinical use of HMP-modified sponges.

Here, we show the development of anticoagulant sponge device for the whole blood auto-transfusion, along with a comprehensive investigation of the anticoagulation mechanism of HMP-modified sponges to address these concerns. Our study not only offers a rapid and convenient method for whole blood auto-transfusion, but also contributes to the advancement of HMP-modified materials in practical clinical applications by shedding light on their inherent anticoagulant mechanisms.

## Results
### Fabrication and characterization of the anticoagulant sponges
The substrate selection criteria include biocompatibility, a large accessible blood-contacting interface, and spontaneous blood absorption and release. For proof of concept, melamine sponge (MS) with micrometer-size pores and porosity of ~93% was chosen. To functionalize the substrate, dopamine (DOPA) was acylated with vinyl groups to create $N$-(3, 4-dihydroxyphenethyl) acrylamide (DOPAm) (Fig. 1a, Supplementary Figs. 2 and 3). The DOPAm and DOPA were co-deposited on the MS to form an active vinyl-functionalized polydopamine foundation layer[30]. HMP was covalently decorated on the substrate through in-situ polymerization of monomers acrylic acid (AA) and 2-acrylamido-2-methyl-1-prroanesulfonic acid (AMPS), in a 3:2 (AA:AMPS) mass ratio, based on plasma recalcification time (PRT) in previous work[29]. Scanning electron microscopy (SEM) images revealed a rough surface morphology with micrometer-sized aggregated clusters after modification, attributable to the DOPAm/DOPA and HMP coatings (Fig. 1a)[31]. The presence of sulfur signals specific to HMP on the surface was confirmed through energy-dispersive X-ray spectroscopy (EDX) and X-ray photoelectron spectroscopy (XPS) surveys (Supplementary Figs. 4 and 5 and Supplementary Table 1). The introduction of DOPAm/DOPA and HMP resulted in a decrease in the nitrogen-to-carbon ratio on the surface, as evidenced by the low theoretical values (Supplementary Table 2). The quantities of deposited PDA and HMP on the surface, determined by the nitrogen-to-carbon ratio obtained from elemental analysis (EA; Supplementary Table 3), were 1.39% and 1.73%, relative to the weight of sponge, respectively. The confirmation of heparin-mimetic functional groups on the MS was achieved using Fourier-transform infrared spectroscopy (FTIR), which showed the characteristic peaks corresponding to sulfonic (1040 cm$^{-1}$) and deprotonated carboxyl (1626 cm$^{-1}$) groups (Supplementary Fig. 6).

The macroporous structure of the sponges was characterized by mercury intrusion porosimetry. Isostatic mercury intrusion, with increasing pressure ranging from 0.2 to 29.8 psia, showed a volume shrinkage of 59.1 cm$^3$ g$^{-1}$ for MS@D-HMP, confirming the highly porous and compressible structure (Supplementary Fig. 7). The porosity and mode pore size of MS@D-HMP were determined to be 67.5% and 134.1 μm, respectively, which are smaller than those of MS (93.0% and 162.7 μm; Fig. 1b and Supplementary Table 4). The rapid sorption process of the sponges was visually assessed by observing the permeation of water droplets. Water droplets (~20 μL) completely penetrated MS and MS@D-HMP within 15 and 24 ms, respectively (Fig. 1c and Supplementary Fig. 8), illustrating their highly absorbent nature. On the surface of MS@D, the water drop formed a dome shape (WCA = 103.2°) due to the hydrophobic property of vinyl-functionalized polydopamine coating. To quantitatively evaluate the rapid sorption process of the sponges, the liquid sorption coefficient $K_s$ was determined using a wicking method[32] (Supplementary Fig. 9). Deionized water and commercially available heparinized (2 IU/mL) goat whole blood were selected as model liquids, both of which adhered to the hydrophobic surface of MS@D without capillary imbibition. The $K_s$ value for deionized water into MS@D-HMP (1.506 kg m$^{-2}$ s$^{-1/2}$) was smaller than that of MS (2.064 kg m$^{-2}$ s$^{-1/2}$), which was attributed to the decreased porosity and pore size (Fig. 1d)[33]. Upon application of goat whole blood to MS@D-HMP, the $K_s$ value further decreased to 1.149 kg m$^{-2}$ s$^{-1/2}$ due to the lower surface tension and higher viscosity of blood[34]. Despite the kinetic reduction, the saturated liquid-sorption time of MS@D-HMP was less than 4 s without external energy input, ensuring a rapid sorption process that facilitates blood collection.

The surface charge of sponges was evaluated in phosphate-buffered saline (PBS; 10 mM, pH = 7.4), revealing an increase in negative surface

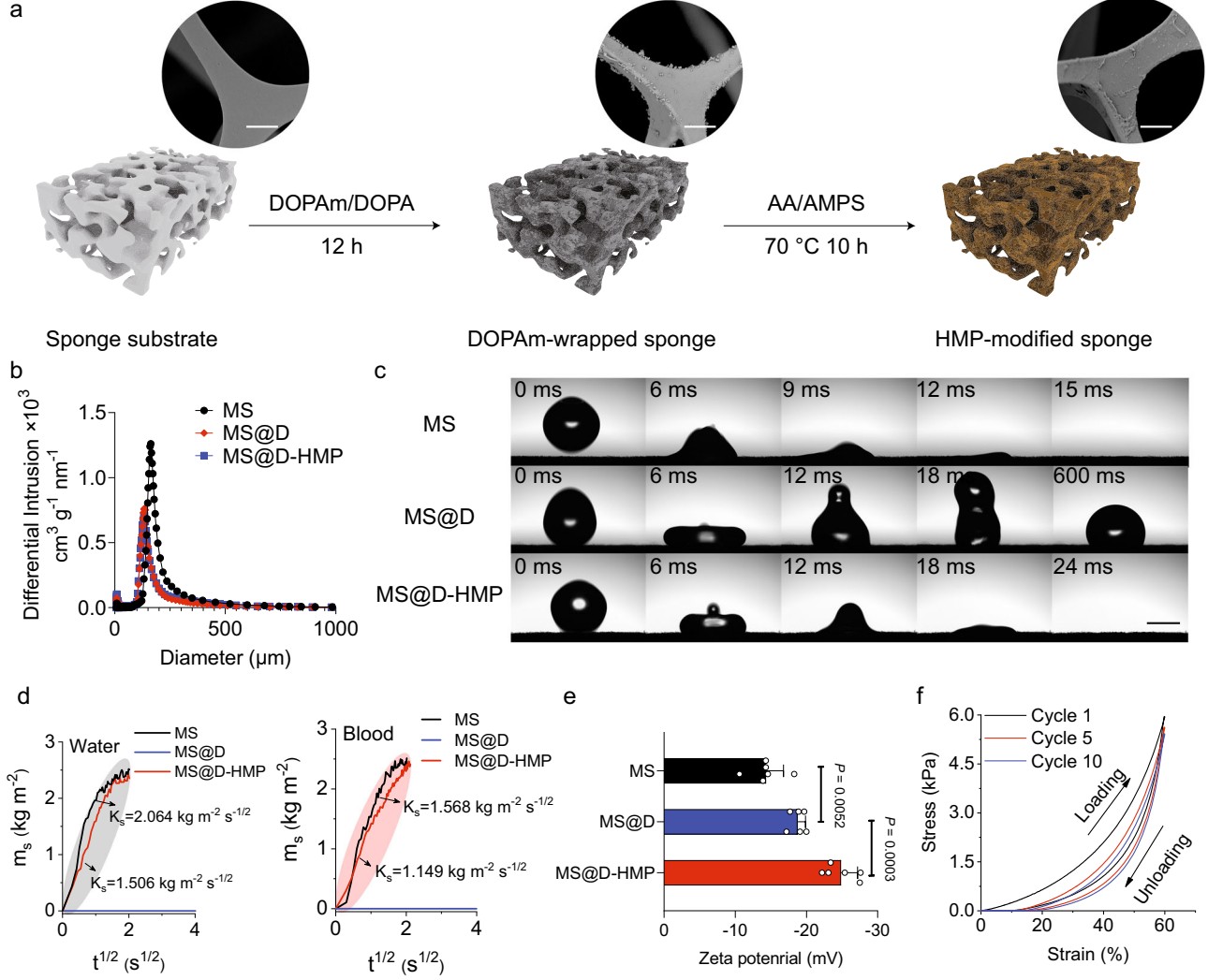

**Fig. 1 | Design and characterization of the sponges for instantaneous blood auto-transfusion. a** Schematic diagram of the preparation of the HMP-modified sponges. The inset SEM images show the rough surface morphology after modification. Scale bars, 10 μm. **b** Pore-size distributions of MS, MS@D and MS@D-HMP. The experiments were performed independently in duplicate, with similar results obtained. **c** Permeating behaviors of water (20 μL droplet) on the surface of MS, MS@D and MS@D-HMP. Scale bars, 2 mm. The experiments were performed independently in duplicate, with similar results obtained. **d** Quantitative evaluation of water (left) and blood (right) sorption into MS@D-HMP. The measured weight of water (left) and blood (right) absorbed into MS@D-HMP per unit contact area ($m_s$) versus the square root of sorption time ($t^{1/2}$). The experiments were performed independently in duplicate, with similar results obtained. **e** Zeta potentials of MS, MS@D and MS@D-HMP ($n = 6$ independent samples, mean ± SD. One-way ANOVA with Bonferroni post-hoc tests). **f** Compressive stress-strain curves of MS@D-HMP after loading-unloading cycles. The experiments were performed independently in duplicate, with similar results obtained. Source data are provided as a Source Data file.

charge following HMP modification (Fig. 1e). T Deprotonation of carboxyl and sulfonic groups in the HMP was necessary to prevent pH-induced denaturation of proteins on the polyanionic surface. The PBS did not experience any significant pH change upon exposure to MS@D-HMP, confirming successful deprotonation (Supplementary Fig. 10). Besides, MS@D-HMP withstood large-strain deformations with elastic recovery (Supplementary Fig. 11). Cyclic compression tests showed no significant structural fatigue (Fig. 1f), making it easier to extrude. Thermogravimetric analysis (TGA) demonstrated that MS@D-HMP withstood a temperature below 200 °C, making it suitable for steam sterilization (Supplementary Fig. 12).

### Instantaneous anticoagulation of whole blood in vitro exposed to the anticoagulant sponges

To illustrate the effect of MS@D-HMP on coagulation inhibition, systematic plasma- and whole blood-based coagulation assays were performed. We detected clotting times of platelet-poor plasma (PPP)

incubated with MS@D-HMP, including activated partial thromboplastin time (aPTT), prothrombin time (PT), and thrombin time (TT). MS@D-HMP significantly prolonged aPTT and TT in a concentration-dependent manner, and slightly prolonged PT (~3 s at 30 mg/300 μL PPP), suggesting that it interfered with the coagulation factors in the intrinsic and common pathway[35] (Fig. 2a). The prolongation for aPTT was observed within 30 s, and reached its maximal value within 10 min (Fig. 2b). However, both MS and MS@D showed weak anticoagulant effects even at high doses (>30 mg/300 μL PPP; Supplementary Fig. 13).

PRTs of MS@D-HMP-incubated PPP were monitored (Fig. 2c, d). The PRT for pristine plasma was 5 min, and a cross-linked fibrin clot formed in the glass vial. In contrast, the plasma incubated with MS@D-HMP (>30 mg per 300 μL plasma) maintained its fluid state for over 180 min. Notably, the serum calcium concentration significantly dropped after incubation due to chelation by the carboxyl groups of MS@D-HMP[36] (Supplementary Table 5). In a calcium ion

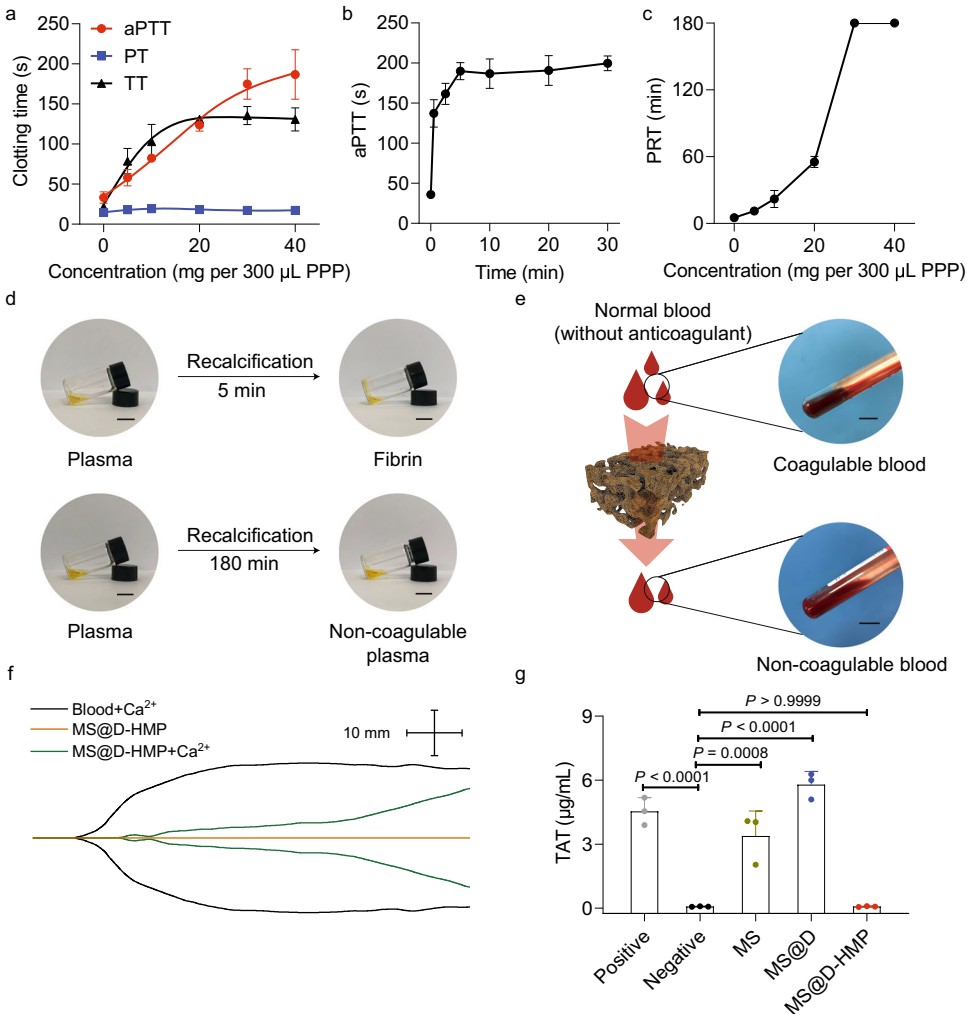

**Fig. 2 | Coagulation inhibition behaviors of the anticoagulant sponges in vitro.**
**a** Concentration-dependent prolongation of clotting times (aPTT, PT and TT) for the citrate-anticoagulated PPP after incubation with MS@D-HMP for 30 min ($n = 3$ biologically independent samples, mean ± SD). **b** aPTT for the citrate-anticoagulated PPP (100 μL) after incubation with MS@D-HMP (10 mg) for different time intervals ($n = 3$ biologically independent samples, mean ± SD).
**c** Concentration-dependent prolongation of PRTs for the citrate-anticoagulated PPP after incubation with MS@D-HMP ($n = 3$ biologically independent samples, mean ± SD). PPP (300 μL) was re-calcified with CaCl₂ solution (final Ca²⁺ concentration at 12.5 mM), and then incubated with MS@D-HMP. The treated PPP was taken out immediately (within 5 s), and monitored for clotting. **d** Photographs of

recalcification of fresh plasma (upper) and MS@D-HMP-treated plasma (lower) in the glass vial. Scale bars, 10 mm. **e** Schematic diagrams and photographs of the whole blood (without anticoagulant) after incubation with MS@D-HMP in vitro (created with BioRender.com). Scale bars, 10 mm. **f** TEG traces for pristine whole blood and blood after incubation with MS@D-HMP. **g** Generated concentrations of TAT in the citrate-anticoagulated whole blood after incubation with MS, MS@D, and MS@D-HMP for 30 min. Positive control (recalcified blood, final Ca²⁺ concentration at 10 mM) and negative control (citrate-anticoagulated blood) are also shown ($n = 3$ biologically independent samples, mean ± SD. One-way ANOVA with Bonferroni post-hoc tests). Source data are provided as a Source Data file.

solution (2 mM, physiological concentration), MS@D-HMP showed a rapid calcium ions adsorption kinetics, reaching equilibrium absorption capacity at 19.6 μg/mg (Supplementary Fig. 14a). The depletion efficiency of calcium ions remained high (85.1%) in the plasma environment, although a decreased equilibrium adsorption capacity (0.60 μg/mg) was observed due to the competitive effect of other cations and shorter incubation time (Supplementary Fig. 14b and Supplementary Table 6). To eliminate the influence of lost calcium ions on coagulability, MS@D-HMP was initially incubated with plasma for 30 min to ensure its interaction with coagulation factors. Subsequently, the plasma was separated from the sponges, and 10 mM calcium ions was added. The resulted PRTs were prolonged by approximately 7 min, reflecting the inactivation of coagulation factors by MS@D-HMP (Supplementary Fig. 15).

The in vitro anticoagulant properties of MS@D-HMP were investigated in human whole blood. Whole blood was collected without anticoagulant using sponge, and remained unclotted for over 3 h,

while the blood without sponge treatment formed a clot within 5 min (Fig. 2e). Thromboelastography (TEG) was employed to characterize the real-time viscoelastic behaviors of clotting blood, offering comprehensive insights into whole blood clot formation compared to plasma-based clotting times[37]. The TEG profile for the MS@D-HMP-incubated whole blood showed a straight line (Fig. 2f), confirming no clot formation. The initial clotting time ($R$) significantly increased from 10.8 min to over 80 min. When the incubated whole blood was recalcified to normal calcium ions level, the coagulation was initiated again, although it took a longer time (Supplementary Table 7). The prolonged $R$ value suggested the decreased activity of coagulation factors, whereas the prolonged $K$ value and the decreased α value suggested an impact on fibrinogen levels and platelets. Thrombin-antithrombin (TAT) complex concentrations were detected in the recalcified whole blood to explore whether coagulation system was initiated, as TAT is a surrogate marker for prothrombotic status. No significant difference was observed in TAT generation between MS@D-HMP-incubated

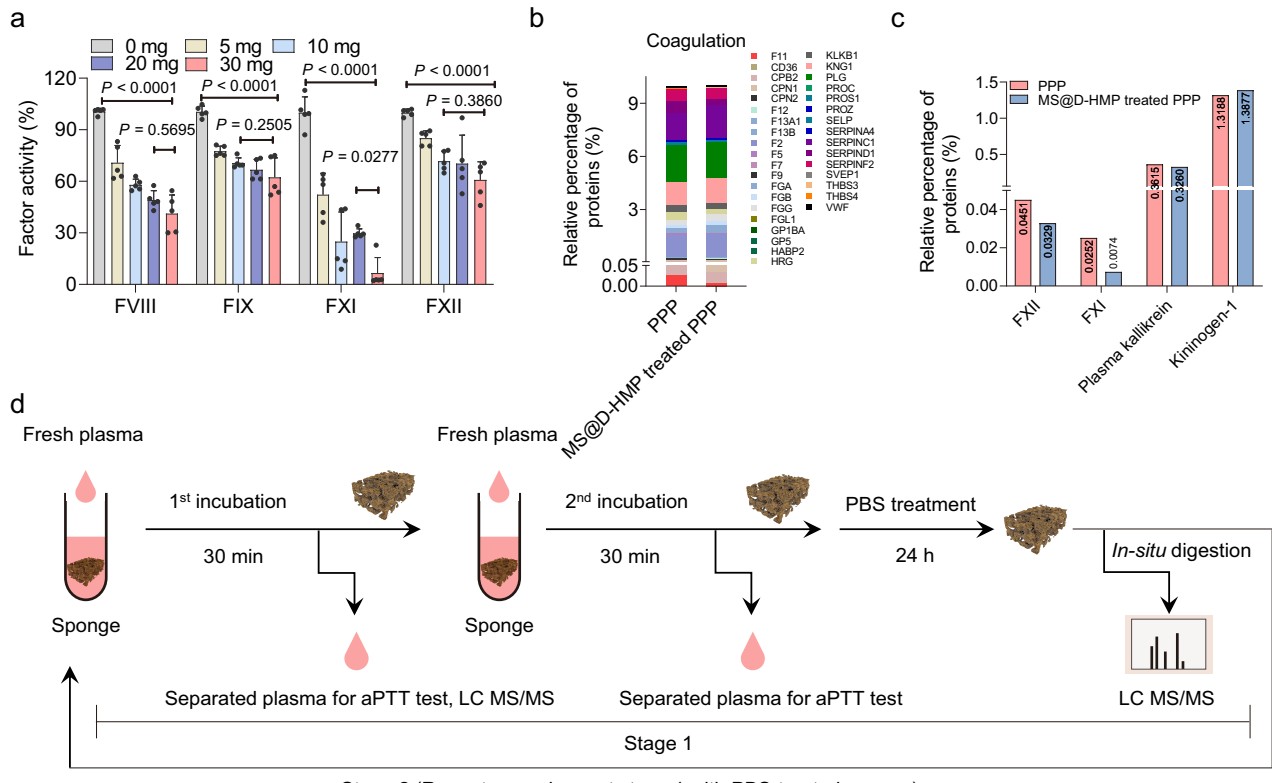

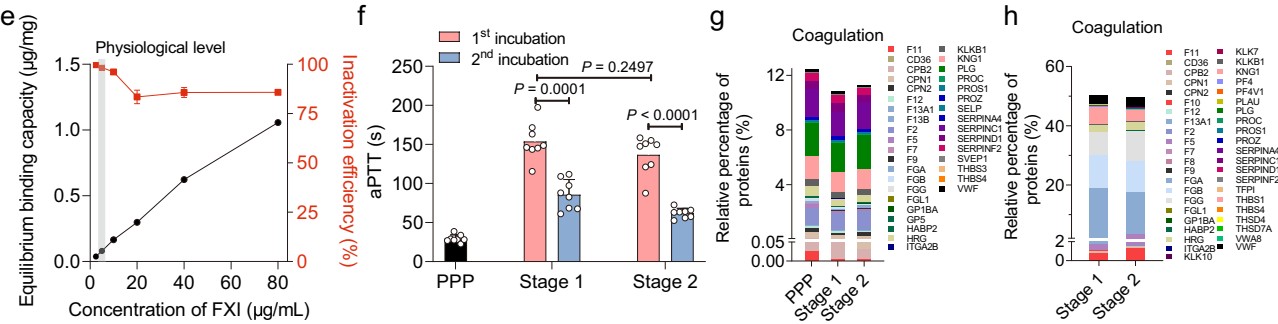

**Fig. 3 | Mechanisms of coagulation factor inactivation of the sponges.**
**a** Activities of FVIII, FIX, FXI and FXII for the normal citrate-anticoagulated PPP after incubation with MS@D-HMP at different concentrations for 30 min. The activities were determined based on aPTT by mixing incubated plasma with corresponding factor deficient plasma ($n = 5$ biologically independent samples, mean ± SD. One-way ANOVA with Bonferroni post-hoc tests). **b** Abundance of coagulation-related proteins in the normal plasma and MS@D-HMP-treated plasma. Citrate-anticoagulated PPP (300 μL) was incubated with MS@D-HMP (30 mg) for 30 min. Values are expressed in relative percentages based on total proteins. **c** Abundance of contact system-related proteins in normal PPP and MS@D-HMP-incubated plasma. Values are expressed in relative percentages based on total proteins. **d** Schematic of two-stage successive incubations combined with PBS treatment. In stage 1, tightly- and loosely-bound proteins were adsorbed on the sponge in 1st incubation with fresh plasma. Such sponge without washing was allowed for 2nd incubation with another fresh plasma. Then, the sponge undergoing stage 1 was

washed by PBS to remove the loosely-bound proteins, and used in stage 2 with similar procedures as those of stage 1 (created with BioRender.com). **e** Equilibrium binding capacity and the corresponding inactivation efficiency of FXI by MS@D-HMP. FXI-deficient plasma (100 μL) supplemented with different amounts of FXI proteins was incubated with MS@D-HMP (6 mg) for 30 min ($n = 3$ biologically independent samples, mean ± SD). **f** aPTT values for the PPP after successive incubations with MS@D-HMP in stages 1 and 2 ($n = 8$ biologically independent samples, mean ± SD. Two-way ANOVA with Bonferroni post-hoc tests).
**g** Abundance of coagulation-related proteins in fresh plasma, plasma after 1st incubation with MS@D-HMP in stages 1, and plasma after 1st incubation with MS@D-HMP in stages 2. Values are expressed in relative percentages based on total proteins. **h** Abundance of coagulation-related proteins in the tightly-bound protein layer on the surface of MS@D-HMP in stages 1 and 2. Values are expressed in relative percentages based on total proteins. Source data are provided as a Source Data file.

whole blood and the negative control (Fig. 2g). These results demonstrated the excellent in vitro anticoagulant properties of MS@D-HMP in both plasma and whole blood.

## FXI adsorption on the anticoagulant sponge blocks the intrinsic coagulation

The effect of MS@D-HMP on intrinsic coagulation factors (FVIII, FIX, FXI and FXII) was investigated. MS@D-HMP demonstrated a

remarkable inhibition of intrinsic factor activities, particularly for FXI. The residual activities increased in the following order: FXI (6.6%) <FVIII (39.8%) <FXII (60.8%) <FIX (65.0%), as the concentration of MS@D-HMP reached 30 mg/300 μL PPP (Fig. 3a). A slight decrease in activity was observed in plasma treated with MS and MS@D (Supplementary Fig. 16). Given the significant reduction in FXI activity, we further examined the inactivation patterns of FXI. Mass-spectrometry-based proteomics analysis provided information on the top 10 most-

abundant proteins in normal plasma and MS@D-HMP-treated plasma (Supplementary Data 1). The sum of the relative percentage of identified coagulation-related proteins in MS@D-HMP-treated plasma (10.04%) was not significantly different from that in normal plasma (10.07%; Fig. 3b). Serine protease inhibitors, such as ATIII, alpha-1-antitrypsin, and plasma C1 inhibitor, remained relatively unchanged (Supplementary Data 1). These inhibitors of FXIa may contribute to the decrease in FXI activity[38,39]. FXIa inhibited by these inhibitors could theoretically be digested to peptides and detected by the proteomic analysis, resulting in an inconspicuous change in FXI intensity. However, the relative percentage of FXI decreased significantly by 70.6% in MS@D-HMP-incubated plasma (Fig. 3c), indicating that the adsorption of FXI by the sponge played a decisive role in the reduced FXI activity. The relative abundances of proteins in the contact system and kallikrein–kinin system, such as FXII, plasma kallikrein and kininogen-1, also decreased in the MS@D-HMP-treated plasma. The assembly of these proteins on the surface might induce probable activation of the contact system with potential chance of clotting[40], which would be discussed in the later section.

## The loosely-bound protein layer suppresses the anticoagulant sponge-FXI interaction

Upon exposure to biological fluids, a phenomenon known as the "protein corona" occurs, whereby proteins are rapidly deposited on the surface of biomaterials due to nonspecific or specific adsorption[41]. This protein layer contains both tightly-bound proteins with strong binding affinity and long lifespan, and loosely-bound proteins, which are highly dynamic and rapidly exchanged. To investigate the relationship between the formation of the protein layer and the anticoagulant properties of MS@D-HMP, the effect of tightly- and loosely-bound proteins on the inactivation of intrinsic factors was examined using a two-stage successive incubation method combined with PBS washing (Fig. 3d). Prior to this, a strong sponge-FXI interaction was demonstrated through equilibrium FXI binding capacity and FXI inactivation efficiency in plasma with varying initial FXI concentrations (Fig. 3e). Notably, the FXI inactivation efficiency reached ~98% at a physiological FXI concentration (5 μg/mL), ensuring a high factor of safety. The equilibrium FXI binding capacity of the sponge reached 80.6 ng/mg at a physiological FXI concentration, and increased proportionally with the initial FXI concentration. Even at an initial FXI concentration of 80 μg/mL (16 times the physiological concentration), a high FXI inactivation efficiency (85.8%) was maintained, indicating a robust anticoagulant capability that could be applied to blood autotransfusion. Interestingly, in the two-stage successive incubations combined with PBS washing (Fig. 3d), the anticoagulant function declined only in the 2nd incubation. In the 1st incubation of stage 1, the fresh sponge significantly prolonged the aPTT (154.0 s), whereas the sponge after adsorbing plasma proteins exhibited weaker anticoagulant ability (aPTT 85.8 s) in the 2nd incubation of stage 1 (Fig. 3f). After the removal of loosely-bound proteins from the surface, the anticoagulant function of the sponge was restored in the 1st incubation of stage 2 (aPTT 136.8 s). This suggests that the loosely-bound protein layer developed on the surface might act as a barrier to the sponge-FXI interaction.

To illustrate the anticoagulant behavior of sponge without the loosely-bound protein layer, two aliquots of plasma with similar aPTT values (154.0 s versus 136.8 s) after the 1st incubation of stages 1 and 2 were collected for proteomics analysis. Compared to fresh plasma, a similar decrease in the relative percentages of FXI (88.5% versus 86.5%) was observed in these two aliquots of plasma (Fig. 3g). This suggests that the sponge exhibited a similar robust anticoagulant function (sponge-FXI interaction) when the loosely-bound protein layer was removed from the surface. Snapshots of tightly-bound proteins on the sponge were obtained after PBS washing, and SDS-PAGE analysis showed similar bands for both stages (Supplementary Fig. 17).

Proteomics analysis provided information about common proteins and their fold changes in relative abundance of different proteins (Supplementary Fig. 18a, b). These identified proteins were classified based on their biological functions[42] (Supplementary Fig. 18c–g and Supplementary Data 2). Compared with stage 1, relative percentage of tightly-bound FXI increased by 55.7% in stage 2 (protein intensity increased by 110.3%; Fig. 3h). Taken together, FXI can be effectively adsorbed into the tightly-bound protein layer, which might be suppressed after the building of the loosely-bound protein layer.

## Anticoagulant sponges activated FXI via contact system, but the contribution of activated FXI is minor

Considering the varied nature and abundance of adsorbed proteins, often with conformational alterations, it is unsurprising that the protein corona may trigger activation of blood-based proteolytic cascades[43]. Negatively charged surfaces can be readily recognized by FXII in plasma, leading to direct activation of contact system and subsequent activation of FXI[44]. We hypothesized that FXI could be activated through the contact system, and subsequently inactivated due to the presence of serine protease inhibitors. It is important to consider the various conformations of FXI, including the activated or inactivated states. The probable states of FXI in the MS@D-HMP-treated plasma can be summarized as follows (Fig. 4a): (i) zymogen (FXIn); 6.6% of FXI remained in the zymogen state. (ii) activated form (FXIa); FXI could be activated through the contact system or be directly activated upon adsorption to the surface, both of which exhibit procoagulant activity. (iii) inactivated form (FXIf); FXI could undergo sponge-induced allosteric alteration, and lose its procoagulant activity.

To evaluate the procoagulant activity of the sponge, the generation of FIXa in the MS@D-HMP-incubated plasma was measured using a fluorometric-based FIXa activity assay. FXIa activates FIX to FIXa in a calcium-dependent manner. The appropriate dose of calcium supplementation required to restore normal calcium levels in the MS@D-HMP-incubated plasma was determined based on the above-mentioned results in Supplementary Table 6. The possibility of direct activation of FIX by MS@D-HMP to generate active FIXa was also considered, and the detailed experimental setup is presented in Supplementary Table 8 and Supplementary Fig. 19. PPP+Citrate showed no obvious change in the generated FIXa compared to pristine PPP, confirming that the background-induced activation of FXI during incubation was negligible (Fig. 4b). MS@D-HMP+Citrate showed no substantial difference compared to PPP+Citrate, suggesting that the MS@D-HMP did not directly activate FIX. However, MS@D-HMP+$Ca^{2+}$ induced a significant increase in the generated FIXa compared to MS@D-HMP+Citrate, confirming the presence of FXIa in the MS@D-HMP-incubated plasma. As the calcium ions were synergistically chelated by the MS@D-HMP, the generated FXIa was unstable and might be rapidly inhibited by serine protease inhibitors (ATIII, alpha-1-antitrypsin, and plasma C1 inhibitor, etc.) in the plasma[38,39].

We then investigated the contact activation induced by MS@D-HMP and its contribution to the inactivated FXI. Activation of contact system initiates the kallikrein/kinin system, as FXIIa cleaves pre-kallikrein to generate plasma kallikrein, which in turn proteolyzes high molecular weight kininogen to generate bradykinin[45]. A competition ELISA assay revealed a significant rise in the generated bradykinin in MS@D-HMP-incubated plasma compared with pristine plasma, indicating the formation of FXIIa (Fig. 4c). Activation of FXII by MS@D-HMP was confirmed via chromogenic assays, showing significant proteolytic activity to chromogenic FXIIa substrate (S-2302) both in the sole FXII solution and normal plasma treated by MS@D-HMP (Fig. 4d, Supplementary Fig. 20). FXII-deficient plasma treated by MS@D-HMP showed no significant proteolytic activity (Fig. 4e), suggesting that contact system is activated in the plasma

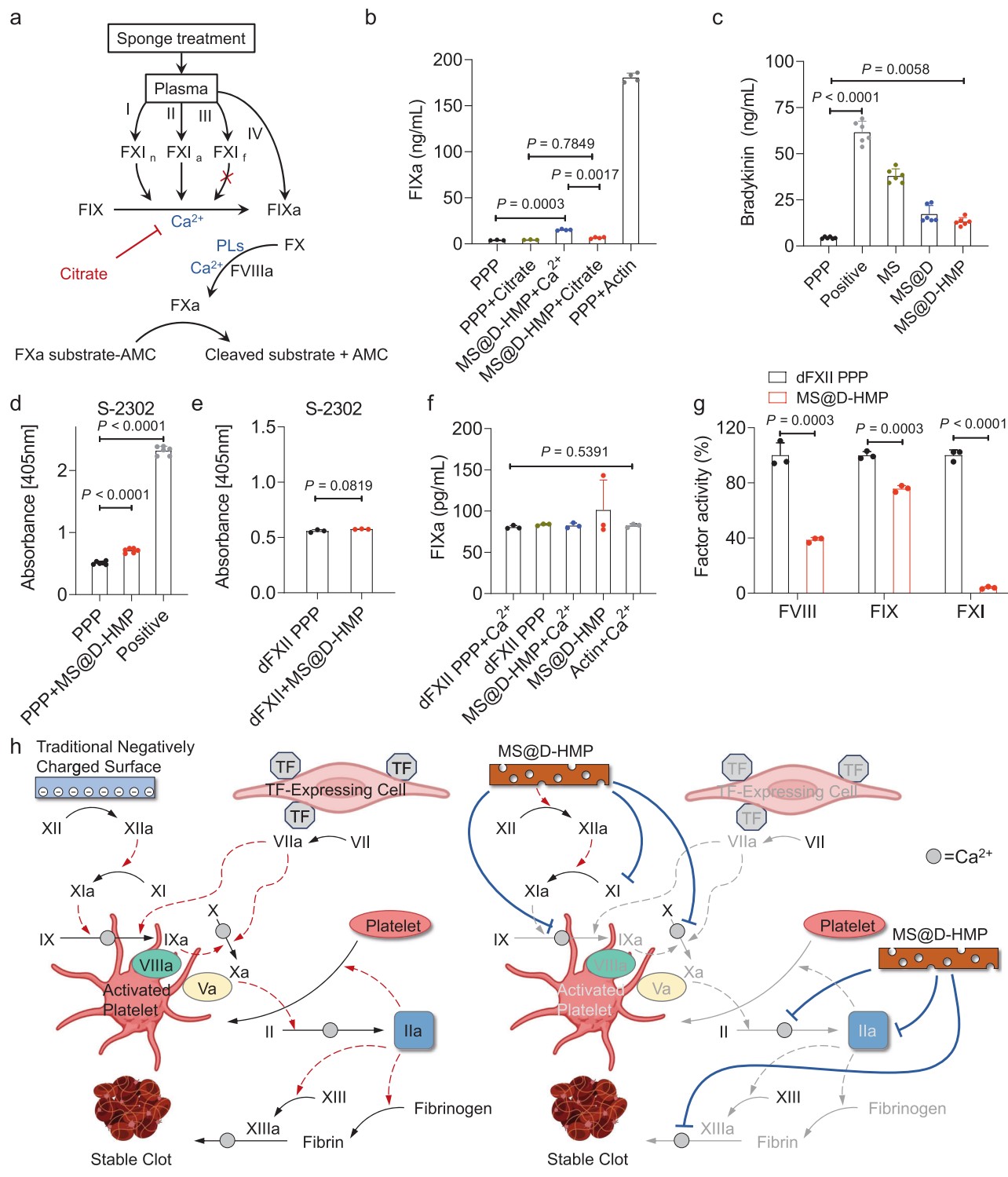

after contacting with MS@D-HMP in a FXII-dependent manner. Next, the generation of FIXa was investigated in the FXII-deficient plasma (dFXII PPP). All groups showed no apparent change compared to dFXII PPP+Ca$^{2+}$ group, indicating that no FXIa was generated in the absence of FXII (Fig. 4f). These results confirmed the subsequent generation of FXIIa, FXIa and FIXa through contact activation in the MS@D-HMP-incubated plasma when sufficient calcium was present. However, the activities of FXI, FIX and FVIII in the MS@D-HMP-incubated dFXII PPP were significantly inhibited to a similar extent as those in normal plasma, suggesting that contact activation only minimally contributes to the inactivation of FXI in the MS@D-HMP-incubated plasma (Fig. 4g).

## Anticoagulant sponges inhibited the common pathway

The prolongation of TT value may have two possible causes: either a deficiency of fibrinogen in the plasma, or the presence of certain inhibitors that impede the action of adscititious thrombin, such as heparin/ATIII complex[46]. The reduction in fibrinogen levels was observed in the MS@D-HMP-incubated plasma (Supplementary Fig. 21a, b). Despite the decrease in fibrinogen, its conversion into fibrin did not occur (as discussed in Fig. 5h). Furthermore, upon the addition of purified fibrinogen to restore its levels to the normal range, no reduction in the prolongation of TT was observed (Supplementary Fig. 21c). Hence, the loss of fibrinogen did not contribute to the prolongation of TT.

**Fig. 4 | Evaluation of inhibited coagulation pathways in the sponges-treated plasma. a** Schematic of exploring the procoagulant activity of FXIa in the MS@D-HMP-incubated plasma via fluorometric-based FIXa activity assay. **b** Quantitative monitoring of FIXa in the normal hirudin-anticoagulated plasma after incubation with MS@D-HMP for 10 min. Actin (aPTT reagent) was used as positive control and citrate was used to block the activation of FIX by FXIa ($n = 4$ biologically independent samples, mean ± SD. One-way ANOVA with Bonferroni post-hoc tests). **c** Generated concentrations of bradykinin in the citrate-anticoagulated plasma after incubation with MS@D-HMP for 20 min. Plasma without treatment and plasma treated with glass power (1 mg/100 μL plasma) for 20 min were used as negative and positive controls, respectively ($n = 6$ biologically independent samples, mean ± SD. One-way ANOVA with Bonferroni post-hoc tests). Evaluation of FXIIa activity in normal plasma (**d**) and FXII-deficient plasma (**e**) after incubation with MS@D-HMP for 15 min using hydrolysis of chromogenic substrate S-2302 at an absorbance λ = 405 nm. Positive control (actin, aPTT reagent) was shown in (**d**) ($n = 6$ biologically independent samples for (**d**), mean ± SD. One-way ANOVA with Bonferroni post-hoc tests for (**d**). $n = 3$ biologically independent samples for (**e**), mean ± SD. Unpaired, two-tailed student's $t$-test for (**e**)). **f** Quantitative monitoring of FIXa in the FXII-deficient plasma after incubation with MS@D-HMP for 10 min ($n = 3$ biologically independent samples, mean ± SD. One-way ANOVA with Bonferroni post-hoc tests). **g** Activities of FVIII, FIX and FXI for the FXII-deficient plasma after incubation with MS@D-HMP for 30 min ($n = 3$ biologically independent samples, mean ± SD. Unpaired, two-tailed student's $t$-test). **h** Schematic of initiated and inhibited coagulation cascade by traditional negatively charged surface and MS@D-HMP, respectively (created with BioRender.com). Source data are provided as a Source Data file.

Before conducting the TT tests, MS@D-HMP had been removed from the incubated PPP. The leaching of anticoagulant coating and its effect on clotting times were negligible (Supplementary Fig. 22). This meant that the adscititious thrombin in TT tests did not come into direct contact with MS@D-HMP. We hypothesized that MS@D-HMP might trigger the generation of unknown thrombin inhibitor in plasma. The effect of two traditional thrombin inhibitors, such as ATIII and heparin cofactor II (HCII), on sponge-mediated anticoagulation was investigated using ATIII-deficient plasma and purified HCII protein. In the case of ATIII-deficient plasma, MS@D-HMP caused a significant prolongation of aPTT and TT, along with a notable inhibition of FXI activity (Supplementary Fig. 23a–c). However, heparin lost its anticoagulant ability in ATIII-deficient plasma. Regarding HCII-mediated thrombin-initiated fibrin formation, MS@D-HMP-treated HCII did not significantly inhibit thrombin activity, whereas the co-presence of heparin and HCII led to significant inhibition of thrombin (Supplementary Fig. 23d, e). These findings suggest that the inhibition of thrombin by MS@D-HMP may not depend on ATIII binding or HCII similar to heparin.

Additional assays were performed to correct clotting times by mixing the sample plasma with fresh normal plasma at various concentrations. Heparin-anticoagulant plasma delayed clotting times when the mixing concentration exceeded 0.02 mL/mL normal PPP, posing a significant bleeding risk. For the MS@D-HMP-incubated plasma, TT values could not be corrected when the mixing concentration reached 0.1 mL/mL normal PPP due to the presence of unknown thrombin inhibitor (protease inhibitors, steric inhibitor, etc) activated by the MS@D-HMP (Supplementary Fig. 24a). However, PT and aPTT values were effectively corrected with the MS@D-HMP-treated plasma until the mixing concentration reached 1 mL/mL normal PPP, demonstrating the rapid restoration of hemostatic function (Supplementary Fig. 24b,c).

Overall, MS@D-HMP exhibits multiple inhibitory effects on the coagulation cascade (Fig. 4h). Intrinsic coagulation factors (particularly FXI) are adsorbed to the sponge surface, thereby blocking the activation of intrinsic pathway arising from blood-surface interaction (such as biomaterials, activated platelet). Although a portion of FXI is activated via the contact system, the activation of the downstream cascade is impeded due to the sponge's calcium-chelating ability. The depletion of calcium ions from blood also hampers other calcium-dependent steps in both the intrinsic and extrinsic coagulation pathways. Additionally, an unknown inhibitor activated by MS@D-HMP inhibited thrombin, thereby obstructing trauma-initiated thrombin generation and amplification in the collected blood.

## Biocompatibility evaluation of the anticoagulant sponges in vitro

The influence of sponges on various blood components was systematically investigated to fulfill their clinical utility. Complete blood count in vitro showed that the residual numbers of blood cells after incubation with MS@D-HMP were 94.3%, 58.9%, and 49.6% for RBCs, white blood cells (WBCs), and platelets, respectively (Fig. 5a and

Supplementary Table 9). These reductions were primarily attributed to the physical interception of porous structures, and a significant portion of the missing blood cells could be recovered through the washing procedure (Supplementary Fig. 25). The distribution of WBCs, RBCs and platelets showed negligible changes (Fig. 5b and Supplementary Fig. 26a). To assess the stress-induced rupture of RBCs when blood was extruded from the sponges, hemolysis experiments were conducted. The hemolysis ratio after treatment with MS@D-HMP fell within the acceptable range (0.92%; Fig. 5c), and the supernatant, upon centrifugation, appeared colorless, suggesting no significant RBC rupture or hemoglobin release (Supplementary Fig. 26b). Optical microscopic images of the collected RBCs also displayed typical biconcave morphologies.

We next investigated the impact of sponges on platelet adhesion and activation. SEM images of sponges after incubation with platelet-rich plasma (PRP) revealed minimal platelets on the sponge surface (Supplementary Fig. 27). Platelet adhesion levels were quantified by lactate dehydrogenase (LDH) assay. MS@D adhered significantly more platelets than MS, attributed to the residual chemical reactivity of decorated polydopamine coating[47] (Fig. 5d). The platelet adhesion was greatly reduced on MS@D-HMP, likely due to the increased hydrophilicity and abundance of electronegative groups[48]. To investigate the effect of sponges on platelet activation, platelet factor 4 (PF4) concentration was quantified, as PF4 is secreted from activated platelet[49]. Compared to pristine blood, the generated PF4 in the MS@D-HMP-incubated blood significantly decreased due to the adsorption of PF4 with a strong cationic charge to the negatively charged surface[50] (Fig. 5e). Platelet activation was also quantified using flow cytometry, based on the expression of platelet activation marker CD62p. Compared with negative control, there was no significant difference in the expression of CD62p in the PRP treated with MS@D-HMP (Fig. 5f, Supplementary Fig. 28a), confirming that MS@D-HMP did not induce significant platelet activation. The monocyte activation was also evaluated based on the expression of monocyte activation marker CD11b. Compared with negative control, there was no significant difference in the expression of CD11b in the MS@D-HMP-incubated monocytes (Fig. 5g, Supplementary Fig. 28b), confirming no significant monocyte activation. Given the activation of contact system, the possible generation of micro-thrombi and subsequent fibrinolysis in the plasma after incubation with MS@D-HMP was investigated based on the D-dimer level. Compared with normal plasma, no significant increase was observed in the D-dimer level in the re-calcified plasma treated with MS@D-HMP after 6 h incubation, demonstrating the absence of micro-thrombi formation and no degradation of fibrin (Fig. 5h).

The effect of sponges on complement activation was also investigated, as sponge activated contact system, and surface deposition of ficolin-2 and immunoglobulin was found in the proteomic analysis. Complement activation leads to the generation of complement anaphylatoxins C3a and C5a[51]. For MS@D-HMP, the generated C3a and C5a showed no significant changes compared to pristine blood, confirming no activation of complement system in the collected blood (Fig. 5i, j).

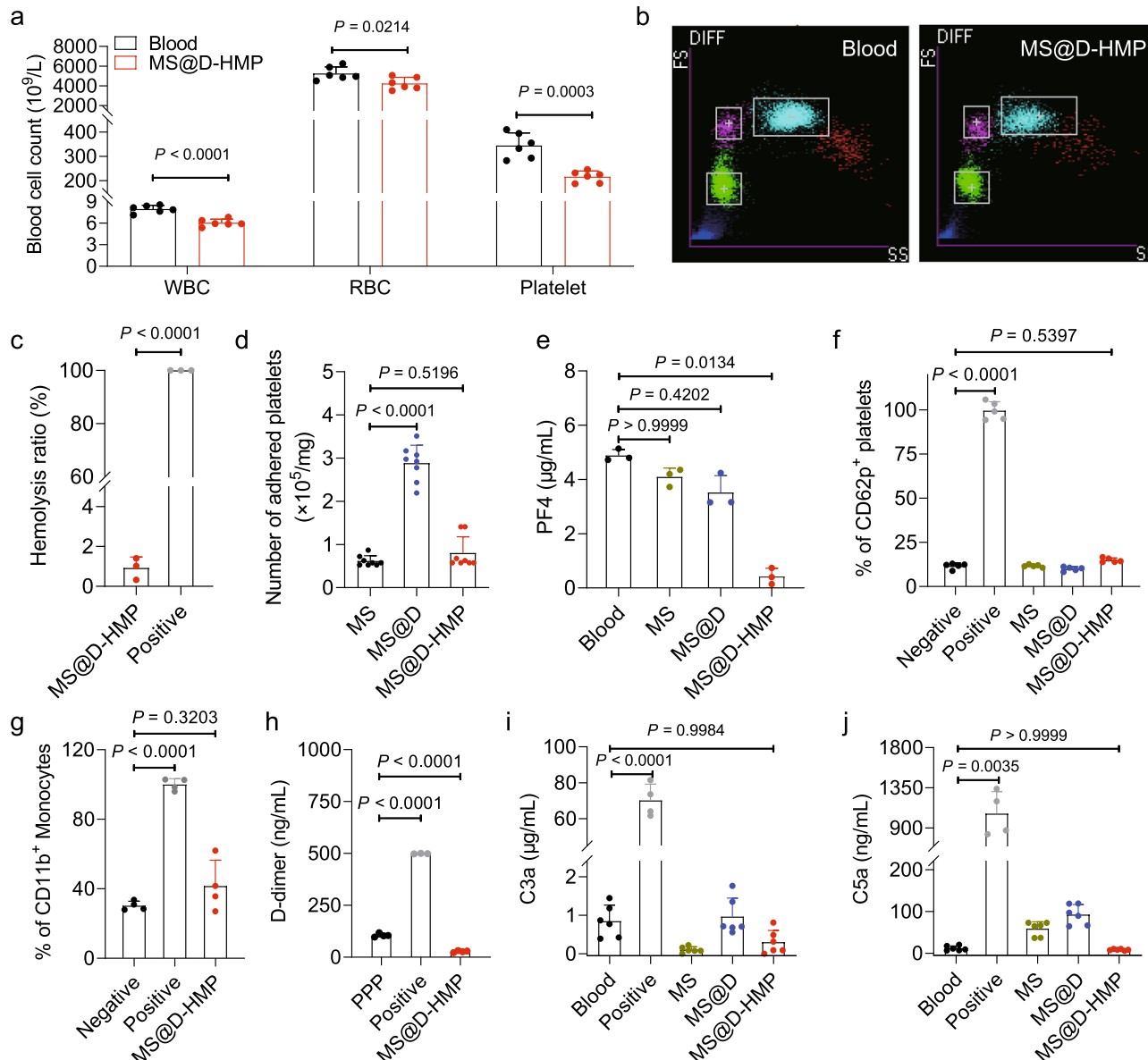

**Fig. 5 | Biocompatibility evaluation of the anticoagulant sponges in vitro.**
**a** Blood cell count for pristine blood and blood after incubation with MS@D-HMP ($n = 6$ biologically independent samples, mean ± SD. Unpaired, two-tailed student's *t*-test). **b** Differential white blood cell count (DIFF) scatter (forward (FS), side (SS)) charts of pristine blood and blood after incubation with MS@D-HMP. Monocyte (pink), lymphocyte (green), neutrophil (cyan), eosinophil (red). **c** Hemolysis ratio of the blood after incubation with MS@D-HMP ($n = 3$ biologically independent samples, mean ± SD. Unpaired, two-tailed student's *t*-test). **d** Quantitative evaluation of adhered platelets on the sponges by LDH assay ($n = 8$ biologically independent samples, mean ± SD. One-way ANOVA with Bonferroni post-hoc tests). **e** Generation of PF4 in the blood after incubation with sponges ($n = 3$ biologically independent samples, mean ± SD. Kruskal–Wallis test with Dunn's multiple comparisons). **f** Platelet activation in PRP based on the expression of platelet activation marker CD62p. Thrombin receptor activating peptide (TRAP, 0.1 mM) and control PRP were used as positive and negative control, respectively ($n = 5$ biologically

independent samples, mean ± SD. One-way ANOVA with Bonferroni post-hoc tests). **g** Monocyte activation in whole blood based on the expression of monocyte activation marker CD11b. Lipopolysaccharide (10 ng/mL) and pristine blood were used as positive and negative control, respectively ($n = 4$ biologically independent samples, mean ± SD. One-way ANOVA with Bonferroni post-hoc tests).
**h** Concentration of D-dimer in the citrate-anticoagulated plasma after incubation with sponge ($n = 4$ biologically independent samples for PPP and MS@D-HMP, $n = 3$ biologically independent samples for Positive, mean ± SD. One-way ANOVA with Bonferroni post-hoc tests). Generation of C3a (**i**) and C5a (**j**) in the blood after incubation with sponges. Cobra-venom factor (CVF, 25 μg/mL) and PBS were used as positive and negative controls, respectively ($n = 6$ biologically independent samples for Blood, MS, MS@D and MS@D-HMP, $n = 4$ biologically independent samples for Positive, mean ± SD. One-way ANOVA with Bonferroni post-hoc tests for (**i**), Kruskal–Wallis test with Dunn's multiple comparisons for (**j**)). Source data are provided as a Source Data file.

Even if contact activation partially triggered complement activation, the resulting products may not cause evident side effects due to adsorption of sponge. Additionally, calcium or magnesium ions are essential for activation of the classical, lectin, or alternative pathways[51]. The crosstalk between contact activation and complement activation may be inhibited by chelating calcium and magnesium ions. Furthermore, the products of contact activation, such as FXIIa and FXIa, may

also be inhibited by serine protease inhibitors, ensuring the safety of the recovered blood after treatment.

## Collection of rabbit whole blood using the anticoagulant sponges
The efficacy of the MS@D-HMP in animals were investigated using healthy New Zealand rabbits. In vitro tests were conducted using rabbit

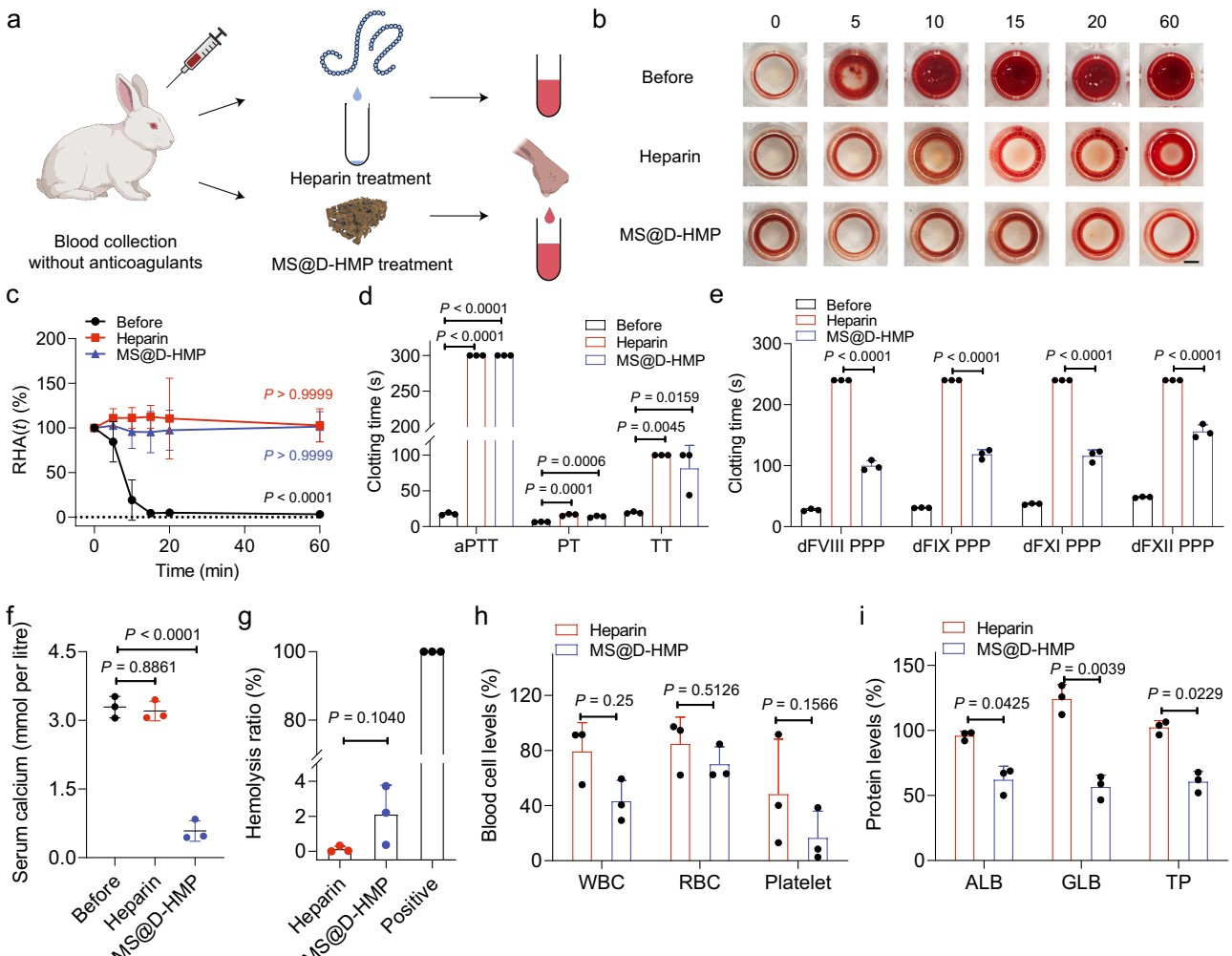

**Fig. 6 | Blood collection and anticoagulation using MS@D-HMP in rabbits.**
**a** Schematic of whole blood collection and instantaneous anticoagulation using heparin or MS@D-HMP in rabbits (created with BioRender.com). **b** Images of WBCTs for heparin-treated group and MS@D-HMP-treated group. The collected blood (150 μL/well) was incubated in a 96-well polystyrene plate, and the un-clotted blood was washed away by PBS (150 μL) after different time intervals. The collected blood without anticoagulant (before) was shown as comparison. Scale bars, 2 mm. **c** Relative hemoglobin absorbance RHA(*t*) plot for the collected whole blood in heparin-treated group and MS@D-HMP-treated group at different time intervals. The free RBCs in the un-clotted blood was resuspended in 1 mL deionized water, and hemoglobin concentration was measured at an absorbance λ = 540 nm (*n* = 3 biologically independent samples, mean ± SD. Two-way ANOVA with Geisser−Greenhouse correction and Bonferroni post-hoc tests). **d** aPTT, PT and TT values for heparin-treated group and MS@D-HMP-treated group (*n* = 3 biologically

independent samples, mean ± SD. One-way ANOVA with Bonferroni post-hoc tests). **e** Activity of intrinsic coagulation factors measured by titration experiments in the dFVIII, dFIX, dFXI and dFXII PPP for heparin-treated group and MS@D-HMP-treated group (*n* = 3 biologically independent samples, mean ± SD. One-way ANOVA with Bonferroni post-hoc tests). **f** Change in the serum calcium concentrations after treatment with heparin or MS@D-HMP (*n* = 3 biologically independent samples, mean ± SD. One-way ANOVA with Bonferroni post-hoc tests). Hemolysis ratio (**g**), blood cell levels (**h**) and protein levels (**i**) of the collected blood in heparin-treated group and MS@D-HMP-treated group (*n* = 3 biologically independent samples, mean ± SD. One-way ANOVA with Bonferroni post-hoc tests for (**g**). Paired, two-tailed student's *t*-test for (**i**) and RBC and Platelet in (**h**). Two-tailed Wilcoxon matched-pairs signed-tank test for WBC in (**h**)). Source data are provided as a Source Data file.

whole blood, comparing the instantaneous anticoagulant properties of MS@D-HMP to heparin. Blood was collected from each animal, and either in heparin- or MS@D-HMP-treated group (Fig. 6a). To visually assess clot formation, whole blood clotting times (WBCTs) were measured. While blood clotted within 5 min in its pristine state, both heparin-treated and MS@D-HMP-treated groups exhibited WBCTs exceeding 60 min (Fig. 6b). Free hemoglobin from unconstrained RBCs that were not trapped in the clots was extracted through PBS washing, and then released into deionized water[52]. The pristine blood showed a dramatic decrease in the free RBCs within 10 min, whereas the heparin-treated and MS@D-HMP-treated groups showed no obvious changes within 60 min, confirming that no blood clot was formed (Fig. 6c).

Both heparin-treated and MS@D-HMP-treated groups demonstrated significant prolongation for aPTT (300 s, detection limit) and

TT (100 s, detection limit) (Fig. 6d). Less prolongation was observed for PT (16.4 and 13.9 s), which may be due to the inhibited thrombin in common pathway. The inactivation of intrinsic coagulation factors was confirmed by titration experiments in the FVIII-, FIX-, FXI-, and FXII-deficient plasma (dFVIII, dFIX, dFXI and dFXII PPP). Delayed initiation of coagulation was observed when MS@D-HMP-treated plasma was added to the corresponding coagulation factor-deficient plasma, suggesting the inactivation of coagulation factors (Fig. 6e). Conversely, the addition of heparin-treated plasma completely abolished the clotting behaviors of plasma due to its potent ATIII-activating activity. The serum calcium concentration in the MS@D-HMP-treated group decreased significantly compared to the pristine blood (Fig. 6f).

For the MS@D-HMP-treated group, the hemolysis ratio was 2.1% (Fig. 6g). The blood cell levels were 43.1% for WBCs, 70.1% for RBCs and

16.5% for platelets, respectively. The reduced blood cells were mainly trapped by the sponge instead of adsorbing on the surface, which can be released by washing process as demonstrated by in vitro experiments (Fig. 5a). The protein levels were 62.0% for ALB, 56.6% for GLB and 60.6% for TP, respectively (Fig. 6h, i). For the heparin-treated group, the hemolysis, blood cell levels and protein levels obtained were theoretical values that do not align with actual clinical observations. These values heavily depend on the surgical parameters, including suction force, rate, and technique, etc[12].

### Efficacy and safety of whole blood auto-transfusion using the anticoagulant sponges in vivo

To investigate whether the MS@D-HMP could enable whole blood auto-transfusion, we conducted a systematic study of blood collection and reinfusion safety using a rabbit femoral artery hemorrhage model. The experimental design is outlined in Fig. 7a. After inducing an arterial hemorrhage by cutting a medium incision (approximately 30% of its circumference) into the femoral artery of a male rabbit under anesthesia, we collected the spurting blood using the MS@D-HMP without adding any extra anticoagulant. The collected blood was promptly transferred to a 50-mL centrifuge tube and monitored for clot formation over a 2 h period. Robust anticoagulation was sustained throughout this duration (Supplementary Movie 1 and Supplementary Fig. 29), and no clot formed on the sponge surface (Supplementary Fig. 30). For the heparin-treated group, blood was drawn from the injured artery using a 5-mL syringe, and administrated heparin through commercial vacuum tubes. Then, the cut artery was compressed to promote hemostasis, and ligated with 5-0 absorbable sutures to prevent potential bleeding. 1 h after blood collection, the rabbits were reinfused with the collected blood (-15 mL) through the right ear vein at a constant speed of 20 mL/min. The rabbits regained consciousness within 1 h following the auto-transfusion procedure, and remained healthy for over a month (Supplementary Movie 2 and Supplementary Fig. 31).

In the MS@D-HMP-treated group, clotting times returned to normal levels within 20 min after reinfusion, as there was no residual anticoagulant in the body (Fig. 7b, c). In contrast, the heparin-treated group showed higher aPTT (38.7 s) and TT (100 s, detection limit) values by the 20-min mark after reinfusion, likely due to the inhibitory effect of heparin on thrombin activity. Titration experiments at different time intervals suggested the restoration of intrinsic coagulation factors in both MS@D-HMP-treated and heparin-treated groups (Supplementary Fig. 32). No significant changes were observed in PT values and the concentration of fibrinogen for either group (Supplementary Fig. 33). To investigate coagulation activation and secondary fibrinolysis, we examined the levels of fibrin degradation products (FDP) and D-dimer in vivo, as the generated FXIIa and FXIa by the MS@D-HMP might activate plasminogen[53]. Both MS@D-HMP-treated and heparin-treated groups showed no significant changes in FDP and D-dimer levels pre- and post-reinfusion, confirming little effect on fibrinolytic system (Fig. 7d, e). The blood cell levels and biochemical parameters of rabbits were measured to detect acute or subclinical conditions. Both groups displayed leukocytosis on day 1 (particularly neutrocytosis) and secondary thrombocytosis on day 5, which could be attributed to acute infection or inflammation following surgery (Fig. 7f and Supplementary Table 10). After administering prophylactic antibiotics, leukocyte levels returned to normal on day 5. No significant changes were observed in other biochemical parameters pre- and post-treatment, including liver function, renal function, complement and immunoglobulin levels, blood lipid levels, and serum electrolyte levels, except for elevated serum alanine aminotransferase (ALT) and aspartate aminotransferase (AST) levels (Fig. 7g–l and Supplementary Data 3). The increased levels of ALT and AST level may be indicative of ischemia-reperfusion-induced liver injury following massive hemorrhage, which subsequently normalized on day 5.

## Discussion

The HMP-modified sponge described in our study exhibited rapid and spontaneous blood sorption along with coagulation inhibition via multiple pathways, which avoided suction-induced rupture of blood cells in the traditional intraoperative auto-transfusion setup[12], and associated thrombo-inflammatory response[40]. Importantly, the sponges could be easily removed, reducing the risk of excessive bleeding caused by anticoagulants. The blood collected using the sponge required no washing, allowing for the retention of valuable components such as platelets and plasma proteins. This may eliminate the need for plasma or platelet transfusion in cases of massive surgical bleeding, thereby simplifying the current intraoperative process. The sponge's portability and ease of handling also make it suitable for various scenarios, including battleground and car accidents. In such unexpected situations, only anticoagulant sponge can directly recover blood cells from the patient's wound outside hospital setting, highlighting its superiority over traditional auto-transfusion technology. However, it is important to consider contaminants like bacteria, fat, bowel contents or amniotic fluid in more complex environments[54], to avoid risks including sepsis, fat embolism syndrome, renal failure, disseminated intravascular coagulation, and potentially death[55,56]. The last part of blood with fat layers on top should be discarded, and further removal of fat particles could be achieved in conjunction with WBC depletion filters[57]. Considering the theoretical risk of tumor disseminating, whole blood auto-transfusion using sponges is still not recommended in cases of malignancy[58]. Independent of this, our results showed that in vivo reinfusion of sponge-collected blood in a highly sterile environment with prophylactic antibiotics administration may not cause detectable abnormality.

The coagulation inhibition achieved with the HMP-modified sponge was comparable to traditional heparin administration in our in vivo model, but heparin can return to the body during auto-transfusion. Heparinized coatings, commonly used to reduce thrombus formation on blood-contacting devices[59], rely on the binding of ATIII and are limited to surface interactions[60]. This limitation becomes apparent when heparinized coatings are applied in the context of whole blood auto-transfusion, where the collected blood is separated from the coatings, and thrombin continuously generates due to activation of intrinsic coagulation pathway arising from non-physiological shear, micro-embolic bubbles, and inorganic polyphosphate from activated platelets, etc[61,62]. Other polysaccharides structurally similar to heparin, such as carrageenan[63], fucoidan[64], dermatan sulfate[65] and alginate sulfate[36], have also been incorporated to blood-contacting surfaces to reduce thrombotic activity by mediating the activities of serine protease inhibitors (e.g., Atiii or hcii) on thrombin[66,67]. However, incorporating polysaccharides without losing activity and leaching presents challenges[21].

Since the last decade, heparin mimetics have been widely used to develop self-anticoagulant materials[24,68,69]. However, both the molecular anticoagulant mechanism of heparin mimetics and the anticoagulant mechanism of the heparin mimetics-modified materials have not been thoroughly explored. After introducing heparin mimetics onto the surface of a material, the anticoagulant mechanism at interface becomes even more elusive, especially with respect to the side effects, and the mechanism of surface-protein interaction which contributes to its anticoagulant function. Carboxylated and sulfonated polymers have been widely adopted for heparin-mimetic modification due to its well-established methods[24,70,71]. Through systematic exploration of the coagulation reactions at the molecular level, we revealed that the carboxylated and sulfonated HMP-modified sponges effectively adsorb FXI to the surface. Additionally, calcium ions were depleted from the plasma. An unknown thrombin inhibitor might be activated by sponge, even in the collected plasma when the sponge was removed. The detailed information about this inhibitor is still unclear and requires further exploration. Unlike heparin, which can be

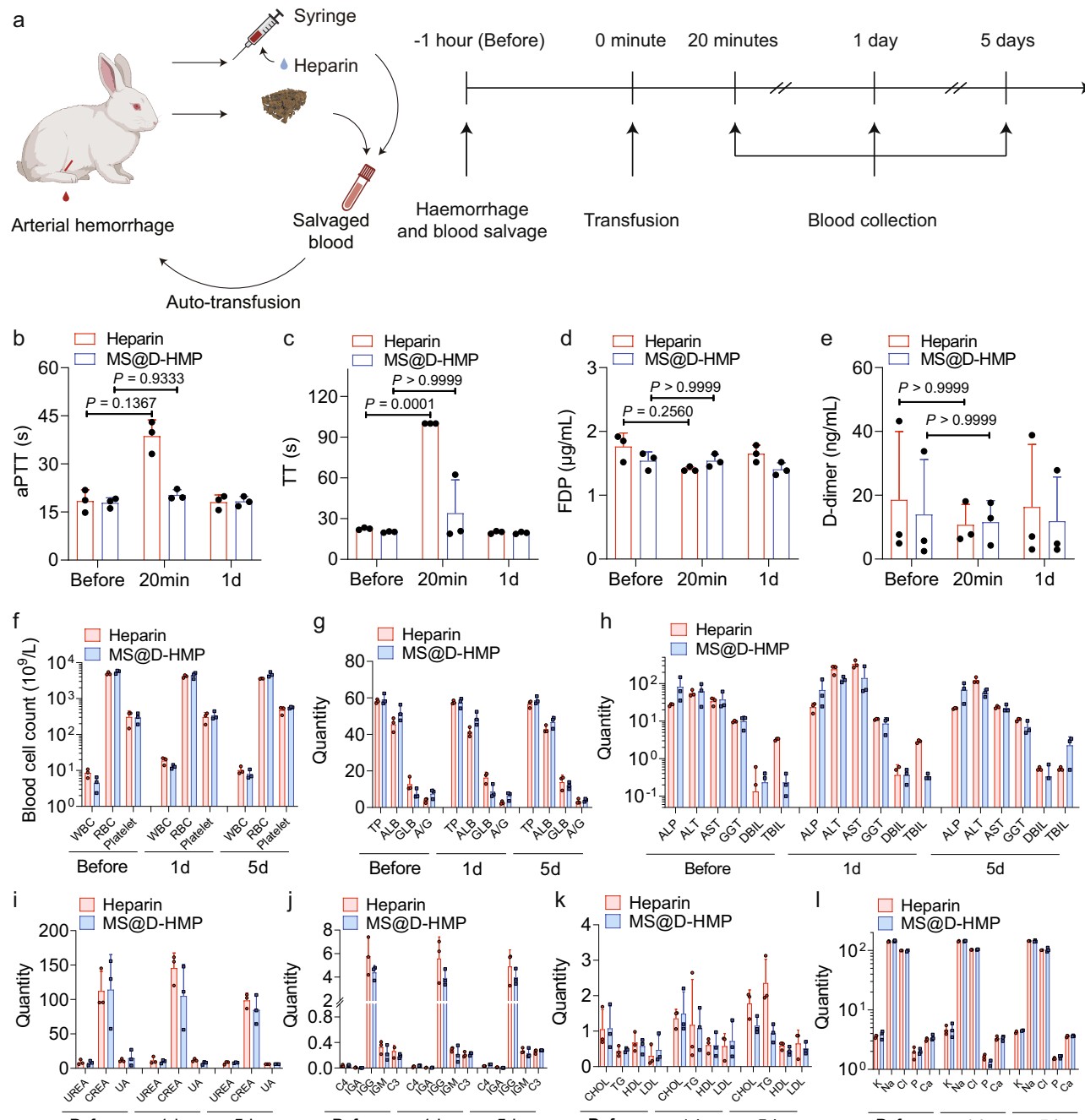

**Fig. 7 | Efficacy and safety of MS@D-HMP in a rabbit femoral artery hemorrhage model. a** Schematic of the whole blood auto-transfusion and experimental time line showing the sequence of events (created with BioRender.com). aPTT (**b**), TT (**c**), FDP (**d**) and D-dimer (**e**) for heparin-treated group and MS@D-HMP-treated group (n = 3 biologically independent samples, mean ± SD. Two-way ANOVA with Geisser–Greenhouse correction and Bonferroni post-hoc tests). **f** Blood cell count for heparin-treated group and MS@D-HMP-treated group (n = 3 biologically independent samples, mean ± SD). Analysis of liver function parameters (**g**, **h**), renal function parameters (**i**), serum complement and immunoglobulins levels (**j**), blood lipids levels (**k**) and serum electrolyte levels (**l**) of the rabbits in heparin-treated group and MS@D-HMP-treated group: **g** total protein (TP; g per litre); albumin (ALB; g per litre); globulin (GLB; g per litre); and albumin globulin ratio (A/G);

**h** alkaline phosphatase (ALP; units per litre); alanine aminotransferase (ALT; units per litre); aspartate aminotransferase (AST; units per litre); gamma-glutamyl transferase (GGT; units per litre); direct bilirubin (DBIL; μmol per litre); total bilirubin (TBIL; μmol per litre); **i** urea (UREA; mmol per litre); creatinine (CREA; μmol per litre); and uric acid (UA; μmol per litre); **j** complement component 4 (C4; g l⁻¹); complement component 3 (C3; g l⁻¹); immunoglobulin A (IGA; g l⁻¹); immunoglobulin G (IGG; g l⁻¹) and immunoglobulin M (IGM; g l⁻¹). **k** cholesterol (CHOL; mmol per litre); triglyceride (TG; mmol per litre); high-density lipoprotein (HDL; mmol per litre); low-density lipoprotein (LDL; mmol per litre); **l** potassium, sodium, chlorine, phosphorus and calcium (quantity in mmol per litre) (n = 3 biologically independent samples, mean ± SD). Source data are provided as a Source Data file.

reversed by antidotes (e.g., protamine and UHRA) via disrupting electrostatic interactions between heparin and ATIII[72], the depletion of FXI and inhibition of thrombin by the HMP-modified sponges are irreversible in vitro, which causes the persistent coagulation inhibition. Furthermore, in patients with reinfusion, the depleted coagulation

factors are replenished from storage pools, and thus restoring the hemostatic function in vivo.

With a combination of different anticoagulation mechanisms provided by the HMP-modified sponge, we ensure its efficiency and safety in whole blood auto-transfusion. The ability to inhibit thrombin

and chelate calcium ions is crucial for recovering blood with introduced tissue factor and prematurely activated thrombin, while the activation of intrinsic pathway via contact system during storage is blocked by depleting FXI and chelating calcium ions. The chelation of calcium ions by the sponge not only avoids activation-induced depletion of coagulation factors, but also helps inhibit the activation of complement system. Hence, valuable components are retained in the collected blood without activation and amplification of coagulation cascade and thrombo-inflammation response, which ensures the safe reinfusion of collected blood without anticoagulant.

Our work pioneers a strategy for engineering transient antithrombic and hemocompatible surfaces using HMP-modified coating, with potential applications in coating various blood-contacting devices including disposable catheters, blood storage bags, and short-term blood purification equipment. Meanwhile, the limitations of HMP-modified surfaces in extended applications, such as extracorporeal membrane oxygenation and biomedical implants, are clarified. The adsorption of FXI might decline along with the development of the loosely-bound protein layer on the surface. Combined with the regeneration of coagulation factors, the inhibition of intrinsic coagulation pathway by the sponge is gradually suppressed, which explains the return of coagulation factor levels in vivo after 2 h treatment in our previous work[29]. Hence, period of validity of HMP-modified surfaces should be carefully estimated to avoid anticoagulation failure and early termination of the treatment. Given our target application for blood auto-transfusion, we obtained results in a relatively static condition, while interfacial shear force and hemodynamics are of paramount significance in blood-purification materials and long-term implants, which might alter the dynamic exchange of protein layer[42]. Additionally, chelation of metal ions in vitro is beneficial for inhibition of coagulation and complement activation in our study. However, the reinfusion of increased volume of sponge-collected blood in vivo may induce hypocalcemia, which necessitates monitoring of calcium ions and calcium supplementation, as traditional massive transfusion protocols suggest[73]. Reconstruction of host electrolyte homeostasis in vivo is also of paramount importance when applying this surface to other fields such as blood purification, wherein the activation of contact system, complement system, and associated immune system that HMP-modified surfaces induce should be considered.

In summary, we demonstrated the effectiveness and safety of HMP-modified sponge towards whole blood auto-transfusion in vivo, and envision that the sponge can be potentially translated to clinics for use in intraoperative blood management. Beyond proof-of-concept, the generality and effectiveness of the mechanisms uncovered in this work could offer insights to improve the safety and hemocompatibility of antithrombotic surfaces, and thus help to move HMP-modified surfaces towards real-world applications.

## Methods

### Ethical statement
All in vitro and in vivo experiments were approved and performed by West China Hospital, Sichuan University, and all the experiments were performed in compliance with the relevant laws and national guidelines (GB/T 16886.4-2003/ISO 10993-4:2002, General Administration of Quality Supervision, Inspection and Quarantine of the People's Republic of China, Standardization Administration of the People's Republic of China). The procedure for collecting blood from human volunteers has been approved by the Institute of Blood Transfusion, Chinese Academy of Medical Sciences' Institutional Review Board (IRB, Ethics approval number: No.202024) with written consent from donors. All procedures involving the use of animals in this study were prospectively reviewed and approved by the Institutional Animal Care and Use Committee. This study was conducted in accordance with the National Institutes of Health Guide for the care and use of laboratory animals (NIH Publications No. 8023, revised 1978). This experiment

conformed to the legal requirement in China and was approved by the ethical committee (No. 2021911 A) of West China Hospital, Sichuan University. During the experimental period, the animals had free access to water and food.

### Preparation of the anticoagulant sponges
**Synthesis of *N*-(3, 4-dihydroxyphenethyl) acrylamide.** Dissolving dopamine hydrochloride (DOPA, 50 mM, Aladdin) and triethylamine (55 mM, Aladdin) in methanol (MeOH, 100 mL, Chron Chemicals) under an ice bath, a tetrahydrofuran solution (6 mL, Chron Chemicals) of acryloyl chloride (60 mM, Aladdin) and a MeOH solution (11 mL) of triethylamine (72 mM) were alternately added. The resulting reaction mixture was then kept at around 25 °C for 1 h with continuous stirring. After the completion of the reaction, the solvent was removed through rotary evaporation. The residue obtained was further dissolved in ethyl acetate (Chron Chemicals), and the solution was subjected to thorough washing with hydrochloric acid (Chron Chemicals) and saturated brine (Aladdin). Subsequently, the organic phase was carefully collected and dried with anhydrous sodium sulfate (Best-Reagent) before being concentrated using rotary evaporation. A series of meticulous recrystallization steps were performed to yield the final product, *N*-(3,4-dihydroxyphenethyl) acrylamide (DOPAm).

**Modification of the sponges.** Melamine sponge (MS, Alibaba) was ultrasonically cleaned, rinsed by ethanol (Chron Chemicals) and deionized water (prepared by a Millipore purification system, Billerica), and blew dried by $N_2$ before use. The vinylated sponge (MS@D) was produced by a mussel-inspired method with minor modifications. Typically, the MS was immersed in the Tris-buffer solution (pH = 8.5, 50 mM, Aladdin) containing DOPAm (5 mg/mL) and DOPA (1 mg/mL) for 12 h with oscillation to obtain MS@D. After washing and drying, anticoagulant sponge (MS@D-HMP) was obtained by immersing MS@D in reaction solution containing acrylic acid (AA, 12 g, Aladdin), 2-acrylamido-2-methyl-1-prroanesulfonic acid (AMPS, 8 g, Aladdin), ammonium persulfate (APS, 0.1 g, Aladdin) and deionized water (80 mL) at 70 °C for 10 h. The optimum ratio of sulfonic and carboxyl groups for the heparin-mimetic polymer was based on our previous study[29]. Briefly, a series of HMP-based surfaces were prepared using monomers (AA, providing carboxyl group; AMPS, providing sulfonic groups) with different feed mass ratios by free radical polymerization. The prepared surfaces were designated as $A_xM_y$, where "x" and "y" represents the feed mass of AA and AMPS, respectively. PRTs were used to evaluate the anticoagulant performance of the surfaces. Among all the surfaces, $A_3M_2$ exhibited the best anticoagulant performance. Hence, a feed mass ratio of 3:2 was considered as the optimum ratio of AA to AMPS in the HMP-based surfaces. Then, the MS@D-HMP was pulled out and immersed in plenty of sodium hydroxide (NaOH, 10 mM, Aladdin) solution for 1 day to achieve complete deprotonation. To remove the unreacted molecules thoroughly, the MS@D-HMP was washed for 5 days using deionized water. Finally, the MS@D-HMP were kept in phosphate buffered saline (PBS, pH 7.4, Aladdin) until use.

**Preparation of soluble heparin-mimetic polymers and polydopamine for comparison.** For the synthesis of soluble heparin-mimetic polymers (HMP), AA (12 g), AMPS (8 g), APS (0.1 g) and deionized water (80 mL) were mixed to obtain homogeneous solution. The obtained solution was heated to 70 °C for 10 h to complete the reaction. Then, the reaction solution was adjusted to neutral under magnetic stirring. HMP was obtained by dialyzing the solution against deionized water for 48 h, and dried by lyophilization. For the synthesis of polydopamine (PDA), Tris-buffer solution (pH = 8.5, 50 mM) containing DOPAm (5 mg/mL) and DOPA (1 mg/mL) was kept for 12 h with oscillation at around 25 °C. The PDA particles were separated by centrifugation ($27800 \times g$) and washed multiple times with deionized water.

## Characterization of the sponges

$^1$H and $^{13}$C nuclear magnetic resonance spectra (NMR) were obtained in DMSO-$d_6$ on a Bruker Avance AV-400 MHz spectrometer (Bruker, Leipzig, Germany). The surface-morphologies of these sponges were obtained by scanning electron microscopy (SEM, Phenom Pure, Phenom World, Netherlands). The surface compositions of the sponges were observed by energy dispersive spectra (EDS, Mahwah, NJ, USA) and X-ray photoelectron spectroscopy (XPS, ThermoFischer Scientific, ESCALAB Xi+, USA). The compositions of the sponges were also investigated using Elemental Analyzer (Euro EA 3000), Fourier transform infrared spectroscopy (FTIR, Nicolet 560, USA) and Thermogravimetric analyzer (METTLLE TOLEDO, TGA/DSC 3+, Switzerland). The surface microstructures of the sponges were observed by FESEM (FESEM, Apreo S HiVac FEI).

Mercury intrusion measurement was acquired from an automated mercury intrusion porosimeter (PoreMaster 33, Quantachrome, USA). A Zetasizer (Nano-ZS90, Malvern, UK) was used to measure the zeta potentials of the sponges (ground to powders and dispersed in the PBS at a concentration of 1 mg/mL). A universal testing machine (CMT4000, China) equipped with a 100 N load mechanical sensor was used to evaluate the compressive properties of the sponges (compression rate at 1 mm per min). Permeating behaviors of one water droplet on the surface of the sponges were monitored on an optical tensiometer (Theta T200, Biolin Scientific, Sweden). OneAttension 2.7 was used to collect the data. The liquid sorption coefficient $K_s$ of the sponges for deionized water and heparinized (2 IU/mL) goat whole blood (Hongquan Biotech.) was measured by a wicking method[32].

## Blood collection and plasma preparation

Fresh human blood was draw from three 24-year-old healthy male donors into vacuum tubes containing 3.8% sodium citrate (Kangjian Inc., Jiangsu, China, 1:9) for clotting time tests, contact activation and kallikrein/kinin system, or containing recombinant hirudin (Boatman Biotech., 30 μg/mL) for complement activation and monocyte activation, or containing ethylene diamine tetraacetic acid (EDTA, Kangjian Inc., Jiangsu, China) for blood count assay. Blood samples were subjected to centrifugation to yield platelet-rich plasma (PRP, $150 \times g$, 15 min) or platelet-poor plasma (PPP, $1000 \times g$, 15 min).

## Anticoagulant behaviors of sponges in vitro

**Clotting times.** Certain amounts of sponges were incubated at 37 °C with 300 μL fresh citrate-anticoagulated PPP for 30 min. Then, the PPP was collected, and the clotting time tests were conducted using an automated coagulometers (CA-500 series, Sysmex, Japan) with its built-in testing program.

**Plasma recalcification times.** 300 μL fresh citrate-anticoagulated PPP was re-calcified with 5 μL CaCl$_2$ solution (final concentration at 12.5 mM), and then incubated with the sponges. The incubated PPP was taken out within 5 s, and monitored for clot formation with the naked eye. Plasma recalcification times (PRTs) were documented upon initial detection of fibrin threads. The plasma was considered as non-coagulation when the PRT was over 180 min. Coagulation was analyzed for every 30 s between the two readings to increase accuracy. The serum calcium concentration before and after treatment was assessed using a Cabas C311 biochemical analyzer (Roche, Switzerland). To eliminate the influence of calcium ion adsorption by the sponges on the prolongation of PRTs, 300 μL fresh PPP was incubated with the sponges for 30 min. Then, the sponges were taken out. The treated PPP was re-calcified with 5 μL CaCl$_2$ solution (final concentration at 12.5 mM), and monitored for clotting.

**The calcium ions adsorption by MS@D-HMP in the calcium ions solution and plasma.** To investigate the calcium ions adsorption kinetics in calcium ions solution, 20 mL calcium ions solution (2 mM) was incubated with 20 mg MS@D-HMP at 37 °C for a certain time (5, 15, 30, 60, 240, 480, 1440 min). Atomic absorption spectrophotometer (SPCA-626D, Shimadzu, Japan) was used to detect the residual concentrations of calcium ions.

To investigate the calcium ions adsorption in plasma, 400 μL hirudin anticoagulant plasma with varying initial calcium ion concentrations was prepared. The re-calcified plasma was incubated with 10 mg MS@D-HMP for 10 min. A Cabas C311 biochemical analyzer (Roche, Switzerland) was used to detect the residual calcium ions in the treated plasma. This is also a preliminary experiment for the posterior fluorescence-based FIXa activity assay as FXIa activates FIX to FIXa in a calcium-dependent way.

**Thromboelastography.** Whole blood anticoagulant properties of the sponges were measured on a thrombosis viscoelastic analysis system (ImproveClot T-400, Improve Medical Instruments Co., Ltd, China). 400 μL fresh citrate-anticoagulated blood was supplemented with CaCl$_2$ solution (23.5 μL, final Ca$^{2+}$ concentration at 11 mM). The resulting blood mixture was promptly introduced into 40 mg MS@D-HMP, squeezed out (within 5 s) and transferred into the thromboelastography (TEG) cup for test. To exclude the influence of the adsorption of calcium by the sponges on the TEG, 400 μL fresh whole blood was subjected to a 30-min incubation with the sponges. Subsequently, 340 μL incubated whole blood was collected and recalcified by adding CaCl$_2$ solution (20 μL, final Ca$^{2+}$ concentration at 11 mM) to induce coagulation.

**Thrombin generation assay.** 100 μL fresh whole blood was added to a 24 well plate and recalcified by adding CaCl$_2$ solution (2.5 μL, final Ca$^{2+}$ concentration at 10 mM) to initiate thrombin generation. For the sponge groups, the re-calcified whole blood was introduced to 10 mg sponges, squeezed out (within 5 s) and immediately transferred into the plate. PBS-treated and citrate-treated whole blood were employed as normal control and positive controls, respectively. The whole blood was subjected to centrifugation to obtain PPP after a 30-min incubation at 37 °C. The concentration of generated thrombin was measured using a Human Thrombin-Antithrombin Complex AssayMax ELISA Kit (Assay Pro, USA).

## Mechanistic exploration of inactivation of coagulation factors by the sponges

**Detection of the activities of coagulation factors in normal plasma.** Certain amounts of sponges were incubated with 300 μL fresh citrate-anticoagulated PPP at 37 °C for 30 min. The activities of corresponding intrinsic coagulation factors were determined based on aPTT. The analysis was performed using intrinsic corresponding factor-deficient plasma (Siemens) in an automatic coagulation analyzer (ACL Elite Pro, werfan) equipped with built-in procedures.

**Detection of the activities of coagulation factors in FXII-deficient plasma.** 30 mg MS@D-HMP was subjected to incubation at 37 °C with 300 μL of FXII-deficient plasma for 30 min. Then, the activities of FXI, FIX and FVIII in the incubated plasma were determined.

**Proteomic identification of proteins in normal plasma and MS@D-HMP treated plasma.** 30 mg MS@D-HMP was incubated with 300 μL fresh PPP for 30 min. The incubated plasma was centrifuged at 4 °C and 12,000 × g for 10 min to remove cellular debris. The resulting supernatant was carefully transferred to a new tube, and the protein concentration was quantified utilizing the BCA assay kit (Beyotime).

For digestion, the volume of the lysate was adjusted to a consistent level, and dithiothreitol (Sigma-Aldrich) was added to attain a final concentration of 5 mM. The samples were then incubated at 56 °C for 30 min. Following reduction, iodoacetamide (Sigma-Aldrich) was added to achieve a final concentration of 11 mM, and the samples were

incubated at room temperature in the dark for 15 min to facilitate alkylation. Subsequently, the alkylated samples were carefully transferred to ultrafiltration tubes and centrifuged at room temperature, 12,000 × g for 20 min. Following centrifugation, the samples underwent three washes with 8 M urea (Sigma-Aldrich), followed by three washes with displacement buffer to effectively remove urea. To initiate enzymatic digestion, trypsin (Promega) was added in a 1:50 ratio (enzyme:protein, w/w), and the samples were allowed to undergo overnight digestion.

The peptide fragments were dissolved in mobile phase A and separated using the EASY-nLC 1200 ultra-high-performance liquid chromatography system (Thermo Fisher Scientific). Mobile phase A consisted of an aqueous solution containing 0.1% formic acid (Fluka) and 2% acetonitrile (ThermoFisher Scientific), while mobile phase B consisted of an aqueous solution containing 0.1% formic acid and 90% acetonitrile. The liquid phase gradient was set as follows: 0–96 min, 4%～20% B; 96–114 min, 20%～32% B; 114–117 min, 32%～80% B; 117–120 min, 80% B, and the flow rate was maintained at 500 nL/min.

The peptides were subjected to capillary source followed by the Orbitrap Exploris™ 480 (ThermoFisher Scientific) mass spectrometry. The ion source voltage was adjusted to 2.3 kV, and the FAIMS compensation voltage (CV) was set to −45 V and −70 V. Both the precursor ions of the peptides and their secondary fragments were detected and analyzed using the high-resolution Orbitrap. The first-stage mass spectrometry scan range was set from 400 to 1200 m/z with a scan resolution of 60000. The fixed starting point for the second-stage mass spectrometry scan was 110 m/z, and the scan resolution for the second stage was set to 30000. TurboTMT was disabled during the analysis. Data acquisition was performed using a data-dependent acquisition (DDA) method. After the first-stage scan, the 15 most intense precursor ions were sequentially selected and fragmented in the HCD collision cell using 27% collision energy. Subsequently, their secondary mass spectra were analyzed in the same order. To improve mass spectrometry efficiency, the automatic gain control (AGC) was set to 75%, and the signal threshold was set to 1E4 ions/s. The maximum injection time was set to 100 ms, and a dynamic exclusion time of 30 s was employed for tandem mass spectrometry scans to prevent redundant scanning of precursor ions.

The resulting MS/MS data were processed using MaxQuant search engine (v.1.6.15.0). Tandem mass spectra were searched against the human SwissProt database (20422 entries) concatenated with reverse decoy database (https://www.uniprot.org/proteomes?facets=proteome_type:1&query=(organism_id:9606); proteome ID: UP000005640; release numbers: 2021_01/2021_01). Peptides underwent trypsin cleavage, with a maximum allowance of 2 missed cleavage sites. The peptide mass tolerance was set at 20 ppm in the first search and 5 ppm in the main search. The fragment mass tolerance was established at 0.02 Da. The false discovery rate was specified to be less than 1%. Protein intensity (I) was the sum of all identified peptide intensities of the specific protein. The relative percentage of protein based on total proteins was calculated using Eq. (1):

$$\text{Relative percentages of protein}\,(\%) = \frac{I_j}{\sum I_j} \times 100\% \qquad (1)$$

where $j$ represents the protein.

Relative abundance ($R$) was calculated to provide the centralized transformation of protein intensity across all samples based on Eq. (2):

$$R_{ij} = \frac{M_{ij}}{\text{Mean}\left(M_j\right)}, \; M_j = \frac{I_{ij}}{\text{Median}\left(I_i\right)} \qquad (2)$$

where $i$ and $j$ represent the sample and the protein, respectively.

**Identification of tightly-bound proteins on the surface of MS@D-HMP.** To investigate the impact of surface proteins on the anticoagulant properties of MS@D-HMP, sponges underwent two stages of treatments with two successive incubations in each stage. In stage 1, 30 mg MS@D-HMP was firstly incubated with 300 μL PPP for 30 min, and the PPP was collected for aPTT tests and LC-MS/MS analysis. Then, the used MS@D-HMP was secondly incubated with 300 μL PPP for 30 min and the PPP was collected for aPTT tests. The MS@D-HMP were washed with PBS for 24 h with oscillation, and in-situ digested for proteomics identification of tightly-bound proteins on the surface of MS@D-HMP. In stage 2, the MS@D-HMP that underwent two successive incubations and PBS washing in stage 1 was used to replace the fresh MS@D-HMP and repeat the same procedure in stage 1. Note that the weight of adsorbed proteins may differ between the two stages (Proteins were in-situ digested into peptides, which were loaded with equal volume), the protein intensity may provide a better description of increased proteins than the relative percentages of proteins.

**The equilibrium binding capacity and the corresponding inactivation efficiency of FXI by sponge.** A certain amount of FXI native protein (Abcam) was mixed with 100 μL FXI-deficient plasma to obtain the final FXI concentrations of 2.5–80 μg/mL. The FXI-supplemented plasma was subjected to incubation at 37 °C for 30 min, followed by an additional 30-min incubation with 6 mg of MS@D-HMP at the same temperature. The residual FXI concentration of the incubated plasma was assessed using a standard curve correlating FXI concentration with FXI activity. Gradient-diluted FXI-supplemented plasma was prepared, and the resulted activity was detected using an automatic coagulation analyzer (ACL Elite Pro, werfan). The equilibrium binding capacity and the corresponding inactivation efficiency can be calculated as follows:

$$\text{Equilibrium binding capacity}\,(\mu g/mg) = \frac{V(C_o - C_e)}{m} \qquad (3)$$

$$\text{Inactivation efficiency}\,(\%) = \frac{C_O - C_e}{C_O} \times 100\% \qquad (4)$$

where $C_O$ and $C_e$ represent the initial and final concentrations of FXI after incubation with MS@D-HMP. $V$ denotes the volume of plasma; $m$ signifies the weight of MS@D-HMP used.

**Detection of the procoagulant behaviors of FXI via fluorescence-based FIXa activity assay.** To investigate the potential procoagulant behaviors of FXI in the MS@D-HMP treated plasma and on the surface of MS@D-HMP, the activity of FIXa in the plasma was determined by a Fluorescence-based Factor IXa Activity Assay Kit (BioVision Inc., USA). If the MS@D-HMP induced the generation of FXIa via contact pathway or direct cleavage, FIX could be cleaved by FXIa and can be easily quantified. Since the cleavage of FIX by FXIa is also calcium-dependent, a preliminary experiment was conducted to eliminate the potential impact of calcium ion adsorption by the sponges. As shown in Supplementary Table 6, supplement of calcium ions in a dose of 5 mM was appropriate for maintaining the normal level of calcium ions. Actin (aPTT reagent, Siemens) was used as positive control. Citrate was used to block the probable activation of FIX by FXIa. The dose of citrate was only sufficient for blocking the probable activation of FIX by FXIa, because the assay kit supplied enough calcium for the latter FX to FXa. Similar experiments were performed in the FXII-deficient plasma to evaluate the effect of contact activation on the generation of FXIa and FIXa. The FXII-deficient plasma was obtained with sodium citrate as anticoagulant. Hirudin (30 μg/mL) was added into the FXII-deficient plasma to prevent potential activation of FXI by thrombin. The detailed experimental setup was shown in Supplementary Table 8.

**Evaluation of activation of kallikrein/kinin system.** Citrate-anticoagulated plasma was supplemented with captopril (0.1 μg/mL, Aladdin) to inhibit the degradation of bradykinin. Then, 10 mg MS@D-HMP was subjected to incubation with 200 μL plasma at 37 °C for 20 min. The resultant plasma was mixed with ice cold ethanol in a ratio of 1:4. The generated bradykinin was quantified using a Bradykinin ELISA Kit (Abcam). Untreated plasma was used as negative control, while plasma treated with glass power (1 mg/100 μL plasma, Supelco Analytical) for 20 min was employed as positive control.

**Activation of factor XII by MS@D-HMP in chromogenic assays.** To investigate whether MS@D-HMP can directly induce activation of FXII, FXIIa chromogenic substrate S-2302 experiment was performed using FXII purified protein. For the positive control, a mixture of 25 μL Actin (aPTT reagent, Siemens) and 50 μL FXII/PBS solution (500 nM) was prepared. As for the negative control, a mixture of 25 μL PBS and 50 μL FXII/PBS solution (500 nM) was prepared. For the sponge, 50 μL of PBS was added to 100 μL of FXII/PBS solution (500 nM) and then incubated with 5 mg of MS@D-HMP. After a 15-min incubation period, 50 μL of the incubated solution was combined with 100 μL substrate S-2302 (Chromogenix, 4 mM). Following a 30-min incubation, the resulting mixture was promptly transferred to a 96-well plate, and the optical density was measured at 405 nm using a plate reader (Thermo Scientific, USA).

The FXIIa amidolytic activity in human plasma with or without the sponges was assessed. For sponge, 100 μL normal saline was added into 200 μL citrate-anticoagulated PPP, and incubated with 10 mg MS@D-HMP. Then, 100 μL 4 mM substrate S-2302 was added. For the positive control, 100 μL Actin (aPTT reagent, Siemens) was introduced into 200 μL citrate-anticoagulated PPP, and then mixed with 100 μL 4 mM substrate S-2302. For the negative control, 100 μL normal saline was mixed with 200 μL citrate-anticoagulated PPP and 100 μL 4 mM substrate S-2302. Following incubation at 37 °C for 15 min, the optical density was detected at 405 nm in a plate reader (Thermo Scientific, USA).

**Detection of the concentration of fibrinogen in plasma.** Certain amounts of sponges were incubated with 300 μL citrate-anticoagulated fresh PPP at 37 °C for 30 min. Then, the PPP was collected, and the concentration of fibrinogen was detected using an automated coagulometers (CA-500 series, Sysmex, Japan) with its built-in testing program.

**Correction of coagulation function by clotting times.** 30 mg MS@D-HMP was incubated at 37 °C with 300 μL citrate-anticoagulated PPP for 30 min. For the correction assay involving fibrinogen replenishment, a specific quantity of human fibrinogen (Adamas life) was introduced into the treated PPP based on the reduced fibrinogen concentration, and the measurement of TT values was performed both before and after the addition of fibrinogen. For the correction assay involving normal plasma replenishment, a specific quantity of fresh normal PPP was introduced into the treated PPP, and the clotting times were measured. Citrate-anticoagulated and heparin-anticoagulated PPP were employed as controls.

**The leakage of the coating from the sponge and its effect on clotting time.** 50 mg MS@D-HMP was incubated with 10 mL PBS buffer for varying time intervals (1 day, 3 days, and 7 days). The eluate was collected, and the UV-vis spectra of eluate were acquired using a spectrophotometer (Shimadzu UV-1750), within the wavelength range of 190 to 400 nm. PDA (0.0625–1 mg/mL) and HMP (0.0625–1 mg/mL) were used as comparisons. Then, 20 μL eluate was mixed with 200 μL PPP, and clotting times were tested.

**The effect of sponge on ATIII and HCII.** For ATIII, 10 mg MS@D-HMP was incubated with 100 μL ATIII-deficient plasma (BioMedica Diagnostics Inc.) for 30 min. The incubated plasma was collected to determine the aPTT, TT and FXI activity. For comparison, heparin was added into the ATIII-deficient plasma at a concentration of 0.1 IU/100 μL PPP (aPTT > 600 s, TT > 240 s for heparin-anticoagulated normal PPP at this concentration). The FXI activities were determined based on aPTT.

For HCII, 6 mg MS@D-HMP was incubated with 60 μL HCII/PBS solution (0.34 μM, ABclonal) for 10 min, and 50 μL incubated solution was collected for HCII + MS@D-HMP group. For comparison, 50 μL PBS solution was incubated for 10 min for blank group. 50 μL HCII/PBS solution (0.34 μM) was incubated for 10 min for HCII group. 50 μL heparin/PBS solution (0.2 IU/μL, Adamas) was incubated for 10 min for heparin group. 50 μL HCII/heparin/PBS solution (0.34 μM for HCII, 0.2 IU/μL for heparin) was incubated for 10 min for HCII+heparin group. These incubated solutions were mixed with 20 μL FIIa (3 IU/mL, Siemens), and incubated for another 5 min. Finally, fibrinogen solution (Bioss) was introduced, resulting in a concentration of 2.5 mg/mL. The turbidity change was measured at 350 nm in a plate reader for 30 min (Thermo Scientific, USA). The residual FIIa activity was calculated based on the standard curve using gradient-diluted FIIa solution.

## Hemocompatibility

**Blood count assay in vitro.** 25 mg MS@D-HMP was incubated with 400 μL EDTA-anticoagulated fresh whole blood for 30 min. The blood was collected, and blood count assays were performed using a hematology analyzer (BC-5100, Mindray). The numbers of blood cells trapped in the sponge was determined by washing. 400 μL whole blood was incubated with 25 mg MS@D-HMP, and the whole blood was collected for blood count assays. Then, the MS@D-HMP was washed with 200 μL PPP twice, and the plasma was collected for blood count assays. Finally, the collected blood cell levels after washing were calculated.

**Red blood cell morphology and hemolysis.** Citrate-anticoagulated blood was diluted with a 2-fold PBS solution, and then centrifuged at $644 \times g$ for 10 min to separate the red blood cells (RBCs). The washing procedure was repeated five times. Subsequently, a 1 mL RBC suspension (at a concentration of $10^8$ cells/mL) was incubated with 50 mg MS@D-HMP at 37 °C. Deionized water and PBS served as the positive and negative controls, respectively. After a 3-h incubation, the suspension underwent centrifugation at $10,304 \times g$ for 3 min. The resulting supernatant was collected, and RBC morphology was examined using a microscope (Zeiss Axioskop 2 Plus). The absorbance of supernatant was measured at 540 nm using a UV-vis spectrometer (UV-1750, Shimadzu). The hemolysis ratio was determined using the following formula:

$$\text{Hemolysis ratio} \, (\%) = \frac{A_{s} - A_{n}}{A_{s} - A_{n}} \times 100\% \tag{5}$$

where $A_s$, $A_p$ and $A_n$ represents the absorbance values of sponge, positive and negative control, respectively.

**Platelet adhesion and platelet activation analysis.** For platelet adhesion analysis, 10 mg MS@D-HMP was incubated with 200 μL citrate-anticoagulated fresh PRP at 37 °C for 2 h. Then, the sponges were rinsed with PBS solution. The adhered platelets to the sponges were fixed using a 2.5 wt% glutaraldehyde (Chron Chemicals)/PBS solution, followed by a gradient dehydration process. The obtained samples were then observed using a FESEM (FESEM, Apreo S HiVac FEI). Lactate dehydrogenase (LDH) assay was used to quantify the

platelet adhesion. The sponges incubated with PRP were further treated with Triton X-100 (1.8% v/v, Aladdin)/PBS solution for 1 h at 37 °C to lyse the adherent platelets and release LDH. The LDH activity in the resulting supernatant was measured utilizing a LDH Cytotoxicity Assay Kit (Cayman, USA). To determine platelet quantities, serial dilutions of platelet suspension were prepared in Tyrode's buffer, and the corresponding LDH activity was measured to create a calibration curve.

To assess platelet activation, 10 mg MS@D-HMP was incubated with 200 μL citrate-anticoagulated whole blood at 37 °C. Then, the blood was centrifuged to separate PPP, and the generated platelet factor 4 (PF4) was quantified using a Human PF4 ELISA kit (Ray-Biotech.). Platelet activation was also assessed using flow cytometry. 15 mg MS@D-HMP was incubated with 200 μL citrate-anticoagulated PRP in 1.5 mL polypropylene (PP) tube at 37 °C for 30 min. Negative control consisted of untreated PRP, while positive control involved PRP treated with thrombin receptor activating peptide (TRAP, Sigma-Aldrich, 0.1 mM). After a 30 min-incubation, aliquots of the PRP were collected. For the preparation of staining solution, 5 μL anti-CD41a-FITC (eBioscience™, #11-0419-42, clone HIP8) and 1.25 μL anti-CD62p-APC (eBioscience™, #17-0626-82, clone Psel.KO2.3) were added in 90 μL PPP. Then, 5 μL samples were diluted in 90 μL staining solution. Following a 15-min incubation in the dark, the samples were halted with flow cytometry staining buffer (300 μL, eBioscience™). The degree of platelet activation was acquired using a flow cytometer (BD FACSCelesta™) by gating platelet-specific events based on anti-CD41a-FITC. The mean fluorescence intensities of platelet activation marker CD62p-APC signal of platelet population were normalized to the intensity of TRAP-activated platelets. All experimental acquisition was performed using BD FACSDiva™ software. Data analysis used FlowJo 10.6.2 software (TreeStar Inc).

**Monocyte activation analysis.** 200 μL hirudin-anticoagulated whole blood was incubated at 37 °C with 15 mg MS@D-HMP in 1.5 mL PP tube. Negative control involved untreated whole blood, while positive control included whole blood treated with lipopolysaccharide (LPS, 10 ng/mL, Sigma-Aldrich). After 30 min, aliquots of the incubated whole blood were collected for assessment of the monocyte activation. 100 μL whole blood was stained with 5 μL anti-CD14-FITC (eBioscience™, #11-0149-42, clone 61D3) and 5 μL anti-CD11b-APC (eBioscience™, #17-0118-42, clone ICRF44). Following a 30-min dark incubation, 1.6 mL BD FACS™ Lysing solution was added to lyse red blood cells for 10 min and fix the remaining cells. Then, the samples were washed with 200 μL flow cytometry staining buffer (eBioscience™), and the volume was adjusted to 300 μL. The degree of monocyte activation was acquired using a flow cytometer (BD FACS-Celesta™) by gating monocyte-specific events based on anti-CD14-FITC. The mean fluorescence intensities of monocyte activation marker CD11b-APC signal of monocyte population were normalized to the intensity of LPS-activated monocytes. All experimental acquisition was performed using BD FACSDiva™ software. Data analysis used FlowJo 10.6.2 software (TreeStar Inc).

**Concentration of D-dimer.** Citrate-anticoagulated PPP (100 μL) was re-calcified with stock solution of calcium ions (final $Ca^{2+}$ concentration at 10 mM), and then incubated with 10 mg MS@D-HMP. After a 30-min incubation at 37 °C, the plasma was collected (sponge was removed for MS@D-HMP-treated group) and incubated for another 6 h. The plasma without recalcification was used as normal control. For positive control, re-calcified plasma was incubated for 1 h, and fibrinolysis was initiated by addition of human plasmin (80 ug/mL, Sigma-Aldrich) with 6 h incubation. Finally, the concentration of D-dimer was detected using a Human D-dimer ELISA Kit (Cusabio, CSB-E05175h).

**Complement activation analysis.** 10 mg MS@D-HMP were incubated with 200 μL hirudin-anticoagulated whole blood at 37 °C. For comparison, blood treated with cobra-venom factor (CVF) at a final concentration of 25 μg/mL (Hengfei Biotech.) served as the positive control, while PBS was used as the negative control. After a 30-min incubation, the complement activation was stopped by adding 5 μL EDTA solution (final concentration at 5 mM). The resulting blood was centrifuged to separate PPP, and the generated C3a and C5a were measured using Complement C3a Human ELISA Kit and Complement C5a Human ELISA Kit (Thermo Scientific, USA).

### Efficacy and safety of the sponges in animals

**Rabbit whole blood anticoagulation and salvage in vitro.** Healthy male New Zealand rabbits, approximately 4 months old, and weighing around 3–4 kg, were used for experiments. MS@D-HMP was subjected to thorough sterilization before use. Each rabbit was anesthetized using 3% sodium pentobarbital (1 mL/kg) through an injector placed in an ear vein. Certain amounts of rabbit whole blood were collected without anticoagulant medications, and immediately treated with MS@D-HMP (0.4 g/mL) or directly collected using commercial heparin vacuum tubes (Kangjian Inc., Jiangsu, China, final concentration at around 10 IU/mL).

The obtained whole blood was evenly divided into 6 portions (150 μL each) and then added into a 96-well tissue culture polystyrene plate one by one. 150 μL PBS was introduced, and the free blood cells were collected. The collected free blood cells were resuspended in 1 mL deionized water and incubated at 37 °C for 15 min. The clotting kinetics were assessed through the relative hemoglobin absorbance (RHA($t$)) plot. At clotting time $t$, the absolute hemoglobin absorbance (HA($t$)) was measured by the spectrometer at 540 nm. HA(0), representing the hemoglobin absorbance at $t = 0$ min, served as the reference. The relative hemoglobin absorbance at clotting time $t$, denoted as RHA($t$), was calculated as HA($t$)/HA(0). A lower hemoglobin absorbance value indicated faster clotting, as it implied that the hemoglobin originated from un-clotted red blood cells. The wells with or without the adhered blood clot were photographed. The process was then repeated for 0, 5, 10, 15, 20 and 60 min all the way. The blood was considered as non-coagulation when the WBCT was over 60 min.

The blood count assays were performed using an automated hematology cell analyzer (ADVIA 2120i, SIEMENS AG FWB, Germany). Clotting times were measured with an automatic coagulometers (CA-7000, Sysmex, Japan). Measurement of biological parameters were conducted in a Cabas C311 biochemical analyzer (Roche, Switzerland). Titration experiments for coagulation factors were conducted using the aPTT assay with corresponding factor-deficient plasma (Siemens) on an automatic coagulation analyzer (ACL Elite Pro, werfan).

**Efficacy and safety of whole blood salvage and refusion by the MS@D-HMP in vivo.** Before treatment, each rabbit was anesthetized using 3% sodium pentobarbital (1 mL/kg) through an injector placed in an ear vein. A 5 cm incision was made in the groin area, parallel to the left femoral artery's course, situated between the abdomen and thigh. The femoral artery was meticulously isolated, and subsequently, a medium incision (ca. 30% of its circumference) was created in the artery to induce arterial hemorrhage. Then, the spurting blood was collected by the sponges without anticoagulant medications and transferred into centrifuge tube. Heparinized whole blood was collected via 5-mL syringes and added into heparin vacuum tubes (Kangjian Inc., Jiangsu, China). The incised artery was compressed to facilitate hemostasis and then ligated using 5-0 absorbable sutures (Jinhuan Medical Products Co., LTD, China) to prevent any potential bleeding. Then, the surgical wounds were meticulously sutured. The rabbits were subsequently cared for and nourished for a specified period. The collected whole blood was stored at room temperature for

1 h. For reinfusion, approximately 15 mL of the collected whole blood was infused through the right ear vein at a constant rate of 20 mL/min. The sponges before and after treatment were fixed using a 2.5 wt% glutaraldehyde (Chron Chemicals)/PBS solution, followed by a gradient dehydration process. The obtained samples were then observed using a FESEM (FESEM, Apreo S HiVac FEI).

The blood count assays were performed using an automated hematology cell analyzer (ADVIA 2120i, SIEMENS AG FWB, Germany). Clotting times were measured with an automatic coagulometers (CA-7000, Sysmex, Japan). Measurement of biological parameters were conducted in a Cabas C311 biochemical analyzer (Roche, Switzerland). Titration experiments for coagulation factors were conducted using the aPTT assay with corresponding factor-deficient plasma (Siemens) on an automatic coagulation analyzer (ACL Elite Pro, werfan). The concentrations of fibrinogen degradation products (FDP) were detected by immunoturbidimetry in an automatic coagulation analyzer (ACL Elite Pro, werfan). The concentrations of D-dimer were detected by ELISA (Rabbit D-dimer ELISA Kit, Zeye Biotech., China).

## Statistical analysis

In general, data are expressed as mean ± SD., indicated by error bars in all graphs. Data was statistically analyzed with unpaired, two-tailed student's $t$-test (Figs. 4e, g, 5a, c, Supplementary Fig. 10), paired, two-tailed student's $t$-test (Fig. 6h, i) or one-way ANOVA followed by Bonferroni post-hoc tests (Figs. 1e, 2g, 4b–d, f, 5d, f–i, 6d–g, Supplementary Figs. 13, 15, 16, 19–23, 25). If data did not reach the criteria for parametric statistics, non-parametric statistics using Wilcoxon Signed-Rank Test (Fig. 6h) or Kruskal–Wallis test (Figs. 5e, j) followed by Dunn's multiple comparisons if two or three samples, respectively, were tested. In time course analyses, mixed two-way ANOVA with Bonferroni post-hoc tests were used (Fig. 3f, 6c, 7b–e, Supplementary Figs. 32, 33). GraphPad Prism v.8.0. was used for statistical analysis. $P < 0.05$ was considered statistically significant.

## Reporting summary

Further information on research design is available in the Nature Portfolio Reporting Summary linked to this article.

## Data availability

Mass spectra were searched against the human SwissProt database (20422 entries) concatenated with reverse decoy database (https://www.uniprot.org/proteomes?facets=proteome_type:1&query=(organism_id:9606); proteome ID: UP000005640; release numbers: 2021_01/2021_01). Proteomic data, including raw data and search results have been deposited in the ProteomeXchange database with dataset identifier "PXD044005". All other data supporting the findings of this study are available within the article and its supplementary files. Any additional requests for information can be directed to, and will be fulfilled by, the lead contact. Source data are provided with this paper.

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

## Acknowledgements

This work was financially sponsored by the National Natural Science Foundation of China (Nos. 52122306 (W.Z.), 52073190 (W.Z.) and U21A2098 (C.Z.)), Natural Science Foundation of Sichuan Province (2023NSFSC0337 (W.Z.)), and 1.3.5 project (ZYJC21014 (W.Z.) and ZYJC21010 (W.Z.)) for disciplines of excellence, West China Hospital Sichuan University. H.J. acknowledges the financial support of the China Scholarship Council (CSC) for the visiting study at the University of British Columbia, and the Research Trainee award from Michael Smith Foundation for Health Research. Yanping Huang at the College of Chemistry and Engineering, and Chao He at the College of Polymer Science and Engineering, Sichuan University, are thanked for their kind support for the SEM analysis. Moreover, we thank our laboratory members for their generous help. J.N.K. is a Tier 1 Canada Research Chair in immunotherapy and immunomodulation materials. We thank PTM BIO (Hangzhou, China) for providing Blood+DIA quantitative proteome analysis.

## Author contributions

T.X. and H.J. conceived the early heparin-mimetic polymer-modified sponge and designed the experiments. T.X. and H.J. prepared, characterized and investigated the anticoagulant mechanism of the sponge. L.X., S.C. and X.L. designed and conducted the animal experiments. R.Z. designed and conducted the studies for the coagulation factors. Y.L. and J.N.K. provided conceptual advice and technical support. T.X. and H.J. drafted the manuscript. W.Z., J.N.K. and C.Z. supervised the study and revised the paper. All authors gave approval to the final version of the manuscript.

## Competing interests

The authors declare no competing interests.
