## [Peer Review File · Nature Communications]

REVIEWER COMMENTS

Reviewer #1 (Remarks to the Author):

Review to Ms NCOMMS-22-37903

Tao Xu, Highly efficient anticoagulant sponges for whole blood auto-transfusion and its mechanism of coagulation factor inactivation

General

The study applies previously described heparin-mimetic polymers (HMP) on a melamine sponge to suppress blood coagulation in surgical auto-transfusion settings. The study applies a smart combination of coagulation-, chromogenic/fluorogenic, and proteomic assays for the fundamental analysis of the anticoagulant mechanism of the HMP coating and the protein corona. Further, whole blood compatibility and in vivo side effects are analyzed in a rabbit model.

Detailed

The abstract and introduction may specify the composition and describe current knowledge of the heparin-mimetic polymer in more detail.

The authors give several indications that the anticoagulant effect of the MS@D-HMP is independent of antithrombin III (ATIII). Can this also be demonstrated directly with ATIII-deficient plasma? Such independence might be of clinical interest because ATIII can be pre-term consumed in clinical settings. What about in influence of heparin cofactor II? – In the case of independence of ATIII independence, is the term “heparin-mimetic polymer” still appropriate?

MS@D-HMP-treated plasma has decreased fibrinogen concentration. For completeness of the study, it would be useful to show that this is not converted to fibrin, e.g. by an ELISA of fibrinopeptides and/or D-dimers (so far only shown in the animal experiment).

HMP, especially the loose corona, shows inhibitory and depletion properties. Are there saturation effects, and is it possible to indicate a capacity?

The authors report a significant drop of calcium ions in plasma and blood after contacting MS@D-HMP. Does this have an impact on ECG or skeletal muscles?

Minor

Line 100: The authors state HMP was engineered “with optimum ratio of sulfonic and carboxyl groups”. They should indicate, concerning which properties the HMP was optimized and how this was determined.

Line 209ff: The experimental setting should be mentioned: Were these experiments performed with plasma or purified coagulation factors? Are the results based on chromogenic tests or clotting assays?

Line 223: Antithrombin is a minor inhibitor of FXIa. – Authors should also report the related results of C1Inh (SERPING1) and alpha1-AT (SERPINA1). (W.A. Wuillemin et al., Blood 85: 1517-1526 (1995))

Line 229ff: The statement of contact activation should contain a linking phrase to the following analysis.

Line 252f: do the authors mean “...inceptive establishment of unique tightly-bound proteins...”?

Line 370/380/599: The authors assume a proteolytic inactivation of thrombin. Why is the activation of a steric inhibitor excluded? – Probably a more conservative formulation would be better.

Line 422ff: PF4 has affinity to negatively charged groups and may be trapped by MS@D-HMP, causing false-low values. The authors should determine an additional marker of platelet activation.

Line 531: The abbreviation FDP is used before introduction.

Manfred Maitz

Leibniz Institute of Polymer Research Dresden

Reviewer #2 (Remarks to the Author):

The manuscript NCOMMS-22-37903 by T. Xu, et al. with title “Highly efficient anticoagulant sponges for whole blood auto-transfusion and its mechanism of coagulation factor inactivation”, addresses the development of anticoagulant sponges based on melamine foams and coated with acrylic acid and 2-acrylamido-2-methyl-1-propanesulfonic acid as a heparin-mimetic polymer. The authors have extensively characterized the sponges prepared and investigated their anticoagulant performance as well as their in vitro biocompatibility and safety of blood salvage and reinfusion in vivo. They also investigated the influence of the coagulation factors by the coated sponge and the plasma protein layer deposited on the sponge when brought in contact with blood plasma.

In general, the manuscript is interesting and the authors have performed a very large number of experiments to support their claims and conclusions. The data are presented and discussed in sufficient detail. Moreover, potential limitations of the developed construct are discussed which is appealing.

However, my main concern is related to the material development proposed in this study. The developed coating is not new and the authors have previously reported similar microspheres, based on the same materials, for extracorporeal blood purification via the inactivation of coagulation factors (reference 25 in the present work). In this previous work, the authors carried out similar experiments on the quantification of individual coagulation factors, bound proteins on the microspheres, etc. They have also shown the anticoagulant effect of the microspheres and the treatment and recovery of a dog using the microspheres on a polypropylene apparatus. (i) Therefore, the authors need to justify the need for the development of this new coated sponge and describe the advantages (if any) of the new construct. Also describe the advantages of the new construct compared to the anticoagulant membranes reported in the literature. To the reader's eye this part of the study, seems to address only the engineering of the new sponge construct and the novelty is limited. (ii) TGA (mass loss) does not support the material deposition on the sponge surface. Could the authors perhaps calculate the amount of deposited material on the porous surface by another technique? I.e. elemental analysis? (iii) In addition, the leakage of the coating material from the sponge surface, after its long-term use, could be determined and discussed in the manuscript.

On the other hand, the authors have investigated the underlying mechanisms of the inactivation of coagulation factors by the sponge. Perhaps this is the most interesting part of the present study, and the authors should focus the manuscript on this part of their work.

Other minor comments:

- 1) The Videos are not particularly enlightening. I could not access the first video, while the second only showed some mice in their cages.
- 2) In Figure 7 it would be helpful for the reader, if the colors for the heparin and MS@D-HMP samples were kept constant for all analyses.

Reviewer #3 (Remarks to the Author):

Cell salvage systems are sometimes used during surgery to avoid allogeneic blood transfusion. Such systems typically involve the collection of shed blood into heparinized saline, filtration, and washing of the blood, followed by reinfusion. To simplify this process, Xu and colleagues have developed sponges impregnated with heparin-mimetic polymer (HMP)-modified microspheres. These sponges adsorb clotting proteins (particularly factor [F] XI) and calcium, thereby decreasing the coagulability of the blood and rendering it amenable to reinfusion.

Although the proposed technology is interesting, there are problems that need to be addressed. These can be divided into major and minor concerns.

MAJOR

1. The paper is densely written, which makes it difficult to follow.
2. The authors need to better identify how the current study advances the field beyond previous publications on HMP-modified microspheres by them (e.g., references 25, 32, and 59) and others (e.g., references 33 to 35).
3. The terminology used in this paper is confusing. The sponges appear to reduce the coagulability of blood by adsorbing clotting factors, particularly those in the intrinsic pathway of coagulation, and calcium. However, the authors use the terms “adsorption” and “inactivation” interchangeably and introduce terms such as “salvage”, “passed through” (page 7, line 179), and “refusion” without providing clear definitions.
4. Despite many in vitro experiments, the mechanism by which the sponges reduce the coagulability of the blood remains unclear. What is the capacity of the sponges to adsorb clotting factors and calcium? This question would be best addressed using purified proteins or calcium and standardized recovery assays. Such assays could be used to determine the binding capacity of the sponges and to determine whether the sponges activate or inactivate the clotting factors.
5. The observation that the sponges prolong the activated partial thromboplastin time (aPTT) and the thrombin time (TT), but not the prothrombin time (PT), is puzzling. I can see how the adsorption of clotting factors in the intrinsic pathway could prolong the aPTT, but it is unclear how this phenomenon explains the prolonged TT. Although the adsorption of fibrinogen by the sponges could result in the prolongation of the TT, it should also prolong the PT. Studies using purified proteins may help to sort this out.
6. What is the effect of soluble HMP versus HMP immobilized on the sponges on blood coagulability? Does the sponge surface induce contact activation? These questions would be best addressed using purified proteins as well as a plasma system.
7. The authors speculate that sponges impregnated with HMP-modified microspheres would obviate the need for filtration and washing of blood collected during surgery. However, such sponges could also adsorb bacteria, fat, or tumor cells. How would this potential problem be addressed and would this problem not dampen the enthusiasm for use of such sponges?

MINOR

1. The size of the sponges does not appear to be standardized across experiments nor is this information provided for all the experiments.

2. The figure legends provide inconsistent amounts of detail.
3. There is inconsistent use of statistical tests to determine the relevance of the findings.

Point-by-point response to the detailed comments by the reviewers of “*Highly efficient anticoagulant sponges for whole blood auto-transfusion and its mechanism of coagulation factor inactivation*” with manuscript ID: NCOMMS-22-37903.

Reviewer #1

“*Tao Xu, Highly efficient anticoagulant sponges for whole blood auto-transfusion and its mechanism of coagulation factor inactivation. The study applies previously described heparin-mimetic polymers (HMP) on a melamine sponge to suppress blood coagulation in surgical auto-transfusion settings. The study applies a smart combination of coagulation-, chromogenic/fluorogenic, and proteomic assays for the fundamental analysis of the anticoagulant mechanism of the HMP coating and the protein corona. Further, whole blood compatibility and in vivo side effects are analyzed in a rabbit model.*”

Response to the general comment:

Thanks for your valuable scientific comments and suggestions to improve the quality of our manuscript. The suggestions have provided us with a deeper perspective on exploring the anticoagulant mechanism of heparin-mimicking surfaces. We have carefully revised the manuscript and added all the necessary data to support our claims. Additionally, we have systematically responded to all your comments, such as the discussion of the previous reports on heparin-like polymer, the effect of sponges on antithrombin III and heparin cofactor II, detection of possible conversion of fibrinogen to D-dimer by ELISA, saturation capacity of FXI and calcium ion adsorption, and alternative evaluation of platelet activation, etc. For the detailed comments, please see below our point-to-point response. Finally, thank you very much for reviewing our manuscript, and looking forward to receive your positive response.

Detailed

Comment #1: “*The abstract and introduction may specify the composition and describe current knowledge of the heparin-mimetic polymer in more detail.*”

Response to comment: Thanks for your good comment. Heparin is the primary anticoagulant used to prevent clot formation induced by biomaterials.¹ However, its structural heterogeneity has led to several limitations such as variable bioavailability, unpredictable dose requirements, the need for frequent coagulation monitoring, and immunological adverse reactions.²⁻⁴ These limitations have prompted the development of novel heparin mimetics as alternative anticoagulants with extended clinical safety profiles that are less burdensome for patients.⁵

Early heparin mimetics were developed based on the tailored uronic backbone, inspired by the specific AT-binding pentasaccharide sequence that is responsible for heparin’s anticoagulant activity.⁶ Extensive research has been conducted to understand the anticoagulant mechanisms of these heparin mimetics, most of which were based on antithrombin III (ATIII)- or heparin cofactor II (HCII)-activating activity.⁸⁻¹³ Although some polyanions exhibited good anticoagulant effects, they were usually not named as heparin mimetics inceptively. Until the last decade, synthetic polymers or nanomaterials that do not strictly resemble heparin in structure (particularly uronic backbone) have been developed and named as heparin mimetics without unified reasons.¹⁴⁻²¹ For example, Zhang et al. defined polymeric heparin-mimetics as synthetic polymers that mimic the chemical structure of

heparin, and tailored composition of the respective functional components such as saccharides, carboxylates and sulfonates.¹⁴ Liu et al. adopted similar definitions and named the polymers with carboxylate and sulfonate groups as heparin-like polymers.¹⁵ In contrast, Chen et al. emphasized that bioactivity of heparin-mimetics should be comparable to that of heparin.¹⁶ Nahain et al. prepared a library of linear, heparin mimetic polymers and evaluated their utility as heparin mimetics based on their anticoagulant properties.^{17, 18} Others even did not give a clear definition of heparin-mimetics.¹⁹⁻²¹ We conclude that these heparin mimetics mainly demonstrate macroscopic anticoagulation results similar to heparin, such as prolonged aPTT and TT. Since the last decade, heparin mimetics have been used to prepare self-anticoagulant materials.²²⁻²⁴ However, both the molecular anticoagulant mechanism and the anticoagulant mechanism of the heparin mimetics-modified materials have not been thoroughly explored, which hinders their applications.

It is clear that the scope of heparin mimetics has been expanding, from highly similar structures with uronic backbone to representative features that mimic the functional groups or highly negative charge of heparin.^{5, 25-27} While these heparin mimetics have been demonstrated similar anticoagulant effects to heparin at a macroscopic level, the diversity in their structures has resulted in a wide range of molecular anticoagulant mechanisms, including ATIII-,^{28, 29} or HCII-activating activity,³⁰ inhibition of factor Xase activity,³¹ and interference with polymerization of fibrin monomer.^{32, 33} In addition, after the introduction of heparin mimetics onto the surface of material, the anticoagulant mechanism at interface becomes even more elusive, especially its function and the mechanism of surface-protein interaction which contributes to the anticoagulant function.³⁴⁻³⁷ Given the diversity of heparin mimetics and the differences in their anticoagulant mechanisms after being grafted onto surfaces, it is impractical to conduct a thorough investigation of each heparin mimetic modified surface. Therefore, our research only focuses on the carboxylated and sulfonated polymer modified surface due to its well-established modification method.

In summary, related anticoagulant substances are called heparin mimetics either because of their structural similarity (uronic backbone, functional groups and highly negative charge) or functional similarity (prolonged aPTT and TT). These heparin mimetics have the advantages of being easy to modify and functionalize, but neither the anticoagulant mechanisms of these molecules nor the heparin mimetics modified surfaces have been clearly explored, which greatly limits their further translation and application.

We have added related information in the Abstract on Page 2 of the revised manuscript: “Herein, we develop a **carboxylated and sulfonated** heparin-mimetic polymer (HMP) modified sponge that could spontaneously adsorb blood ($1.149 \text{ kg/m}^2 \text{ s}^{-1/2}$) along with instantaneous anticoagulation.” and in the Introduction on Page 3 of the revised manuscript: “**Heparin mimetics, with an expanding scope ranging from structures closely resembling uronic backbone to other representative features that mimic the functional groups of heparin²⁵⁻²⁸, have been extensively developed to address concerns associated with heparins.**” and on Page 3 of the revised manuscript: “Based on our findings, we hypothesized that the multiple inhibitory effects of **carboxylated and sulfonated** HMP-modified materials on the coagulation cascade could enable their use in blood auto-transfusion without additional anticoagulants, and thereby achieve the unprecedented whole blood auto-transfusion in clinical trauma-induced hemorrhage.” and on Page 3 of the revised manuscript: “**However, the introduction of heparin mimetics onto the material surface adds complexity which necessitates deeper understanding the mechanism of self-anticoagulant function at the interface.**” and in the Outlook on Page 20 of the revised manuscript: “**Since the last**

decade, heparin mimetics have been widely used to develop self-anticoagulant materials^{24, 69, 70}. However, both the molecular anticoagulant mechanism of heparin mimetics and the anticoagulant mechanism of the heparin mimetics-modified materials have not been thoroughly explored. After introducing heparin mimetics onto the surface of a material, the anticoagulant mechanism at interface becomes even more elusive, especially with respect to the side effects, and the mechanism of surface-protein interaction which contributes to its anticoagulant function. Carboxylated and sulfonated polymers have been widely adopted for heparin-mimetic modification due to its well-established methods^{24, 71, 72}. Through systematic exploration of the coagulation reactions at the molecular level, we revealed that the carboxylated and sulfonated HMP-modified sponges effectively adsorb FXI to the surface.”

Comment #2: “The authors give several indications that the anticoagulant effect of the MS@D-HMP is independent of antithrombin III (ATIII). Can this also be demonstrated directly with ATIII-deficient plasma? Such independence might be of clinical interest because ATIII can be pre-term consumed in clinical settings. What about in influence of heparin cofactor II? – In the case of independence of ATIII independence, is the term “heparin-mimetic polymer” still appropriate?”

Response to comment: Thanks for your professional suggestions. Exploration of the effect of the MS@D-HMP on ATIII and heparin cofactor II (HCII) is crucial to investigate whether the MS@D-HMP performs its anticoagulant function like heparin does and its mechanism of action. Hence, clotting times of the ATIII-deficient plasma after contacting with the MS@D-HMP were determined. The MS@D-HMP induced a significant prolongation for aPTT and TT, and inhibited the activity of FXI significantly (**Supplementary Figs. 23a-c**). In contrast, heparin lost its coagulation inhibition ability in the absence of ATIII. These data support that the MS@D-HMP may not inhibit thrombin through ATIII as heparin does.

Thrombin inhibition by HCII is possible at high heparin concentration.³⁸ Unfortunately, imports of commercial HCII-deficient plasma have been banned in China due to the impact of the COVID-19 pandemic. Hence, the effect of the MS@D-HMP on HCII was investigated using purified HCII protein instead. FIIa cleaves fibrinogen and triggers the spontaneous fibrin polymerization, which caused turbidity change (**Supplementary Fig. 23d**). Neither HCII nor heparin alone significantly inhibited fibrin polymerization induced by thrombin, whereas a significantly inhibited thrombin activity was observed with co-existence of heparin and HCII (**Supplementary Fig. 23e**). In contrast, the MS@D-HMP-treated HCII caused no significant inhibition of the thrombin activity. This result indicates that the inhibitory effects of the MS@D-HMP are on common pathway, which is not dependent on HCII as heparin does. Actually, this result may need further investigation if the following two conditions are present. One is that HCII may be adsorbed by sponge, another is that the effect of sponge on HCII relies on other protein layer adsorbed from plasma. The suggested HCII-deficient plasma could exclude the effect of these two possibilities. Unfortunately, such an experiment cannot be carried out in the near future due to uncontrollable factors, but we still hope to make more explorations in our future work.

For the rationality of the term “heparin-mimetic polymer” with ATIII-independence, as our reply to your **Comment #1**, anticoagulant substances are called heparin mimetics either because of their structural similarity (uronic backbone, functional groups and highly negative charge) or functional similarity (prolonged aPTT and TT). Hence, we believe that the use of term “heparin-mimetic polymer”

is still appropriate considering the similarity in the functional groups and prolonged aPTT and TT.

Supplementary Figure 23. Effect of ATIII and HCII on the anticoagulant behaviors of sponge. a-c aPTT values (a), FXI activity (b) and TT values (c) of ATIII-deficient plasma after incubation with MS@D-HMP. Heparin-anticoagulated ATIII-deficient plasma (1 $\mu\text{g}/100 \mu\text{L}$ ATIII-deficient plasma) was used for comparison (n = 3 biologically independent samples, mean \pm SD. One-way ANOVA with Bonferroni post-hoc tests). d Effect of MS@D-HMP on the inhibition of FIIa-mediated fibrin polymerization in the presence of HCII. Turbidity change was monitored by absorbance at 350 nm. (n = 3 biologically independent samples, mean \pm SD). e Calculated residual FIIa activity in the presence of HCII (n = 3 biologically independent samples, mean \pm SD. One-way ANOVA with Bonferroni post-hoc tests).

We updated the manuscript with related information on Page 12 of the revised manuscript: “The effect of two traditional thrombin inhibitors, such as ATIII and heparin cofactor II (HCII), on sponge-mediated anticoagulation was investigated using ATIII-deficient plasma and purified HCII protein. In the case of ATIII-deficient plasma, MS@D-HMP induced a significant prolongation for aPTT and TT, and inhibited the activity of FXI significantly (Supplementary Figs. 23a-c). However, heparin lost its anticoagulant ability in ATIII-deficient plasma. Regarding HCII-mediated thrombin-initiated fibrin formation, the MS@D-HMP-treated HCII did not cause any significant inhibition of thrombin activity, while thrombin was significantly inhibited with the co-existence of heparin and HCII (Supplementary Figs. 23d-e). These results indicated that the inhibition of thrombin by MS@D-HMP may not rely on

ATIII binding or HCII similar to heparin.”

The experimental details about the effect of ATIII and HCII on sponge-mediated anticoagulation have been added on Page 10 of the revised supplementary information: “To investigate the effect of the MS@D-HMP on ATIII, 10 mg MS@D-HMP was incubated with 100 μ L ATIII-deficient plasma (BioMedica Diagnostics Inc.) for 30 min. The incubated plasma was collected to determine the aPTT, TT and FXI activity. For comparison, heparin was added into the ATIII-deficient plasma at a concentration of 0.1 IU/100 μ L PPP (aPTT > 600 s, TT > 240 s for heparin-anticoagulated normal PPP at this concentration). The FXI activities were determined based on activated partial thromboplastin time (aPTT). Briefly, 5 μ L incubated ATIII-deficient plasma was added in a test cup and diluted with 45 μ L buffer solution. After incubating at 37 $^{\circ}$ C for 30 s, the complex solution was mixed with 50 μ L corresponding factor deficient plasma (Coagulation Factor XI deficient plasma; Sysmex; incubated 10 min at 37 $^{\circ}$ C before use). After another 30-second incubation at 37 $^{\circ}$ C, 50 μ L aPTT reagent was added, followed with adding 50 μ L CaCl₂ solution (25 mM) after incubating at 37 $^{\circ}$ C for 3 min, and then the aPTT value was measured in an automatic coagulation analyzer (ACL Elite Pro, werfan). Commercial normal plasma (Sysmex) was used to determine standard curve through the instrument's built-in gradient dilution procedure. Finally, the FXI factor activity was calculated based on the standard curve.

To investigate the effect of the MS@D-HMP on HCII, 6 mg MS@D-HMP was incubated with 60 μ L HCII/PBS solution (0.34 μ M) for 10 min, and 50 μ L incubated solution was collected for HCII+MS@D-HMP group. For comparison, 50 μ L PBS solution was incubated for 10 min for blank group. 50 μ L HCII/PBS solution (0.34 μ M) was incubated for 10 min for HCII group. 50 μ L heparin/PBS solution (0.2 IU/ μ L) was incubated for 10 min for heparin group. 50 μ L HCII/heparin/PBS solution (0.34 μ M for HCII, 0.2 IU/ μ L for heparin) was incubated for 10 min for HCII+heparin group. These incubated solutions were mixed with 20 μ L FIIa (3 IU/mL), and incubated for another 5 min. Finally, fibrinogen solution was added to obtain a final concentration at 2.5 mg/mL. The turbidity change was measured at 350 nm for 30 min in a plate reader (Thermo Scientific, USA). The residual FIIa activity was calculated based on the standard curve using gradient-diluted FIIa solution.”

Comment #3: “MS@D-HMP-treated plasma has decreased fibrinogen concentration. For completeness of the study, it would be useful to show that this is not converted to fibrin, e.g. by an ELISA of fibrinopeptides and/or D-dimers (so far only shown in the animal experiment).”

Response to comment: Thanks for your thoughtful comments. We fully agree with you about the probability that the decreased fibrinogen might be converted into fibrin. Hence, the potential conversion of fibrinogen to fibrin in the MS@D-HMP-incubated plasma is considered in the following aspects.

Firstly, fibrin formation is initiated by thrombin-mediated cleavage of fibrinopeptides A and B from the respective A α and B β chains of fibrinogen. As indicated in the concentration of thrombin-antithrombin complex, there was no significant generation of thrombin in the MS@D-HMP-incubated plasma. So, the decreased fibrinogen may not be converted to fibrin by the generated thrombin.

Secondly, the formed fibrin would improve the elastic properties after being cross-linked, and thrombelastography can detect the change in the viscoelasticity of the blood. The thrombelastography showed that there was no change in the viscoelasticity in the blood after contacting with MS@D-HMP,

which meant fibrin was not formed macroscopically.

Thirdly, micro-fibrin (micro-thrombi) may generate in the blood after contacting with MS@D-HMP. Such fibrin can be broken down into soluble parts (fibrin degradation products) in the presence of plasmin, which thus causes depletion of fibrinogen. Plasminogen, a circulating plasma zymogen, can be converted to plasmin by tissue plasminogen activator (tPA), urokinase-type plasminogen activator (uPA) and proteases in the intrinsic coagulation cascade including kallikrein, factor XIa and factor XIIa.^{39, 40} Of note, MS@D-HMP initiated activation of contact system, which may accelerate the plasmin activation and induce the degradation of potentially generated micro-fibrin (micro-thrombi). Hence, it is important to explore the possible generation of micro-fibrin (micro-thrombi) and fibrinolysis in the plasma after incubation with MS@D-HMP. As suggested, we detected the concentration of D-dimer in the plasma via ELISA method (Human D-Dimer ELISA Kit, Cusabio, CSB-E05175h). Compared with normal citrate-anticoagulated plasma, an increased concentration of D-dimer was observed in the re-calcified plasma after 6-hour incubation (**Fig. 5h**). In contrast, no significant increase in the concentration of D-dimer was observed for the re-calcified plasma treated by MS@D-HMP after 6-hour incubation, which meant that micro-thrombi did not form and no fibrin was degraded in the incubated plasma. Given that the ELISA test measures the concentration of D-dimer in the liquid phase, it is crucial to acknowledge that the test may not detect the degradation of micro-thrombi on the sponge surface. But our main focus still lies in the state of blood after contacting with sponge. In addition, related questions had been considered in animal experiments (**Supplementary Figure 29**), and no blood clot was observed on the sponge surface after contacting with blood. Overall, the decreased fibrinogen in the plasma after incubation with MS@D-HMP was not converted to fibrin.

Fig. 5 h Concentration of D-dimer in the citrate-anticoagulated plasma after incubation with sponge. Plasma was re-calcified (final Ca^{2+} concentration at 10 mM), and then incubated with MS@D-HMP. Plasma without recalcification were used as normal control. For positive control, re-calcified plasma was incubated for 1 h, and fibrinolysis was initiated by addition of human plasmin (80 $\mu\text{g}/\text{mL}$) ($n = 3-4$ biologically independent samples, mean \pm SD. One-way ANOVA with Bonferroni post-hoc tests).

The related discussions about the potential degradation of fibrinogen have been added on Page 14 of revised manuscript: “Given the activation of contact system, the possible generation of micro-thrombi and subsequent fibrinolysis in the plasma after incubation with MS@D-HMP was investigated based on the D-dimer level. Compared with normal plasma, no significant increase was observed for the D-dimer level in the re-calcified plasma treated by MS@D-HMP after 6-hour incubation, which meant that micro-thrombi did not form and no fibrin was degraded (Fig. 5h).” and on Page 12 of revised manuscript: “The loss of fibrinogen was observed in the MS@D-HMP-incubated plasma (Supplementary Figs. 21a-b). However, the decreased fibrinogen was not converted to fibrin (as discussed in Fig. 5h).”

The experimental details about detection of D-dimer have been added on Page 12 of the revised supplementary information: “To rule out the possibility of generation of micro-fibrin (micro-thrombi) and fibrinolysis in the plasma after incubation with MS@D-HMP, and then converted into D-dimer, the concentration of D-dimer in the plasma was detected via ELISA method (Human D-dimer ELISA Kit, Cusabio, CSB-E05175h). Citrate-anticoagulated PPP (100 μ L) was re-calcified with stock solution of calcium ions (final Ca^{2+} concentration at 10 mM), and then incubated with 10 mg MS@D-HMP. After incubation at 37 $^{\circ}$ C for 30 min, the plasma was collected (sponge was removed for MS@D-HMP-treated group) and incubated for another 6 h. The plasma without recalcification was used as normal control. For positive control, re-calcified plasma was incubated for 1 h, and fibrinolysis was initiated by addition of human plasmin (80 μ g/mL, Sigma-Aldrich) with 6-hour incubation. Finally, the detection of D-dimer was conducted according to the respective instruction manuals. At least 4 parallel sample groups were applied to get a reliable value, and the results were expressed as mean \pm SD.”

Comment #4: “HMP, especially the loose corona, shows inhibitory and depletion properties. Are there saturation effects, and is it possible to indicate a capacity?”

Response to comment: Thanks for your insightful comments. The saturation effects and capacity would be meaningful when estimating the sponge required during use.

For depletion of FXI, we only discussed the equilibrium binding capacity since the sponge cannot reach the saturated binding capacity under practical conditions. The equilibrium FXI binding capacity and FXI inactivation efficiency by the MS@D-HMP were determined using FXI-deficient plasma after adding different amounts of purified FXI protein. 6 mg sponge was exactly immersed in 100 μ L plasma for 30 min to ensure adequate contact between sponge and plasma. Interestingly, FXI equilibrium binding capacity by the sponge reached 80.6 ng/mg with physiological level of FXI (5 μ g/mL), and it almost increased proportionally with the initial FXI concentration. FXI inactivation efficiency reached \sim 98 % with physiological FXI level (**Fig. 3e**). High FXI inactivation efficiency (85.8 %) remained even when the initial FXI concentration increased to 80 μ g/mL (16 times the physiological level). Such FXI binding and inactivation efficiency that the sponge achieved at ideally high initial FXI concentration should have promised a robust theoretical anticoagulant ability that was overqualified for our practical application. Although saturated binding capacity might be obtained with further increasing initial FXI concentration and operation time, such ideal conditions are not available in practical use. Hence, the saturation FXI binding capacity may sometimes not be direct indicative of the practical anticoagulant ability, which necessitates the preliminary experiments to ensure a high factor of safety.

Fig. 3 e Equilibrium binding capacity and the corresponding inactivation efficiency of FXI by MS@D-HMP. FXI-deficient plasma (100 µL) supplemented with different amounts of FXI proteins was incubated with MS@D-HMP (6 mg) for 30 min (n = 3 biologically independent samples, mean ± SD).

For the chelation of calcium ions, we initially investigated the time-dependent calcium ions adsorption behavior by MS@D-HMP in a solution of calcium ions (2 mM, physiological level). The MS@D-HMP showed a rapid calcium ions adsorption kinetics with an equilibrium adsorption capacity at 19.6 µg/mg (**Supplementary Fig. 14a**). Then, the efficacy of calcium ions adsorption was confirmed in plasma environment. Hirudin-anticoagulated plasma with different initial calcium ions concentrations was obtained by addition of stock solution of calcium ions. The resulted plasma was incubated with MS@D-HMP for 10 min, and the adsorption amount was determined. Although high depletion efficiency (85.1 %) was achieved in the plasma environment with an initial calcium ions concentration of 2 mM, the equilibrium adsorption capacity was only 0.60 µg/mg, which was much lower than that in the calcium ions solution (**Supplementary Fig. 14b** and **Supplementary Table 6**). The decreased equilibrium calcium ions adsorption capacity in plasma might be resulted from the competitive effect of other cations in the plasma and shorter incubation time. Hence, we recommend to refer the equilibrium binding capacity of calcium ions for the MS@D-HMP measured in plasma conditions.

Supplementary Figure 14. Calcium ions adsorption behaviors of MS@D-HMP in the solution of calcium ions and plasma. a Calcium ions adsorption amount of MS@D-HMP in calcium ions solution (2 mM) at different incubation time (n = 3 independent samples, mean ± SD). **b** Equilibrium binding capacity and the corresponding depletion efficiency of calcium ions by MS@D-HMP in

hirudin-anticoagulated plasma environment (n = 3 biologically independent samples, mean ± SD).

For inhibition of thrombin, inactivation was independent of adsorption, but relied on an inhibitor generated in plasma after incubation with MS@D-HMP. Identifying such inhibitor is a crucial step to achieve the precise quantification of the capacity, but can hardly be realized without appropriate high-throughput and *in-situ* technical means. Actually, efforts have been made to exclude the probability of ATIII and HCII in our reply to your **Comment #2**. Understanding the mechanism of thrombin inhibition induced by HMP-modified surface is of interest in our future work.

Overall, the equilibrium binding capacity of FXI and calcium ions under different initial adsorbate concentrations has been determined, and compared with the corresponding actual capacity when used in physiological level. We found that it is still difficult to estimate the appropriate amount of sponge under practical conditions on the basis of equilibrium capacity alone. Patients' blood may vary with disease states, which causes different pressures on the anticoagulant behavior of the sponge. For example, recovering blood with introduced tissue factor and prematurely activated thrombin relies on the ability to inhibit thrombin and chelate calcium ions, while activation of intrinsic pathway via contact system during storage needs to be blocked by depletion of FXI and chelation of calcium ions. Considering the inconsistency between the equilibrium binding capacity and practical anticoagulant performance, it is recommended to re-consider the amount of sponge used in combination with the actual capacity pre-determined from the applied conditions.

The related discussions about calcium ions binding capacity have been added on Page 6 of the revised manuscript: “In a solution of calcium ions (2 mM, physiological concentration), MS@D-HMP showed a rapid calcium ions adsorption kinetics with an equilibrium adsorption capacity at 19.6 µg/mg (Supplementary Fig. 14a). Calcium ions depletion efficiency remained high (85.1 %) in the plasma environment, although a decreased equilibrium adsorption capacity (0.60 µg/mg) was observed due to the competitive effect of other cations and shorter incubation time (Supplementary Fig. 14b and Supplementary Table 6).”

The related discussions about FXI binding capacity have been added on Page 8 of the revised manuscript: “Strong sponge-FXI interaction was presented beforehand through equilibrium FXI binding capacity and FXI inactivation efficiency in plasma with different initial FXI concentrations (Fig. 3e). Of note, the FXI inactivation efficiency reached ~98 % with physiological FXI concentration (5 µg/mL), thus ensuring a high factor of safety. The equilibrium FXI binding capacity by sponge reached 80.6 ng/mg with a physiological FXI concentration, and almost increased proportionally to the initial FXI concentration. High FXI inactivation efficiency (85.8 %) remained even when the initial FXI concentration increased to 80 µg/mL (16 times the physiological concentration), which should promise a robust anticoagulant ability when used for blood auto-transfusion.”

The experimental details about determining calcium ions binding capacity have been added on Page 5-6 of the revised supplementary information: “The calcium ions adsorption kinetics by MS@D-HMP was determined in calcium ions solution. 20 mL calcium ions solution (2 mM) was incubated with 20 mg MS@D-HMP at 37 °C for a certain time (5, 15, 30, 60, 240, 480, 1440 min). The residual concentrations of calcium ions were determined by an atomic absorption spectrometer (Shimadzu SPCA-626D, Japan), and then the adsorption amounts were calculated.

The calcium ions adsorption by MS@D-HMP was also investigated in plasma. Certain amounts of calcium ions were added into 400 µL hirudin anticoagulant plasma. The re-calcified plasma was

incubated with 10 mg MS@D-HMP for 10 min. The calcium concentration in the treated plasma was determined by a biochemical analysis instrument (Cabas C311, Roche, Switzerland). Then the adsorption amounts were calculated. This is also a preliminary experiment for the posterior fluorescence-based FIXa activity assay as FXIa activates FIX to FIXa in a calcium-dependent way.”

The experimental details about determining FXI binding capacity have been added on Page 8 of the revised supplementary information: “The equilibrium binding capacity and the corresponding inactivation efficiency of FXI by MS@D-HMP were determined using the purified FXI protein. A certain amount of FXI native protein was mixed with 100 μ L FXI-deficient plasma to obtain the final FXI concentrations of 2.5, 5, 10, 20, 40 and 80 μ g/mL. The FXI-supplemented plasma was incubated at 37 $^{\circ}$ C for 30 min, and then incubated with 6 mg MS@D-HMP for another 30 min at 37 $^{\circ}$ C. The residual FXI concentration of the incubated plasma was determined based on a standard curve for FXI concentration versus FXI activity. Gradient-diluted FXI-supplemented plasma was prepared, and the resulted activity was detected using an automatic coagulation analyzer (ACL Elite Pro, werfan) based on similar method as described in **Experimental Section 5.1**. The equilibrium binding capacity and the corresponding inactivation efficiency can be calculated as follows:

$$\text{Equilibrium binding capacity} = \frac{V(C_0 - C_e)}{m} \quad (2)$$

$$\text{Inactivation efficiency} = \frac{C_0 - C_e}{C_0} \times 100 \% \quad (3)$$

where C_0 and C_f are the concentrations of FXI in plasma before and after incubation with MS@D-HMP. V is the volume of the plasma, m is the weight of MS@D-HMP used.”

Comment #5: “The authors report a significant drop of calcium ions in plasma and blood after contacting MS@D-HMP. Does this have an impact on ECG or skeletal muscles?”

Response to comment: Thanks for your good comment. We fully agree with your concerns about the possible effect of drop of calcium ions in plasma on ECG or skeletal muscles after reinfusion. In our study, the blood transfusion volume in the rabbit femoral artery hemorrhage model was set to be 10 % of the total blood volume of rabbits (approximately 15 mL for rabbits weighing about 3-4 kg). In this case, the use of blood salvage is beneficial to reduce the likelihood of severe postoperative anemia as red blood cells take weeks to completely replace.⁴¹ When collecting blood using MS@D-HMP *in vitro*, a drop in the concentration of calcium ions in the blood was observed (**Fig. 6f**). The reinfusion of such blood caused no hypocalcemia *in vivo* because 10 % of reinfused blood would decrease the concentration of calcium ions *in vivo* by 8 %, which was still within the normal range (**Fig. 7I, Supplementary Table 13**).

A more severe blood loss should also be taken into account. In this case, the reinfusion of blood collected by the MS@D-HMP can cause a further decrease in the calcium ions with increased reinfusion volume. This may have an impact on ECG or skeletal muscles, which necessitates the monitoring of calcium ions and calcium supplementation. In fact, severe hypocalcemia is always associated with traditional massive transfusion protocols, because the citrate anticoagulant contained in stored allogeneic blood can bind calcium ions in the recipients.⁴² Despite rapid hepatic metabolism of citrate, massive transfusion may result in a large influx of citrate, amplified by an impairment in citrate metabolism due to hypothermia, hypoperfusion and liver dysfunction.⁴³ On the other hand, for current auto-transfusion technology with normal saline as the recommended wash solution, acid-base

and electrolyte abnormalities including the decrease in calcium ions are also observed in the clinical setting.⁴⁴ Hence, continuous monitoring of calcium ions and calcium supplementation is needed for all these technologies with increased transfusion volume. Compared with these technologies, recovery and reinfusion of blood using MS@D-HMP still has substantial advantages to avoid metabolic alkalosis caused by accumulation of citrate during transfusion of allogeneic blood. In addition, the collected blood could be stored and processed *in vitro*, and provides a valuable window of time for subsequent clinical operations. Overall, caution is necessary when using MS@D-HMP with increased reinfusion volume, otherwise the obtained hypocalcemia would have a negative impact on ECG and skeletal muscles.

The related discussions and associated limitation have been added on Page 21 of the revised manuscript: “However, the reinfusion of increased volume of sponge-collected blood *in vivo* may induce hypocalcemia, which necessitates monitoring of calcium ions and calcium supplementation, as traditional massive transfusion protocols suggest⁷⁴.”

Minor

Comment #6: “Line 100: The authors state HMP was engineered “with optimum ratio of sulfonic and carboxyl groups”. They should indicate, concerning which properties the HMP was optimized and how this was determined.”

Response to comment: Thanks for your comment and we apologize for missing details. The optimum ratio of sulfonic and carboxyl groups in the heparin-mimetic polymer (HMP) was based on our previous study⁴⁵. Briefly, a series of HMP-based surfaces were prepared using monomers (acrylic acid (AA; providing carboxyl groups) and 2-acrylamido-2-methyl-1-propanesulfonic acid (AMPS; providing sulfonic groups)) with different feed mass ratios of AA to AMPS by free radical polymerization. The prepared surfaces were named as A_xM_y, where “x:y” means the feed mass ratios of AA and AMPS. Plasma recalcification time (PRT) tests were used to evaluate the anticoagulant performance of the surfaces in real plasma environment (**Fig. R1**). Among all the surfaces, A₃M₂ exhibited the best anticoagulant performance. Only 5 mg materials can significantly prolong the PRT of 300 μL PPP. Hence, a feed mass ratio of 3:2 was considered as the optimum ratio of AA to AMPS in the HMP-based surfaces.

Fig. R1 Digital photos of previous PRT tests to determine the optimum ratio of AA and AMPS in the HMP-based surfaces. HMP-based surfaces were prepared with different feed mass ratios of AA and AMPS. When the concentration of surfaces decreased from 15 mg to 5 mg per 300 μ L PPP, only PPP incubated with A₃M₂ remained flow state after recalcification.⁴⁵ Hence, a feed mass ratio of 3:2 was considered as the optimum ratio.

The related details have been added on Page 3-4 of the revised manuscript: “HMP was covalently decorated on the substrate through in-situ polymerization of monomers acrylic acid (AA) and 2-acrylamido-2-methyl-1-propanesulfonic acid (AMPS), with the feed mass ratio of 3:2 (AA:AMPS) based on plasma recalcification time (PRT) in previous work²⁹.” and on Page 3 of the revised supplementary information: “The optimum ratio of sulfonic and carboxyl groups for the heparin-mimetic polymer was based on our previous study². Briefly, a series of HMP-based surfaces were prepared using monomers (AA, providing carboxyl group; AMPS, providing sulfonic groups) with different feed mass ratios of AA to AMPS by free radical polymerization. The prepared surfaces were named as A_xM_y, where “x:y” means the feed mass ratios of AA and AMPS. Plasma recalcification time tests were used to evaluate the anticoagulant performance of the surfaces in real plasma environment. Among all the surfaces, A₃M₂ exhibited the best anticoagulant performance. Hence, a feed mass ratio of 3:2 was considered as the optimum ratio of AA to AMPS in the HMP-based surfaces.”

Comment #7: “Line 209ff: The experimental setting should be mentioned: Were these experiments performed with plasma or purified coagulation factors? Are the results based on chromogenic tests or clotting assays?”

Response to comment: Thanks for your thoughtful comments and we apologize for the missing details. The effect of the sponges on intrinsic coagulation factors (FVIII, FIX, FXI and FXII) was determined in citrate-anticoagulated normal plasma (**Fig. 3a**), or factor-deficient plasma (**Fig. 4g**). The activities

of corresponding coagulation factors were determined based on activated partial thromboplastin time (aPTT). Briefly, 5 μ L incubated plasma was added in a test cup and diluted with 45 μ L buffer solution. After incubation at 37 °C for 30 s, the complex solution was mixed with 50 μ L corresponding factor deficient plasma (Coagulation Factor VIII deficient plasma, Coagulation Factor IX deficient plasma, Coagulation Factor XI deficient plasma and Coagulation Factor XII deficient plasma; Sysmex; incubated 10 min at 37 °C before use). After another 30-second incubation at 37 °C, 50 μ L aPTT reagent was added, followed with adding 50 μ L CaCl₂ solution (25 mM) after incubation at 37 °C for 3 min, and then the aPTT value was measured in an automatic coagulation analyzer (ACL Elite Pro, werfan). Commercial normal plasma (Sysmex) was used to determine standard curve through the instrument's built-in gradient dilution procedure. Finally, the corresponding factor activity was calculated based on the standard curve.

The related experimental details have been added in the figure caption on Page 10 of the revised manuscript: “Activities of FVIII, FIX, FXI and FXII for the normal citrate-anticoagulated PPP after incubation with MS@D-HMP at different concentrations for 30 min. The activities were determined based on aPTT by mixing incubated plasma with corresponding factor deficient plasma.” and in the figure caption on Page 14 of the revised manuscript: “Activities of FVIII, FIX and FXI for the FXII-deficient plasma after incubation with MS@D-HMP for 30 min.” and on Page 6 of the revised supplementary information: “The effect of the sponges on the activities of intrinsic coagulation factors (FVIII, FIX, FXI and FXII) was determined in the citrate-anticoagulated PPP incubated with the sponges. Certain amounts of sponges were pre-immersed in PBS overnight and then incubated at 37 °C for 1 h. After that, the PBS was thoroughly removed, and 300 μ L fresh PPP was introduced and incubated at 37 °C for 30 min. The activities of corresponding coagulation factors were determined based on activated partial thromboplastin time (aPTT). Briefly, 5 μ L incubated plasma was added in a test cup and diluted with 45 μ L buffer solution. After incubating at 37 °C for 30 s, the complex solution was mixed with 50 μ L corresponding factor deficient plasma (Coagulation Factor VIII deficient plasma, Coagulation Factor IX deficient plasma, Coagulation Factor XI deficient plasma and Coagulation Factor XII deficient plasma; Sysmex; incubated 10 min at 37 °C before use). After another 30-second incubation at 37 °C, 50 μ L aPTT reagent was added, followed with adding 50 μ L CaCl₂ solution (25 mM) after incubating at 37 °C for 3 min, and then the aPTT value was measured in an automatic coagulation analyzer (ACL Elite Pro, werfan). Commercial normal plasma (Sysmex) was used to determine standard curve through the instrument's built-in gradient dilution procedure. Finally, the corresponding factor activity was calculated based on the standard curve.”

Comment #8: “Line 223: Antithrombin is a minor inhibitor of FXIa. – Authors should also report the related results of C1Inh (SERPING1) and alpha1-AT (SERPINA1). (W.A. Willemin et al., Blood 85: 1517-1526 (1995))”

Response to comment: Thanks for your professional comment. We carefully read the suggested reference and fully agree with you about the crucial role of these serine protease inhibitors including C1 inhibitor (C1INH), antithrombin and alpha-1-antitrypsin (a1-AT) in the inhibition of FXIa.⁴⁶ It is possible that these serine protease inhibitors contributed to the decreased FXI activity. In this case, FXI was significantly activated to FXIa in the plasma after incubation with MS@D-HMP, and then the generated FXIa was inhibited by these serine protease inhibitors. There is also another possibility that the decreased FXI activity was derived from the depletion after FXI being adsorbed by MS@D-HMP. Which of these two possibilities is dominant can be distinguished via proteomic analysis of

plasma after incubation with MS@D-HMP. If FXIa was inhibited by serine protease inhibitors in the plasma, these FXIa could still be digested to peptides, and detected in the proteomic analysis. This meant that there would not be significant difference in the abundance of FXI in the plasma pre- and post-incubation with sponge. In contrast, if FXI was adsorbed on the surface of sponge from the plasma, these FXI could no longer be detected in plasma in the proteomic analysis. This meant that decreased abundance of FXI in the plasma after incubation with sponge would be observed. The final result showed the decreased relative percentage of FXI in the plasma (**Fig. 3c**), which confirmed that the adsorption of FXI by the sponge played a decisive role in the decreased activity.

Even so, we believe that these serine protease inhibitors should theoretically contribute to the decreased FXI activity. Reasons are as follows. Partial FXI would be activated to FXIa by sponge. These FXIa had procoagulant potential to activate FIX to FIXa, whereas this process was blocked due to the chelation of calcium ions. It is highly possible that these FXIa would be bound and inhibited by the mentioned serine protease inhibitors. However, the precise contribution of these serine protease inhibitors on the decreased FXI activity is hard to be calculated. We confirmed that the sponge did not activate FXI directly (**Fig. 4f**), but could activate FXI via the contact system (**Fig. 4b-e**). Proteins in the contact system, such as FXIIa and kallikrein, may also be inhibited by these serine protease inhibitors.^{47, 48} Hence, it is difficult to control a single variable during exploration, without which precise contribution of these serine protease inhibitors can hardly be obtained.

The related discussions have been added on Page 8 of the revised manuscript: “Serine protease inhibitors, such as ATIII, alpha-1-antitrypsin, and plasma C1 inhibitor, remained relatively unchanged (Supplementary Table 8). These are inhibitors of FXIa and may contribute to the decreased FXI activity^{38,39}. FXIa inhibited by these inhibitors could theoretically be digested to peptides and detected by the proteomic analysis, resulting in unobvious intensity change of FXI. However, the relative percentage of FXI dramatically decreased by 70.6 % in MS@D-HMP-incubated plasma (Fig. 3c), which meant that the adsorption of FXI by sponge played a decisive role in the decreased FXI activity.” and on Page 11 of the revised manuscript: “the generated FXIa was unstable and might be quickly inhibited by serine protease inhibitors (ATIII, alpha-1-antitrypsin, and plasma C1 inhibitor, etc.) in the plasma^{38,39}.” and on Page 15 of the revised manuscript: “The products of contact activation, such as FXIIa and FXIa, may also be inhibited by serine protease inhibitors, ensuring the safety of recovered blood after treatment.”

Comment #9: “Line 229ff: The statement of contact activation should contain a linking phrase to the following analysis.”

Response to comment: Thanks for your good comment. The statement in Line 229ff was an indirect evidence of contact activation based on the results of mass-spectrometry-based proteomics. We fully agree with you about the necessity of a linking phrase as the detailed exploration was discussed in the next part of manuscript.

In addition, after confirming the activation of contact system by sponge (**Figs. 4c-e**), a linking phrase is also need for the following analysis of hemocompatibility (**Fig. 5**). The activation of contact system may initiate the subsequent thrombo-inflammatory reactions (activation of complement system and blood cells), for which we have investigated in the following section but did not discuss in connection with the results of contact activation. The ELISA method showed that, compared with control blood, no significant increase in the concentration of C3a and C5a was observed in the blood after incubation

with MS@D-HMP (**Figs. 5i-j**). Actually, this result did not take into account the false reduction due to the adsorption of C3a and C5a by sponge if complement system was activated after contact activation. But from the perspective of safety, the main focus still lies in the state of blood after contacting with sponge. No significant activation of complement system and blood cells were observed in the blood after treatment (**Figs. 5e-g**). Hence, even if contact activation initiated the complement system, the resulted products may not produce obvious side effects due to the adsorption of sponge. The products of contact activation, such as FXIIa and FXIa, may be adsorbed and inhibited by serine protease inhibitors, and the connection between contact activation and complement activation can be inhibited by chelation of calcium and magnesium ions by the MS@D-HMP, thus ensuring the safety of blood after treatment.

The related linking phrases have been added on Page 8 of the revised manuscript: “Assembly of these proteins on the surface might induce a probable activation of contact system with potential chance of clotting⁴⁰, which would be discussed in the later section.” and on Page 15 of the revised manuscript: “Even if contact activation partially triggered complement activation, the resulted products may not produce obvious side effects due to adsorption of sponge. Furthermore, calcium or magnesium ions are essential for activation of the classical, lectin, or alternative pathways⁵¹. The crosstalk between contact activation and complement activation may be inhibited by chelation of calcium and magnesium ions. The products of contact activation, such as FXIIa and FXIa, may also be inhibited by serine protease inhibitors, ensuring the safety of recovered blood after treatment.”

Comment #10: “Line 252f: do the authors mean “...inceptive establishment of unique tightly-bound proteins...”?”

Response to comment: Thanks for your professional comment, and we apologize for the inappropriate explanation. The sentence in Line 252f used to mean that the inactivation of intrinsic factors might be achieved during the inceptive establishment of unique loosely-bound protein layer on the surface of MS@D-HMP. Actually, we should acknowledge that this may be a hasty speculation needing more rigorous investigation. We have review the related results carefully, and believe that it’s more appropriate to adopt a more conservative statement that the adsorption of FXI by the surface of sponge caused the inactivation of FXI in the plasma, and such function declined in the presence of the loosely-bound protein layer on the surface. Reasons are as follows:

Firstly, as our reply to your **Comment #4**, the sponge showed robust FXI equilibrium binding capacity, which indicated strong sponge-FXI interaction (**Fig. 3e**). The FXI equilibrium binding capacity almost increased proportionally to the initial FXI concentration in plasma. High inactivation efficiency (85.8 %) was achieved even when the initial FXI concentration at 80 µg/mL (16 times the physiological level), which should promise a robust anticoagulant ability when used for blood auto-transfusion.

However, in the two-stage successive incubations coupled with clotting time tests (**Fig. 3d**), anticoagulant function declined only in the 2nd incubation. Briefly, in the 1st incubation of stage 1, pristine sponge was incubated with fresh plasma for the 1st time. The incubated plasma was separated from the sponge, and clotting time tests were performed. Sponge significantly prolonged the aPTT value of incubated plasma to 154.0 s (**Fig. 3f**). Then, in the 2nd incubation of stage 1, the sponge after adsorbing plasma proteins was allowed to incubate with fresh plasma for the 2nd time. The incubated plasma was also separated from the sponge, and clotting time tests were performed. However, the

sponge after adsorbing plasma proteins exerted weaker anticoagulant ability, and the aPTT value of incubated plasma was only prolonged to 85.8 s. After that, the loosely-bound proteins on the surface of sponge were removed by PBS washing. In the 1st incubation of stage 2, the washed sponge was incubated with fresh plasma again. The incubated plasma was separated from the sponge, and clotting time tests were performed. In this case, the anticoagulant ability of the sponge was restored, and the aPTT value of incubated plasma was significantly prolonged to 136.8 s. This meant that the presence of loosely-bound protein layer on the surface played a negative role when sponge exerted anticoagulant function.

To further illustrate the anticoagulant behavior of the sponge without loosely-bound protein layer, the plasma with aPTT values significantly prolonged during the above-mentioned procedures was used for proteomic analysis. In the 1st incubation of stage 1, pristine sponge was incubated with fresh plasma for the 1st time. The incubated plasma was collected, and proteomic analysis was performed. In the 1st incubation of stage 2, the washed sponge was incubated with fresh plasma again. The incubated plasma was collected, and proteomic analysis was performed. As shown in **Fig. 3g**, compared with fresh plasma, similar decreased relative percentage of FXI (88.5 % versus 86.5 %) and protein intensity (89.4 % versus 87.0 %) was observed in these two aliquots of plasma with comparable aPTT values (154.0 s versus 136.8 s). This meant that the sponge exerted similar robust anticoagulant function (adsorption of FXI) after the disassociation of loosely bound protein layer by PBS washing.

Finally, the tightly-bound proteins on the sponge were used for proteomic analysis. Sponges after 2nd incubation of stage 1 and 2 were washed by PBS. Proteins were *in-situ* digested into peptides, which were loaded with equal volume. As show in **Fig. 3h**, compared with stage 1, relative percentage of tightly-bound FXI increased by 55.7 % in stage 2 (protein intensity increased by 110.3 %). Note that the weight of adsorbed proteins may differ between the two stages, protein intensity may provide a better description of increased proteins than relative percentage based on total proteins (protein intensity divided by the sum of all identified protein intensities). This meant that FXI was effectively adsorbed into the tightly-bound protein layer when the loosely-bound protein layer had been removed. However, reaching a conclusion that tightly-bound FXI results in the inactivation of FXI in the plasma requires the premise that the amount of decreased FXI in plasma corresponds to the amount of tightly-bound FXI on the surface. Precise quantification of the FXI by mass spectrometry still needs more rigorous validation, e.g., exact correspondence between abundance and mole fraction of protein (particularly at low mole fractions).⁴⁹ Mass quantification by ELISA is also limited by antibody specificity considering the possible activation and conformational change of FXI on the surface. Even though, this is still a very interesting phenomenon, and we will make more explorations in our future work.

Hence, we believe that it's more appropriate to use the statement that inactivation of FXI by sponge might decline along with the development of the loosely-bound protein layer on the surface.

Fig. 3 d Schematic of two-stage successive incubations combined with PBS treatment. In stage 1, tightly- and loosely-bound proteins were adsorbed on the sponge in 1st incubation with fresh plasma. Such sponge without washing was allowed for 2nd incubation with another fresh plasma. Then, the sponge undergoing stage 1 was washed by PBS to remove the loosely-bound proteins, and used in stage 2 with similar procedures as those of stage 1. **e** Equilibrium binding capacity and the corresponding inactivation efficiency of FXI by MS@D-HMP. FXI-deficient plasma (100 μ L) supplemented with different amounts of FXI proteins was incubated with MS@D-HMP (6 mg) for 30 min ($n = 3$ biologically independent samples, mean \pm SD). **f** aPTT values for the PPP after successive incubations with MS@D-HMP in stages 1 and 2 ($n = 8$ biologically independent samples, mean \pm SD. Two-way ANOVA with Bonferroni post-hoc tests). **g** Abundance of coagulation-related proteins in fresh plasma, plasma after 1st incubation with MS@D-HMP in stages 1, and plasma after 1st incubation with MS@D-HMP in stages 2. Values are expressed in relative percentages based on total proteins. **h** Abundance of coagulation-related proteins in the tightly-bound protein layer on the surface of MS@D-HMP in stages 1 and 2. Values are expressed in relative percentages based on total proteins.

The related discussions about the effect of surface-bound proteins on anticoagulant behavior of sponges have been reorganized on Page 8-9 of the revised manuscript: “To understand the correlation between the establishment of the protein layer and the anticoagulant behavior of MS@D-HMP, the effect of tightly- and loosely-bound proteins on the inactivation of intrinsic factors was investigated using two-stage successive incubations combined with PBS washing (Fig. 3d). Strong sponge-FXI interaction was presented beforehand through equilibrium FXI binding capacity and FXI inactivation efficiency in plasma with different initial FXI concentrations (Fig. 3e). Of note, the FXI inactivation efficiency reached ~98 % with physiological FXI concentration (5 μ g/mL), thus ensuring a high factor of safety. The equilibrium FXI binding capacity by the sponge reached 80.6 ng/mg with a physiological FXI concentration, and almost increased proportionally to the initial FXI concentration. High FXI inactivation efficiency (85.8 %) remained even when the initial FXI concentration increased to 80 μ g/mL (16 times the physiological concentration), which should promise a robust anticoagulant ability when used for blood auto-transfusion. Interestingly, in the two-stage successive incubations combined

with PBS washing (Fig. 3d), anticoagulant function declined only in the 2nd incubation. In the 1st incubation of stage 1, fresh sponge significantly prolonged the aPTT (154.0 s), while the sponge after adsorbing plasma proteins exerted weaker anticoagulant ability (aPTT 85.8 s) in the 2nd incubation of stage 1 (Fig. 3f). After the removal of loosely-bound proteins on the surface, the anticoagulant function of the sponge was restored in the 1st incubation of stage 2 (aPTT 136.8 s). This suggested that the developed loosely-bound protein layer on the surface might involve as the barrier for the sponge-FXI interaction.

To illustrate the anticoagulant behavior of sponge without loosely-bound protein layer, two aliquots of plasma with comparable aPTT values (154.0 s versus 136.8 s) after the 1st incubation of stages 1 and 2 were collected for proteomics analysis. Compared to fresh plasma, a similar decrease in relative percentages of FXI (88.5 % versus 86.5 %) was observed in these two aliquots of plasma (Fig. 3g). This suggested that the sponge exerted similar robust anticoagulant function (sponge-FXI interaction) with the loosely-bound protein layer removed from the surface. Snapshots of tightly-bound proteins on the sponge were obtained after PBS washing. SDS-PAGE analysis showed similar bands for both stages (Supplementary Fig. 17). Proteomics analysis provided information about common proteins and fold changes in relative abundance of different proteins (Supplementary Figs. 18 a-b). The identified proteins were bioinformatically grouped according to their biological functions⁴² (Supplementary Figs. 18 c-g, Supplementary Table 9). Compared with stage 1, relative percentage of tightly-bound FXI increased by 55.7 % in stage 2 (protein intensity increased by 110.3 %). Taken together, FXI can be effectively adsorbed into the tightly-bound protein layer, which might be suppressed after the building of the loosely-bound protein layer.”

The experimental details have been added on Page 8 of the revised supplementary information: “In stage 1, 30 mg MS@D-HMP was firstly incubated with 300 μ L PPP for 30 min, and the PPP was collected for aPTT tests and LC-MS/MS analysis.”

Comment #11: “Line 370/380/599: The authors assume a proteolytic inactivation of thrombin. Why is the activation of a steric inhibitor excluded? – Probably a more conservative formulation would be better.”

Response to comment: Thanks for your thoughtful comments and we apologize for the inappropriate statement. The possibility for the activation of a steric inhibitor in the MS@D-HMP-incubated PPP cannot be excluded. In fact, the regulation of coagulation cascade involves many steric inhibitors, including antithrombin, heparin cofactor II, thrombomodulin, tissue factor pathway inhibitor, etc.⁵⁰ We can only speculate that unknown inhibitor except ATIII caused the inactivation of thrombin.

The related discussions have been corrected on Page 12 of the revised manuscript: “TT values could not be corrected when the mixing concentration reached 0.1 mL/mL normal PPP due to the unknown thrombin inhibitor (protease inhibitors, steric inhibitor, etc) activated by the MS@D-HMP.” and on Page 12 of the revised manuscript: “An unknown inhibitor activated by the MS@D-HMP inhibited thrombin” and on Page 21 of the revised manuscript: “An unknown thrombin inhibitor might be activated by sponge, even in the collected plasma when the sponge was removed”.

Comment #12: “Line 422ff: PF4 has affinity to negatively charged groups and may be trapped by MS@D-HMP, causing false-low values. The authors should determine an additional marker of platelet activation.”

Response to comment: Thanks for your good comments. PF4 is a multimeric protein with a molecular weight of 29,000 Daltons and strong cationic charge and binds with high affinity to unfractionated heparin.⁵¹ The resulted heparin/PF4 complex can be recognized by antibodies to activate platelets via FcγIIA, which causes immune heparin-induced thrombocytopenia during heparin therapy.⁵² In addition, PF4 also binds other anionic glycosaminoglycans with lower affinity, including heparan sulfates and chondroitin sulfates.⁵³ Hence, the decreased concentration of PF4 in the blood after contacting with MS@D-HMP was resulted from the adsorption of PF4 by the negatively charged surface. To evaluate the real situation of platelet activation induced by MS@D-HMP, we quantified the level of platelet activation by flow cytometry. Anti-CD41a antibody reacting with platelet GPIIb was used for identification of platelets, while anti-CD62p antibody which targets at P-selectin was utilized for evaluation of activated platelets. Compared with negative control, there was no significant difference for the expression of CD62p in the PRP treated with MS@D-HMP. Hence, despite the adsorption of PF4, MS@D-HMP did not induce significant platelet activation.

Fig. 5 f Effect of sponges on platelet activation in PRP, measured on the basis of expression of platelet activation marker CD62p. Thrombin receptor activating peptide (TRAP, 0.1 mM) and control PRP were used as positive and negative control, respectively (n = 5 biologically independent samples, mean ± SD. One-way ANOVA with Bonferroni post-hoc tests).

The related discussions have been added on Page 14 of the revised manuscript: “Compared to pristine blood, the generated PF4 in the MS@D-HMP-incubated blood significantly decreased due to the adsorption of PF4 with a strong cationic charge to the negatively charged surface⁵⁰ (Fig. 5e). The real level of platelet activation was quantified by flow cytometry based on the expression of platelet activation marker CD62p. Compared with negative control, there was no significant difference in the expression of CD62p in the PRP treated with MS@D-HMP (Fig. 5f), confirming that MS@D-HMP did not induce significant platelet activation.”

The experimental details have been added on Page 11 of the revised supplementary information: “The level of platelet activation was also evaluated by flow cytometry. 200 μL citrate-anticoagulated PRP was incubated at 37 °C with 15 mg MS@D-HMP in 1.5 mL polypropylene (PP) tube. PRP without treatment and PRP treated with thrombin receptor activating peptide (TRAP, Sigma-Aldrich, 0.1 mM) were used as negative and positive control, respectively. After 30 min, aliquots of the incubated PRP were collected for assessment of the platelet activation. For the preparation of staining solution, 5 μL anti-CD41a-FITC (eBioscience™, #11-0419-42) and 1.25 μL anti-CD62p-APC (eBioscience™, #17-

0626-82) were added in 90 μ L PPP. Then, 5 μ L samples were diluted in 90 μ L staining solution. After incubation in the dark for 15 min, the samples were further stopped with 300 μ L flow cytometry staining buffer (eBioscience™). The level of platelet activation was analyzed in a flow cytometer (BD FACSCelesta™) by gating platelets specific events based on anti-CD41a-FITC. The mean fluorescence intensities of platelet activation marker CD62p-APC signal of platelet population were normalized to the intensity of TRAP-activated platelets. Data analysis used FlowJo 7.2 software (TreeStar Inc).”

Comment #13: “Line 531: The abbreviation FDP is used before introduction.”

Response to comment: Thanks for your good comments, and we apologize for the mistake. The related problems have been corrected on Page 18 of the revised manuscript: “Coagulation activation and secondary fibrinolysis were investigated by detecting the fibrin degradation products (FDP) and D-dimer levels *in vivo*”.

References

1. Ronco C, *et al.* Renal replacement therapy in acute kidney injury: controversy and consensus. *Crit. Care* **19**, (2015).
2. Kalathottukaren MT, Haynes CA, Kizhakkedathu JN. Approaches to prevent bleeding associated with anticoagulants: current status and recent developments. *Drug Deliv. Transl. Res.* **8**, 928-944 (2018).
3. Blossom DB, *et al.* Outbreak of adverse reactions associated with contaminated heparin. *N. Engl. J. Med.* **359**, 2674-2684 (2008).
4. Manson L, Weitz JI, Podor TJ, Hirsh J, Young E. The variable anticoagulant response to unfractionated heparin *in vivo* reflects binding to plasma proteins rather than clearance. *J. lab. clin. med.* **130**, 649-655 (1997).
5. Paluck SJ, Nguyen TH, Maynard HD. Heparin-mimicking polymers: synthesis and biological applications. *Biomacromolecules* **17**, 3417-3440 (2016).
6. Choay J, Petitou M, Lormeau JC, Sinay P, Casu B, Gatti G. Structure-activity relationship in heparin - a synthetic pentasaccharide with high-affinity for anti-thrombin-III and eliciting high anti-factor-Xa activity. *Biochem. Biophys. Res. Commun.* **116**, 492-499 (1983).
7. Vanboeckel CAA, Petitou M. The unique antithrombin-III binding domain of heparin - a lead to new synthetic antithrombotics. *Angew. Chem. Int. Ed.* **32**, 1671-1690 (1993).
8. Petitou M, *et al.* Synthesis of thrombin-inhibiting heparin mimetics without side effects. *Nature* **398**, 417-422 (1999).
9. Grootenhuis PDJ, Westerduin P, Meuleman D, Petitou M, Vanboeckel CAA. Rational design of synthetic heparin analogs with tailor-made coagulation-factor inhibitory activity. *Nat. Struct. Biol.* **2**, 736-739 (1995).
10. Herbert JM, *et al.* SR123781A, a synthetic heparin mimetic. *Thromb. Haemost.* **85**, 852-860 (2001).
11. Avci FY, Karst NA, Linhardt RJ. Synthetic oligosaccharides as heparin-mimetics displaying anticoagulant properties. *Curr. Pharm. Des.* **9**, 2323-2335 (2003).
12. Demir M, *et al.* Anticoagulant and antiprotease profiles of a novel natural heparinomimetic mannopentaose phosphate sulfate (PI-88). *Clin. Appl. Thromb./Hemost.* **7**, 131-140 (2001).
13. Hayakawa Y, Hayashi T, Hayashi T, Niiya K, Sakuragawa N. Selective activation of heparin-cofactor-II by a sulfated polysaccharide isolated from the leaves of artemisia-princeps. *Blood Coagul. Fibrinolysis* **6**, 643-649 (1995).
14. Zhang A, *et al.* The role of carboxylic groups in heparin-mimicking polymer-functionalized surfaces for blood compatibility: enhanced vascular cell selectivity. *Colloids Surf. B.* **201**, 111653 (2021).

15. Liu Y, *et al.* Heparin reduced dialysis through a facile anti-coagulant coating on flat and hollow fiber membranes. *J. Membr. Sci.* **595**, 117593 (2020).
16. Chen X, *et al.* Sulfonate groups and saccharides as essential structural elements in heparin-mimicking polymers used as surface modifiers: optimization of relative contents for antithrombogenic properties. *ACS Appl. Mater. Interfaces* **10**, 1440-1449 (2018).
17. Al Nahain A, Ignjatovic V, Monagle P, Tsanaktsidis J, Vamvounis G, Ferro V. Anticoagulant heparin mimetics via raft polymerization. *Biomacromolecules* **21**, 1009-1021 (2020).
18. Al Nahain A, Ignjatovic V, Monagle P, Tsanaktsidis J, Vamvounis G, Ferro V. Sulfonated RAFT copolymers as heparin mimetics: synthesis, reactivity ratios, and anticoagulant activity. *Macromol. Biosci.* **20**, 2000110 (2020).
19. Wang H, *et al.* Artificial extracellular matrix composed of heparin-mimicking polymers for efficient anticoagulation and promotion of endothelial cell proliferation. *ACS Appl. Mater. Interfaces* **14**, 50142-50151 (2022).
20. Liu Y, *et al.* Anticoagulation polyvinyl chloride extracorporeal circulation catheters for heparin-free treatment. *J. Mater. Chem. B* **10**, 8302-8314 (2022).
21. Niu Y, *et al.* An aptasensor based on heparin-mimicking hyperbranched polyester with anti-biofouling interface for sensitive thrombin detection. *Biosens. Bioelectron.* **101**, 174-180 (2018).
22. Nie C, *et al.* Novel heparin-mimicking polymer brush grafted carbon nanotube/PES composite membranes for safe and efficient blood purification. *J. Membr. Sci.* **475**, 455-468 (2015).
23. Ji H-f, Xiong L, Shi Z-q, He M, Zhao W-f, Zhao C-s. Engineering of hemocompatible and antifouling polyethersulfone membranes by blending with heparin-mimicking microgels. *Biomater. Sci.* **5**, 1112-1121 (2017).
24. Yang F, Guo G, Wang Y. Inflammation-triggered dual release of nitroxide radical and growth factor from heparin mimicking hydrogel-tissue composite as cardiovascular implants for anti-coagulation, endothelialization, anti-inflammation, and anti-calcification. *Biomaterials* **289**, 121761 (2022).
25. Monien BH, Cheang KI, Desai UR. Mechanism of poly(acrylic acid) acceleration of antithrombin inhibition of thrombin: implications for the design of novel heparin mimics. *J. Med. Chem.* **48**, 5360-5368 (2005).
26. Al Nahain A, Ignjatovic V, Monagle P, Tsanaktsidis J, Ferro V. Heparin mimetics with anticoagulant activity. *Med. Res. Rev.* **38**, 1582-1613 (2018).
27. Coombe DR, Kett WC. Heparin mimetics. *Handb. Exp. Pharmacol.*, 361-383 (2012).
28. Ito Y, Liu L, Imanishi Y. Interaction of poly(sodium vinyl sulfonate) and its surface graft with antithrombin-III. *J. Biomed. Mater. Res.* **25**, 99-115 (1991).
29. Jeske W, Fareed J. Antithrombin III-mediated and heparin-cofactor II-mediated anticoagulant and antiprotease actions of heparin and its synthetic analogs. *Semin. Thromb. Hemost.* **19**, 241-247 (1993).
30. Onishi M, Miyashita Y, Motomura T, Yamashita S, Sakamoto N, Akashi M. Anticoagulant and antiprotease activities of a heparinoid sulfated glucoside-bearing polymer. *J. Biomater. Sci. Polym. Ed.* **9**, 973-984 (1998).
31. Sugidachi A, Asai F, Koike H. Anticoagulant and antiprotease activities of aprosulate sodium, a new synthetic polyanion, in human plasma and purified systems. *Blood Coagul. Fibrinolysis* **5**, 773-779 (1994).
32. Tamada Y, Murata M, Hayashi T, Goto K. Anticoagulant mechanism of sulfonated polyisoprenes. *Biomaterials* **23**, 1375-1382 (2002).
33. Sano M, Tamada Y, Niwa K, Morita T, Yoshino G. Sulfated sericin is a novel anticoagulant influencing the blood coagulation cascade. *J. Biomater. Sci. Polym. Ed.* **20**, 773-783 (2009).
34. Charef S, TaponBretaudiere J, Fischer AM, Pfluger F, Jozefowicz M, Labarre D. Heparin-like functionalized polymer surfaces: discrimination between catalytic and adsorption processes during the course of thrombin inhibition. *Biomaterials* **17**, 903-912 (1996).
35. Fougnot C, Jozefowicz M, Rosenberg RD. Catalysis of the generation of thrombin-antithrombin complex by

- insoluble anticoagulant polystyrene derivatives. *Biomaterials* **5**, 94-99 (1984).
36. Belattar N. Affinity adsorption of human vitamin K-dependent coagulation factor IX onto heparin-like poly (styrene sodium sulfonate) adsorbent. *Mater. Sci. Eng. C* **27**, 849-854 (2007).
 37. Fougnot C, Jozefowicz M, Bara L, Samama M. Interactions of anticoagulant insoluble modified polystyrene resins with plasmatic proteins. *Thromb. Res.* **28**, 37-46 (1982).
 38. Tollefsen DM, Majerus DW, Blank MK. Heparin cofactor-II - purification and properties of a heparin-dependent inhibitor of thrombin in human-plasma. *J. Biol. Chem.* **257**, 2162-2169 (1982).
 39. Cesarman-Maus G, Hajjar KA. Molecular mechanisms of fibrinolysis. *Br. J. Haematol.* **129**, 307-321 (2005).
 40. Nielsen VG, Steenwyk BL, Gurley WQ. Contact activation prolongs clot lysis time in human plasma: role of thrombin-activatable fibrinolysis inhibitor and Factor XIII. *J. Heart Lung Transplant.* **25**, 1247-1252 (2006).
 41. Klein AA, *et al.* Association of anaesthetists guidelines: cell salvage for peri-operative blood conservation 2018. *Anaesthesia* **73**, 1141-1150 (2018).
 42. Byerly S, *et al.* Transfusion-related hypocalcemia after trauma. *J. Am. Coll. Surg.* **223**, E210-E210 (2016).
 43. Perkins JG, Cap AP, Weiss BM, Reid TJ, Bolan CE. Massive transfusion and nonsurgical hemostatic agents. *Crit. Care Med.* **36**, S325-S339 (2008).
 44. Halpern NA, Alicea M, Seabrook B, Spungen AM, McElhinney AJ, Greenstein RJ. Cell saver autologous transfusion: metabolic consequences of washing blood with normal saline. *J. Trauma* **41**, 407-415 (1996).
 45. Song X, *et al.* Transient blood thinning during extracorporeal blood purification via the inactivation of coagulation factors by hydrogel microspheres. *Nat. Biomed. Eng.* **5**, 1143-1156 (2021).
 46. Wuillemin WA, *et al.* Inactivation of factor XIa in human plasma assessed by measuring factor XIa-protease inhibitor complexes - major role for C1-inhibitor. *Blood* **85**, 1517-1526 (1995).
 47. Deagostini A, Lijnen HR, Pixley RA, Colman RW, Schapira M. Inactivation of factor-XII active fragment in normal plasma - predominant role of C1bar-inhibitor. *J. Clin. Invest.* **73**, 1542-1549 (1984).
 48. Harpel PC, Lewin MF, Kaplan AP. Distribution of plasma kallikrein between C1bar inactivator and alpha-2-macroglobulin in plasma utilizing a new assay for alpha-2-macroglobulin-kallikrein complexes. *J. Biol. Chem.* **260**, 4257-4263 (1985).
 49. Shin J-B, *et al.* Molecular architecture of the chick vestibular hair bundle. *Nat. Neurosci.* **16**, 365-374 (2013).
 50. Esmon CT. The interactions between inflammation and coagulation. *Br. J. Haematol.* **131**, 417-430 (2005).
 51. Greinacher A, Alban S, Dummel V, Franz G, Muellereckhardt C. Characterization of the structural requirements for a carbohydrate-based anticoagulant with a reduced risk of inducing the immunological type of heparin-associated thrombocytopenia. *Thromb. Haemost.* **74**, 886-892 (1995).
 52. Reilly MP, *et al.* Heparin-induced thrombocytopenia/thrombosis in a transgenic mouse model requires human platelet factor 4 and platelet activation through Fc gamma RIIA. *Blood* **98**, 2442-2447 (2001).
 53. Zucker MB, Katz IR. Platelet factor-IV - production, structure, and physiological and immunological action. *Proc. Soc. Exp. Biol. Med* **198**, 693-702 (1991).

Reviewer #2 (Remarks to the Author):

“The manuscript NCOMMS-22-37903 by T. Xu, et al. with title “Highly efficient anticoagulant sponges for whole blood auto-transfusion and its mechanism of coagulation factor inactivation”, addresses the development of anticoagulant sponges based on melamine foams and coated with acrylic acid and 2-acrylamido-2-methyl-1-propanesulfonic acid as a heparin-mimetic polymer. The authors have extensively characterized the sponges prepared and investigated their anticoagulant performance as well as their in vitro biocompatibility and safety of blood salvage and reinfusion in vivo. They also investigated the influence of the coagulation factors by the coated sponge and the plasma protein layer deposited on the sponge when brought in contact with blood plasma. In general, the manuscript is interesting and the authors have performed a very large number of experiments to support their claims and conclusions. The data are presented and discussed in sufficient detail. Moreover, potential limitations of the developed construct are discussed which is appealing.”

Response to the general comment:

Thanks for your valuable scientific comments and suggestions to improve the quality of our manuscript. Meanwhile, we also want to thank you for your high appreciation of our work and we are highly encouraged by your positive comments. Based on your good suggestions, the manuscript has been carefully corrected and all the necessary data have been added to support our claims. We also understand and agree with your concerns on the unclear issues, which we think that these comments are highly important for us to further improve the quality of this manuscript. We have systematically responded to all your questions, such as advantages of the sponge construct, the elemental analysis of sponge, re-organization and prominence of the underlying anticoagulant mechanisms, videos, and the inconstant colors, etc. For the detailed comments, please see below our point-to-point response. Finally, thank you very much for reviewing our manuscript, and looking forward to receiving your positive responses.

Comment #1: *“However, my main concern is related to the material development proposed in this study. The developed coating is not new and the authors have previously reported similar microspheres, based on the same materials, for extracorporeal blood purification via the inactivation of coagulation factors (reference 25 in the present work). In this previous work, the authors carried out similar experiments on the quantification of individual coagulation factors, bound proteins on the microspheres, etc. They have also shown the anticoagulant effect of the microspheres and the treatment and recovery of a dog using the microspheres on a polypropylene apparatus.*

(1) *Therefore, the authors need to justify the need for the development of this new coated sponge and describe the advantages (if any) of the new construct. Also describe the advantages of the new construct compared to the anticoagulant membranes reported in the literature. To the reader’s eye this part of the study, seems to address only the engineering of the new sponge construct and the novelty is limited.”*

Response to comment: Thanks for your good comments. We believe that our work provides substantial insights in the exploration of anticoagulant mechanisms and the development of a new application using the self-anticoagulant concept. Unlike the previous report, in this work we have shown that the anticoagulant concept needs to be further developed to whole blood auto-transfusion and keep the whole blood without clotting, and should have transfusion quality. We believe that our

contributions in these two areas (material development for a new application, and mechanism of heparin-mimics on a surface) can help to drive further development of heparin mimetics.

(I) The exploration of anticoagulant mechanisms of heparin-mimetic polymer (HMP)-modified surface

The limitations of heparin have led to the development of novel heparin mimetics as alternative anticoagulants.¹ Early heparin mimetics were developed with a tailored uronic backbone, inspired by the specific AT-binding pentasaccharide sequence responsible for heparin's anticoagulant activity.^{2, 3} Extensive research has been conducted to understand the anticoagulant mechanisms of these heparin mimetics, most of which were based on antithrombin III (ATIII)- or heparin cofactor II (HCII)-activating activity.⁴⁻⁹ However, as the scope of heparin mimetics has expanded to include representative features that mimic the functional groups or highly negative charge of heparin, the development of new heparin mimetics has outpaced in-depth research into their anticoagulant mechanisms.^{1, 10-12} While these heparin mimetics have exerted similar anticoagulant effects to heparin at a macroscopic level (such as prolonged aPTT and TT), the diversity in their structures has resulted in a wide range of molecular anticoagulant mechanisms, including ATIII-,^{13, 14} HCII-activating activity,¹⁵ inhibition of factor Xase activity,¹⁶ and interference with polymerization of fibrin monomer.^{17, 18} Moreover, after introducing heparin mimetics onto the surface of the material, the anticoagulant mechanism at the interface becomes even more elusive, especially with regards to the potential bleeding side effects, and the mechanism of surface-protein interaction which is important for the surface to function as a self-anticoagulant material.¹⁹⁻²²

We have already contributed to exploring the underlying anticoagulant mechanisms of HMP-modified surface in our previous study such as HMP-modified microspheres.²³ However, these contributions were limited to the preliminary determination of the binding of intrinsic coagulation factors, leaving a substantial gap in our understanding of the detailed mechanisms. Specifically, we have yet to fully comprehend how the material mediates the binding and inactivation of specific coagulation factors (e.g., significantly inhibited FXI), the relationship between coagulation factors binding (e.g., contact system-related FXII and FXI) and surface-induced contact activation, etc. To this end, we seek to delve deeper into this area by developing HMP-modified sponge. We recognize the importance of exploring material-protein interactions as a means to analyze these processes thoroughly. This perspective will provide valuable insights into how the material interacts with proteins, enabling us to examine the binding and inactivation of coagulation factors mediated by the material itself. Furthermore, the formation of a protein layer on the material surface plays a crucial role in bridging the gap between the material and the blood system. This protein layer serves as a vital link, facilitating a systematic examination of the material's influence on the blood system. Through this approach, we can gain a deeper understanding of how the material impacts the overall functionality of the blood system and derive pertinent usage precautions.

Finally, our mechanistic exploration demonstrates the multiple mechanisms of MS@D-HMP to reduce clotting: (i) inhibition of intrinsic pathway mainly through the adsorption of FXI. The loosely-bound protein layer suppresses the anticoagulant sponge-FXI interaction. (ii) chelation of calcium ions. (iii) inhibition of common pathway through inhibition of thrombin. Inhibition of thrombin might not rely on ATIII and HCII like heparin does. These provide sufficient theoretical support for the efficacy for MS@D-HMP in short-term anticoagulation. It ensures the effective anticoagulation of the sponge

against blood with disease states. Recovering blood with introduced tissue factor and prematurely activated thrombin relies on the ability to inhibit thrombin and chelate calcium ions, while activation of intrinsic pathway via contact system during storage needs to be blocked by depletion of FXI and chelation of calcium ions. In addition, the safety of sponge was confirmed with negligible effects on blood cells and the complement system. Based on the well-established mechanisms, the potential failure risks of this surface are also predicted and confirmed, such as the contact activation, and the possibility of surface-induced coagulation activation after recovery of coagulation factors and calcium ions *in vivo*. We believe our work sheds new light on a long-standing question on what governs hemocompatibility, elucidating that the unique protein layer acts as a bridge connecting the bio-interface with the blood system. The proposed research paradigm can facilitate the in-depth investigation of complex interactions between other HMP-based materials and the blood system. Therefore, our material design and mechanistic exploration hold the potential to stimulate further research and advance the progress of this field.

(ii) The development of a new application based on a new material to recover blood and auto-transfusion.

Our experience with HMP-modified microspheres as solid anticoagulants in extracorporeal blood circuits in our previous work²³ was important initial research into this new application. Although some understanding on the anticoagulation mechanism is explored, a deeper study is warranted for further exploration of different applications in HMP research. Our published self-anticoagulant microspheres will not be suitable for blood salvage and re-transfusion application. So, we have to build the self-anticoagulant idea further and develop a sponge that can absorb whole blood rapidly without activation and preserve quality of the blood for reinfusion. Thus, we embarked on the work described here. The development of a novel application provides a steady stream of impetus for the further development and translation of the HMP-modified materials. This is exactly the reason that we have conceived another important application for the HMP-modified surface in whole blood auto-transfusion. We believe that the systematic validation of HMP-modified sponge in this clinically unmet need with huge market prospects can also increase research community's enthusiasm for such self-anticoagulant materials.

The use of anticoagulant sponge for auto-transfusion has substantial advantages compared with traditional technology. In a highly clean and sterile condition, various blood cells and plasma proteins could be collected by sponge maintaining their normal state, and the reinfusion of blood facilitates the faster recovery of coagulation function *in vivo*. In unexpected situations, such as car accidents, battlefield scenarios, etc. The portable blood salvage might be achieved by using the sponge to provide a valuable window of opportunity for subsequent clinical operations. This is a breakthrough which has never been realized, as there is no suitable and portable device to recover blood cells directly from the patient's wound outside hospital setting, highlighting the distinction between sponge and traditional auto-transfusion technology.

In conclusion, developing this novel application scenario not only addresses a clinically urgent need with significant translational potential but also fuels further development and translation of HMP-modified materials. The systematic validation of HMP-modified sponge in the field of whole blood auto-transfusion has substantial benefits and paves the way for exciting advancements in this area.

All these points have been discussed in our revised manuscript accordingly. Especially, to emphasize

the novelty and need of our work more strongly, we have added related discussion in the Abstract on Page 2 of the revised manuscript: “Our work not only develops a safe and convenient approach for whole blood auto-transfusion, but also provides the mechanism of action of self-anticoagulant HMP-modified surfaces.” and in the Outlook on Page 20 of the revised manuscript: “The portability and easy handling ability of the sponge also open up the possibility of multi-scenario applications (such as battleground, car accidents, etc.). In these unexpected situations, only anticoagulant sponge can recover blood cells directly from the patient’s wound outside hospital setting, highlighting the distinction between sponge and traditional auto-transfusion technology.” and in the Outlook on Page 20-21 of the revised manuscript: “However, both the molecular anticoagulant mechanism of heparin mimetics and the anticoagulant mechanism of the heparin mimetics-modified materials have not been thoroughly explored. After introducing heparin mimetics onto the surface of a material, the anticoagulant mechanism at interface becomes even more elusive, especially with respect to the side effects, and the mechanism of surface-protein interaction which contributes to its anticoagulant function. Carboxylated and sulfonated polymers have been widely adopted for heparin-mimetic modification due to its well-established methods^{24, 71, 72}. Through systematic exploration of the coagulation reactions at the molecular level, we revealed that the carboxylated and sulfonated HMP-modified sponges effectively adsorb FXI to the surface. In addition, calcium ions were depleted from the plasma. An unknown thrombin inhibitor might be activated by sponge, even in the collected plasma when the sponge was removed.”

(2) “TGA (mass loss) does not support the material deposition on the sponge surface. Could the authors perhaps calculate the amount of deposited material on the porous surface by another technique? I.e. elemental analysis?”

Response to comment: Thanks for your good comments, and we apologize for the unclear explanation. Elemental analysis has been performed to determine the amount of deposited material on the surface of the sponge. As shown in **Supplementary Table 3**, the amount of deposited PDA calculated from N/C and N/H were 1.39 % and 1.06 %, respectively. The amount of deposited HMP calculated from N/C and N/H were 1.73 % and 2.86 %, respectively.

Supplementary Table 3. Elemental analysis of sponges before and after modification. PDA and HMP were used to calculate the compositions of MS@D-HMP. The deposited PDA calculated from N/C and N/H were 1.39 % and 1.06 %, respectively. The deposited HMP calculated from N/C and N/H were 1.73 % and 2.86 %, respectively.

Sample	N (%)	C (%)	H (%)
PDA	5.98	55.15	5.28
HMP	2.74	42.13	7.25
MS	39.54	31.38	4.93
MS@D	41.8	33.60	5.26
MS@D-HMP	39.5	32.28	5.11

The related information has been added on Page 4 of the revised manuscript: “The percentages of the deposited PDA and HMP on the surface were 1.39 % and 1.73 % with respect to the weight of sponge, respectively, as determined by the nitrogen-to-carbon ratio of elemental analysis (EA; Supplementary Table 3).”

The experimental details have been added on Page 22 of the revised manuscript: “SEM, EDX, XPS,

EA, mercury intrusion porosimeter, TGA, and FTIR were performed to characterize the sponges.” and Page 4 of the revised supplementary information: “To investigate the compositions of the sponges, MS, MS@D and MS@D-HMP were completely freeze-dried, and then ground into powders. Then elemental analysis (EA) of the powders was obtained through an Elemental Analyzer (Euro EA 3000).”

The details about the preparation of heparin-mimetic polymers and polydopamine have been added on Page 4 of the revised supplementary information: “For the synthesis of soluble heparin-mimetic polymers (HMP), 12 g AA, 8 g AMPS, 0.1 g APS and 80 g deionized water were mixed to obtain homogeneous solution. The obtained reaction solution was heated to 70 °C for 10 h to initiate the free radical polymerization. After that, the pH of the resulted HMP solution was adjusted to 7 with NaOH solution under magnetic stirring. Finally, the solution was dialyzed against deionized water for 48 h to obtain the HMP, and lyophilized for further use. For the synthesis of polydopamine (PDA), DOPAM (5 mg/mL) and DOPA (1 mg/mL) were dissolved in Tris-buffer solution (pH=8.5, 50 mM), and kept for 12 h with oscillation at room temperature. The PDA particles were separated by centrifugation (20000 rpm) and washing multiple times with deionized water.”

(3) *“In addition, the leakage of the coating material from the sponge surface, after its long-term use, could be determined and discussed in the manuscript.”*

Response to comment: Thanks for your insightful comment. We fully agree with your concern about the leakage of anticoagulant coating from the sponge surface. This can not only be a potential risk in its long-term application, but also pose misleading to the evaluation of anticoagulant performance and the exploration of underlying anticoagulant mechanisms. Based on the previous work, the PDA deposited coatings leaked significantly during initial 3 days of incubation in PBS buffer with an obvious decrease in the thickness.²⁴ After that, the thickness remained relatively stable for 1 week, 2 weeks and 3 weeks. Hence, we mentioned that the MS@D-HMP was immersed in deionized water for 5 days after deprotonation, and then kept in PBS until use to remove the unreacted molecules and unstable coatings thoroughly. The PBS was refreshed 3 times a day. To investigate the leakage of the coating material from the sponge surface, the as-prepared MS@D-HMP was incubated with PBS buffer for different time intervals (1 day, 3 days, and 7 days). UV-vis spectra were utilized to detect the leakage in the eluate after incubation. PDA and HMP were chosen as the control samples, as they are the representative components of anticoagulant coating. The characteristic absorption peaks for PDA and HMP solution were both around 200 nm (**Supplementary Fig. 22a,b**). No obvious absorption peak was observed for the eluate after incubation for 1 day, 3 days and 7 days (**Supplementary Fig. 22c**). In addition, the eluate was mixed with PPP, and the clotting times including aPTT, PT and TT were determined. Compared with the control PPP, the addition of eluate had negligible effect on the aPTT, PT and TT values (**Supplementary Fig. 22d-f**). Hence, the leakage of the anticoagulant coating and its effect on coagulation system are negligible after the preliminary washing treatment.

Supplementary Figure 22. Leakage of coating from sponge and its effect on clotting time. a-c UV-vis spectra for gradient-diluted polydopamine (a), HMP (b), and the eluate after incubation for 1 day, 3 days and 7 days (c). d-f Clotting times including aPTT (d), PT (e), and TT (f) of PPP mixed with the eluate after incubation for 1 day, 3 days and 7 days (n = 3 biologically independent samples, mean \pm SD. One-way ANOVA with Bonferroni post-hoc tests).

Despite the negligible effects of the leakage of the anticoagulant coating from surface, the sponge was also not designed for long-term use. In fact, most of blood-contacting materials are disposable due to protein adsorption, surface thrombosis, functional decline, and bacterial contamination after exposure to the environment. For MS@D-HMP, the inactivation of FXI via adsorption to the surface has saturation effect due to the suppression of loosely-bound protein layer, so does the chelation of calcium ions. Anticoagulation can fail during long-term use without surface cleaning, which may cause reduced blood quality. In addition, exposure of bacteria and lipopolysaccharide during long-term use may increase the risk of sepsis in patients. Hence, sponge is meant to be disposable and for single use only.

The related information has been added on Page 12 of the revised manuscript: “The leaching of anticoagulant coating and its effect on clotting times were negligible (Supplementary Fig. 22).”

The experimental details have been added on Page 10 of the revised supplementary information: “To investigate the leakage of the coating material from the sponge surface, 50 mg MS@D-HMP was incubated with 10 mL PBS buffer for different time intervals (1 day, 3 days, and 7 days). The eluate after incubation was collected. UV-vis spectra (UV-1750, Shimadzu) were recorded at the wavelengths from 190 to 400 nm using a 1 cm path length quartz cell. PDA (0.0625-1 mg/mL) and HMP (0.0625-1 mg/mL) were used as comparisons. Then, 20 μ L eluate was mixed with 200 μ L PPP, and clotting times were tested.”

Comment #2: “On the other hand, the authors have investigated the underlying mechanisms of the inactivation of coagulation factors by the sponge. Perhaps this is the most interesting part of the present study, and the authors should focus the manuscript on this part of their work.”

Response to comment: Thanks for your thoughtful comments, and we totally agree with your opinion. The exploration of mechanisms is indeed the core of this work, and we believe that investigating mechanisms can inspire researchers in related fields. Unfortunately, in the original document, this part may have been written too densely, which resulted in the significance not being effectively conveyed. Hence, we have logically divided this part into four sections and given it more emphasis in terms of length, while also providing additional arguments to make the conclusions more completed and credible. For example, we addressed issues such as the equilibrium adsorption capacity of FXI to the surface, and the effect of antithrombin III and heparin cofactor II on inactivation of thrombin by sponge, etc.

Given that this article is an interdisciplinary work, we have also reserved some content on materials science and practical applications to make the information accessible to readers from different backgrounds. In fact, the third reviewer paid attention to the application part of this article. We understand that the exploration of anticoagulation mechanisms may not be friendly to many material experts and doctors, while the analysis of protein behavior on surface may be challenging for hematologists. Therefore, we strive to clarify the logic between the different parts and improve the readability of this article.

We have added related information in the Abstract on Page 2 of the revised manuscript: “Our work not only develops a safe and convenient approach for whole blood auto-transfusion, **but also provides the mechanism of action of self-anticoagulant HMP-modified surfaces.**”

The related information about the mechanistic explorations have been reorganized. Based on the additional results, the original subheading and related content on Page 8-12 of the revised manuscript: “Mechanisms of the inactivation of coagulation factors by the anticoagulant sponges” have been split into four sections including “**FXI adsorption on the anticoagulant sponge blocks the intrinsic coagulation**”, “**The loosely-bound protein layer suppresses the anticoagulant sponge-FXI interaction**”, “**Anticoagulant sponges activated FXI via contact system, but the contribution of activated FXI is minor.**”, and “**Anticoagulant sponges inhibited the common pathway**”.

Other minor comments:

Comment #3: “*The Videos are not particularly enlightening. I could not access the first video, while the second only showed some mice in their cages.*”

Response to comment: Thanks for your valuable comments. We have checked the videos and confirmed with the editor about the accessibility of the videos. In order to facilitate obtaining information, we have also captured some of the content of the videos into pictures. The first video displayed the clotting behaviors of blood collection using MS@D-HMP in the rabbit femoral artery hemorrhage model (**Supplementary Fig. 28**). To compare the clotting times, control whole blood near the wound was also collected without anticoagulant using through a 5-mL syringe. It is obvious that clot formed in the control blood within 5 min, while blood collected using MS@D-HMP remained non-clotted for over 1 h. The second video displayed the healthy survival of rabbits one month after the whole blood auto-transfusion (**Supplementary Fig. 30**).

Supplementary Figure 28. Clotting behaviors of blood collected using MS@D-HMP in the rabbit femoral artery hemorrhage model. To compare the clotting times, control whole blood near the wound was also collected without anticoagulant using through a 5-mL syringe. It is obvious that clot formed in the control blood within 5 min, while blood collected using MS@D-HMP remained non-clotted for over 1 h.

Supplementary Figure 30. Survival of rabbits one month after the whole blood auto-transfusion.

The related figures have been added on Page 18 of the revised manuscript: “Robust anticoagulation lasted over 2 h (Supplementary Video 1, Supplementary Fig. 28)” and “The rabbits woke up within 1 h after auto-transfusion, and lived healthily for over a month (Supplementary Video 2, Supplementary

Fig. 30).”

Comment #4: “In Figure 7 it would be helpful for the reader, if the colors for the heparin and MS@D-HMP samples were kept constant for all analyses.”

Response to comment: Thanks for your good comments. We have unified the colors for the heparin-treated and MS@D-HMP-treated group.

Original Fig. 7

Revised Fig. 7

References

1. Paluck SJ, Nguyen TH, Maynard HD. Heparin-mimicking polymers: synthesis and biological applications. *Biomacromolecules* **17**, 3417-3440 (2016).
2. Choay J, Petitou M, Lormeau JC, Sinay P, Casu B, Gatti G. Structure-activity relationship in heparin - a synthetic pentasaccharide with high-affinity for anti-thrombin-III and eliciting high anti-factor-Xa activity. *Biochem. Biophys. Res. Commun.* **116**, 492-499 (1983).
3. Vanboeckel CAA, Petitou M. The unique antithrombin-III binding domain of heparin - a lead to new synthetic antithrombotics. *Angew. Chem. Int. Ed.* **32**, 1671-1690 (1993).
4. Petitou M, *et al.* Synthesis of thrombin-inhibiting heparin mimetics without side effects. *Nature* **398**, 417-422 (1999).
5. Grootenhuis PDJ, Westerduin P, Meuleman D, Petitou M, Vanboeckel CAA. Rational design of synthetic heparin

- analogs with tailor-made coagulation-factor inhibitory activity. *Nat. Struct. Biol.* **2**, 736-739 (1995).
6. Herbert JM, *et al.* SR123781A, a synthetic heparin mimetic. *Thromb. Haemost.* **85**, 852-860 (2001).
 7. Avci FY, Karst NA, Linhardt RJ. Synthetic oligosaccharides as heparin-mimetics displaying anticoagulant properties. *Curr. Pharm. Des.* **9**, 2323-2335 (2003).
 8. Demir M, *et al.* Anticoagulant and antiprotease profiles of a novel natural heparinomimetic mannopentaose phosphate sulfate (PI-88). *Clin. Appl. Thromb./Hemost.* **7**, 131-140 (2001).
 9. Hayakawa Y, Hayashi T, Hayashi T, Niiya K, Sakuragawa N. Selective activation of heparin-cofactor-II by a sulfated polysaccharide isolated from the leaves of artemisia-princeps. *Blood Coagul. Fibrinolysis* **6**, 643-649 (1995).
 10. Monien BH, Cheang KI, Desai UR. Mechanism of poly(acrylic acid) acceleration of antithrombin inhibition of thrombin: implications for the design of novel heparin mimics. *J. Med. Chem.* **48**, 5360-5368 (2005).
 11. Al Nahain A, Ignjatovic V, Monagle P, Tsanaktsidis J, Ferro V. Heparin mimetics with anticoagulant activity. *Med. Res. Rev.* **38**, 1582-1613 (2018).
 12. Coombe DR, Kett WC. Heparin mimetics. *Handb. Exp. Pharmacol.*, 361-383 (2012).
 13. Ito Y, Liu L, Imanishi Y. Interaction of poly(sodium vinyl sulfonate) and its surface graft with antithrombin-III. *J. Biomed. Mater. Res.* **25**, 99-115 (1991).
 14. Jeske W, Fareed J. Antithrombin III-mediated and heparin-cofactor II-mediated anticoagulant and antiprotease actions of heparin and its synthetic analogs. *Semin. Thromb. Hemost.* **19**, 241-247 (1993).
 15. Onishi M, Miyashita Y, Motomura T, Yamashita S, Sakamoto N, Akashi M. Anticoagulant and antiprotease activities of a heparinoid sulfated glucoside-bearing polymer. *J. Biomater. Sci. Polym. Ed.* **9**, 973-984 (1998).
 16. Sugidachi A, Asai F, Koike H. Anticoagulant and antiprotease activities of aprosulate sodium, a new synthetic polyanion, in human plasma and purified systems. *Blood Coagul. Fibrinolysis* **5**, 773-779 (1994).
 17. Tamada Y, Murata M, Hayashi T, Goto K. Anticoagulant mechanism of sulfonated polyisoprenes. *Biomaterials* **23**, 1375-1382 (2002).
 18. Sano M, Tamada Y, Niwa K, Morita T, Yoshino G. Sulfated sericin is a novel anticoagulant influencing the blood coagulation cascade. *J. Biomater. Sci. Polym. Ed.* **20**, 773-783 (2009).
 19. Charef S, TaponBretaudiere J, Fischer AM, Pfluger F, Jozefowicz M, Labarre D. Heparin-like functionalized polymer surfaces: discrimination between catalytic and adsorption processes during the course of thrombin inhibition. *Biomaterials* **17**, 903-912 (1996).
 20. Fougnot C, Jozefowicz M, Rosenberg RD. Catalysis of the generation of thrombin-antithrombin complex by insoluble anticoagulant polystyrene derivatives. *Biomaterials* **5**, 94-99 (1984).
 21. Belattar N. Affinity adsorption of human vitamin K-dependent coagulation factor IX onto heparin-like poly(styrene sodium sulfonate) adsorbent. *Mater. Sci. Eng. C* **27**, 849-854 (2007).
 22. Fougnot C, Jozefowicz M, Bara L, Samama M. Interactions of anticoagulant insoluble modified polystyrene resins with plasmatic proteins. *Thromb. Res.* **28**, 37-46 (1982).
 23. Song X, *et al.* Transient blood thinning during extracorporeal blood purification via the inactivation of coagulation factors by hydrogel microspheres. *Nat. Biomed. Eng.* **5**, 1143-1156 (2021).
 24. Mei Y, *et al.* Polymer-nanoparticle interaction as a design principle in the development of a durable ultrathin universal binary antibiofilm coating with long-term activity. *ACS Nano* **12**, 11881-11891 (2018).

Reviewer #3 (Remarks to the Author):

“Cell salvage systems are sometimes used during surgery to avoid allogeneic blood transfusion. Such systems typically involve the collection of shed blood into heparinized saline, filtration, and washing of the blood, followed by reinfusion. To simplify this process, Xu and colleagues have developed sponges impregnated with heparin-mimetic polymer (HMP)-modified microspheres. These sponges adsorb clotting proteins (particularly factor [F] XI) and calcium, thereby decreasing the coagulability of the blood and rendering it amenable to reinfusion. Although the proposed technology is interesting, there are problems that need to be addressed. These can be divided into major and minor concerns.”

Response to the general comment:

Thanks for your valuable scientific comments and suggestions to improve the quality of our manuscript. Based on your suggestions, the manuscript has been carefully corrected and all the necessary data have been added to support our claims. We also understand and agree with your concerns on the unclear issues, which we think that these comments are highly important for us to further improve the quality of this manuscript. We have systematically responded to all your questions, such as the readability of the manuscript, the advances of anticoagulant sponge to this field, the confusing terminology, the binding capacity of sponge, the underlying mechanisms of prolonged TT values, the effect of soluble HMP versus HMP-modified sponges, contact activation induced by sponge, and inconsistent use of statistical tests, etc. For the detailed comments, please see below our point-to-point response. Finally, thank you very much for reviewing our manuscript, and looking forward to receiving your positive responses.

MAJOR

Comment #1: *“The paper is densely written, which makes it difficult to follow.”*

Response to comment: Thanks for your good comment. We have carefully reviewed our writing style and have taken the following steps to address this issue:

(I) Reorganizing content

The content has been reorganized (particularly in the parts of mechanistic exploration) in a logical and intuitive manner to improve the overall coherence and readability of the article. We have ensured that the article is well-structured, with clear headings and subheadings to guide readers through the content. For example,

The original subheading and related content on Page 8-12 of the revised manuscript: “Mechanisms of the inactivation of coagulation factors by the anticoagulant sponges” have been split into four sections including “FXI adsorption on the anticoagulant sponge blocks the intrinsic coagulation”, “The loosely-bound protein layer suppresses the anticoagulant sponge-FXI interaction”, “Anticoagulant sponges activated FXI via contact system, but the contribution of activated FXI is minor”, and “Anticoagulant sponges inhibited the common pathway”.

The original subheading and related content on Page 16-18 of the revised manuscript: “Efficacy and safety of blood salvage and reinfusion *in vivo*” have been split into two sections including “Collection of rabbit whole blood using the anticoagulant sponges” and “Efficacy and safety of whole blood auto-transfusion using the anticoagulant sponges *in vivo*”.

Furthermore, we have removed most of the uncertain and speculative analyses, or moved them to the supplementary information to ensure that the main article focuses on the most important and reliable conjectures.

(II) Adding details

Combined with our reply to your **Comment #9**, each figure legend has been carefully checked to ensure that each caption can be understood in isolation from the main text. Each legend begins with a brief title sentence for the whole figure and continues with a short statement of what is depicted in the figure. The experimental conditions and statistical analysis methods are added when appropriate.

(III) Simplifying language

The language used in the article has been simplified to make it more accessible to a broader audience. Combined with our reply to your **Comment #3**, we have eliminated technical jargon when possible and have used clear, concise language to convey our research findings. In addition, most of complex sentences have been broken up into shorter, more manageable chunks to improve readability and comprehension. We believe that each sentence is clear and concise, and that ideas flow logically from one sentence to the next.

(IV) Polishing language

We have sought input from a professional native speaker to provide feedback on the writing style and structure of the article. We have incorporated their suggestions to ensure that the article is clear, concise, and effective in conveying our findings to the intended audience.

We believe that these steps have addressed the issue of dense writing style in our article, and we are confident that our findings are now more accessible and understandable to a wider audience. The related modifications have been implemented in the revised manuscript and the revised supplementary information.

Comment #2: *“The authors need to better identify how the current study advances the field beyond previous publications on HMP-modified microspheres by them (e.g., references 25, 32, and 59) and others (e.g., references 33 to 35).”*

Response to comment: Thanks for your good comments. We believe that our work provides substantial insights in the exploration of anticoagulant mechanisms and the development of a new application using the self-anticoagulant concept. Unlike the previous report, in this work we have shown that the anticoagulant concept needs to be further developed to whole blood auto-transfusion and keep the whole blood without clotting, and should have transfusion quality. We believe that our contributions in these two areas (material development for a new application, and mechanism of heparin-mimics on a surface) can help to drive further development of heparin mimetics.

(I) The exploration of anticoagulant mechanisms of heparin-mimetic polymer (HMP)-modified surface

The limitations of heparin have led to the development of novel heparin mimetics as alternative anticoagulants.¹ Early heparin mimetics were developed with a tailored uronic backbone (reference 35 included²), inspired by the specific AT-binding pentasaccharide sequence responsible for heparin’s anticoagulant activity.^{3, 4} Extensive research has been conducted to understand the anticoagulant

mechanisms of these heparin mimetics, most of which were based on antithrombin III (ATIII)- or heparin cofactor II (HCII)-activating activity.⁵⁻¹⁰ However, as the scope of heparin mimetics has expanded to include representative features that mimic the functional groups or highly negative charge of heparin, the development of new heparin mimetics has outpaced in-depth research into their anticoagulant mechanisms.^{1, 11-13} While these heparin mimetics have exerted similar anticoagulant effects to heparin at a macroscopic level (such as prolonged aPTT and TT), the diversity in their structures has resulted in a wide range of molecular anticoagulant mechanisms, including ATIII-,^{14, 15} HCII-activating activity,¹⁶ inhibition of factor Xase activity,¹⁷ and interference with polymerization of fibrin monomer.^{18, 19} Moreover, after introducing heparin mimetics onto the surface of the material, the anticoagulant mechanism at the interface becomes even more elusive, especially with regards to the potential bleeding side effects, and the mechanism of surface-protein interaction which is important for the surface to function as a self-anticoagulant material.²⁰⁻²³

We have already contributed to exploring the underlying anticoagulant mechanisms of HMP-modified surface in our previous studies such as HMP-modified microspheres (reference ^{25, 24}). However, these contributions were limited to the preliminary determination of the binding of intrinsic coagulation factors, leaving a substantial gap in our understanding of the detailed mechanisms. Specifically, we have yet to fully comprehend how the material mediates the binding and inactivation of specific coagulation factors (e.g., significantly inhibited FXI), the relationship between coagulation factors binding (e.g., contact system-related FXII and FXI) and surface-induced contact activation, etc. To this end, we seek to delve deeper into this area by developing HMP-modified sponge. We recognize the importance of exploring material-protein interactions as a means to analyze these processes thoroughly. This perspective will provide valuable insights into how the material interacts with proteins, enabling us to examine the binding and inactivation of coagulation factors mediated by the material itself. Furthermore, the formation of a protein layer on the material surface plays a crucial role in bridging the gap between the material and the blood system. This protein layer serves as a vital link, facilitating a systematic examination of the material's influence on the blood system. Through this approach, we can gain a deeper understanding of how the material impacts the overall functionality of the blood system and derive pertinent usage precautions.

Finally, our mechanistic exploration demonstrates the multiple mechanisms of MS@D-HMP to reduce clotting: (i) inhibition of intrinsic pathway mainly through the adsorption of FXI. The loosely-bound protein layer suppresses the anticoagulant sponge-FXI interaction. (ii) chelation of calcium ions. (iii) inhibition of common pathway through inhibition of thrombin. Inhibition of thrombin might not rely on ATIII and HCII like heparin does. These provide sufficient theoretical support for the efficacy for MS@D-HMP in short-term anticoagulation. It ensures the effective anticoagulation of the sponge against blood with disease states. Recovering blood with introduced tissue factor and prematurely activated thrombin relies on the ability to inhibit thrombin and chelate calcium ions, while activation of intrinsic pathway via contact system during storage needs to be blocked by depletion of FXI and chelation of calcium ions. In addition, the safety of sponge was confirmed with negligible effects on blood cells and the complement system. Based on the well-established mechanisms, the potential failure risks of this surface are also predicted and confirmed, such as the contact activation, and the possibility of surface-induced coagulation activation after recovery of coagulation factors and calcium ions *in vivo*. We believe our work sheds new light on a long-standing question on what governs hemocompatibility, elucidating that the unique protein layer acts as a bridge connecting the bio-

interface with the blood system. The proposed research paradigm can facilitate the in-depth investigation of complex interactions between other HMP-based materials and the blood system. Therefore, our material design and mechanistic exploration hold the potential to stimulate further research and advance the progress of this field.

(ii) The development of a new application based on a new material to recover blood and auto-transfusion.

Our experience with HMP-modified microspheres as solid anticoagulants in extracorporeal blood circuits in our previous work²⁴ was important initial research into this new application. Although some understanding on the anticoagulation mechanism is explored, a deeper study is warranted for further exploration of different applications in HMP research. Our published self-anticoagulant microspheres will not be suitable for blood salvage and re-transfusion application. So, we have to build the self-anticoagulant idea further and develop a sponge that can absorb whole blood rapidly without activation and preserve quality of the blood for reinfusion. Thus, we embarked on the work described here. The development of a novel application provides a steady stream of impetus for the further development and translation of the HMP-modified materials. This is exactly the reason that we have conceived another important application for the HMP-modified surface in whole blood auto-transfusion. We believe that the systematic validation of HMP-modified sponge in this clinically unmet need with huge market prospects can also increase research community's enthusiasm for such self-anticoagulant materials.

The use of anticoagulant sponge for auto-transfusion has substantial advantages compared with traditional technology. In a highly clean and sterile condition, various blood cells and plasma proteins could be collected by sponge maintaining their normal state, and the reinfusion of blood facilitates the faster recovery of coagulation function *in vivo*. In unexpected situations, such as car accidents, battlefield scenarios, etc., portable blood salvage might be achieved by using the sponge to provide a valuable window of opportunity for subsequent clinical operations. This is a breakthrough which has never been realized, as there is no suitable and portable device to recover blood cells directly from the patient's wound outside hospital setting, highlighting the distinction between sponge and traditional auto-transfusion technology.

In conclusion, developing this novel application scenario not only addresses a clinically urgent need with significant translational potential but also fuels further development and translation of HMP-modified materials. The systematic validation of HMP-modified sponge in the field of whole blood auto-transfusion has substantial benefits and paves the way for exciting advancements in this area.

All these points have been discussed in our revised manuscript accordingly. Especially, to emphasize the novelty and need of our work more strongly, we have added related discussion in the Abstract on Page 2 of the revised manuscript: “Our work not only develops a safe and convenient approach for whole blood auto-transfusion, but also provides the mechanism of action of self-anticoagulant HMP-modified surfaces.” and in the Outlook on Page 20 of the revised manuscript: “The portability and easy handling ability of the sponge also open up the possibility of multi-scenario applications (such as battleground, car accidents, etc.). In these unexpected situations, only anticoagulant sponge can recover blood cells directly from the patient's wound outside hospital setting, highlighting the distinction between sponge and traditional auto-transfusion technology.” and in the Outlook on Page 20-21 of the revised manuscript: “However, both the molecular anticoagulant mechanism of heparin

mimetics and the anticoagulant mechanism of the heparin mimetics-modified materials have not been thoroughly explored. After introducing heparin mimetics onto the surface of a material, the anticoagulant mechanism at interface becomes even more elusive, especially with respect to the side effects, and the mechanism of surface-protein interaction which contributes to its anticoagulant function. Carboxylated and sulfonated polymers have been widely adopted for heparin-mimetic modification due to its well-established methods^{24, 71, 72}. Through systematic exploration of the coagulation reactions at the molecular level, we revealed that the carboxylated and sulfonated HMP-modified sponges effectively adsorb FXI to the surface. In addition, calcium ions were depleted from the plasma. An unknown thrombin inhibitor might be activated by sponge, even in the collected plasma when the sponge was removed.”

Comment #3: “*The terminology used in this paper is confusing. The sponges appear to reduce the coagulability of blood by adsorbing clotting factors, particularly those in the intrinsic pathway of coagulation, and calcium. However, the authors use the terms “adsorption” and “inactivation” interchangeably and introduce terms such as “salvage”, “passed through” (page 7, line 179), and “refusion” without providing clear definitions.*”

Response to comment: Thanks for your good comments, and we apologize for the confusing terminology. We used the terms “adsorption” and “inactivation” interchangeably to give more accurate descriptions for the effects of the material on the blood. MS@D-HMP inhibited intrinsic pathway by reducing the overall activity of corresponding intrinsic coagulation factors (particularly FXI). Such decreased activity of coagulation factors was defined as the term “inactivation”. This inactivation is achieved by the adsorption of FXI. The term “adsorption” here referred to the transfer of FXI from the plasma to the surface of the MS@D-HMP. Of note, there may be other reasons for the inactivation of FXI, which was why it was important to make this distinction. For example, the inactivation of FXI can be derived from anticoagulant substances in plasma (C1 inhibitor and antithrombin III) when FXI had been activated to FXIa via contact system. Such speculation was highly possible, but contributed little to the overall inactivation of FXI in plasma. Hence, the term “adsorption” was used when analyzing the results of proteomics, where we are confident that the decreased relative abundance of FXI in the plasma resulted from the adsorption by sponge. When we want to describe the overall effect of sponge on FXI, we use the term “inactivation” to cover all possibilities especially used when functional analyses have been described. This information has been added to the main text.

MS@D-HMP also inhibited common pathway as indicated by prolonged TT values. The detailed reasons were discussed in the **Comment #5** below. In general, prolonged TT values show the decreased concentration of fibrinogen in sample plasma, or inhibition of bovine thrombin (added TT reagent) by substances in sample plasma. However, the adsorption of fibrinogen by the MS@D-HMP was proved to contribute little to the prolongation of TT values. The effects of MS@D-HMP on antithrombin III and heparin cofactor II might also be minimal. After that, we speculated that the inhibition of thrombin may be derived from the unknown inhibitor generated in the MS@D-HMP-incubated plasma. In this case, “adsorption” failed to account for the underlying effects of material on common pathways, which is why another term “inhibition” was used.

The term “salvage” was derived from the medical terms “autologous blood salvage (also called auto-transfusion)”, which refers to the procedures to recover and wash the patient’s blood, and then reinfuse the red blood cells to the patient. We used it inappropriately to describe the single process of using a

sponge to recover blood. We had intended to use the term “passed through” to describe the specific procedures by which the blood was incubated with the sponge, and then separated from the sponge for TEG tests. The term “refusion” was a spelling mistake, and the corrected one should be “reinfusion”, which means transfusion of the collected patient’s blood back to the patient. To avoid misunderstanding, these confusing terms have been replaced with more accurate and direct descriptions. For example:

(I) The term “pass through”

Page 6 of the revised manuscript: “a straight-line TEG profile was observed when recalcified whole blood **passed through** the MS@D-HMP” **have been corrected with “The TEG profile for the MS@D-HMP-incubated whole blood showed a straight line”**.

(II) The term “refusion”

Page 12 of the revised manuscript: “to investigate the coagulation function after **refusion** of plasma with different anticoagulant treatment. **Refusion** of heparin-anticoagulant plasma delayed the clotting times when the **refusion** concentration exceeded 0.02 mL/mL normal PPP...when the **refusion** concentration reached 0.1 mL/mL normal PPP...since the sponge is removed before **refusion**...until the **refusion** concentration reached 1 mL/mL normal PPP” **have been corrected with “to correct clotting times by **mixing** sample plasma with fresh normal plasma at different concentrations. Heparin-anticoagulant plasma delayed clotting times when the **mixing** concentration exceeded 0.02 mL/mL normal PPP...when the **mixing** concentration reached 0.1 mL/mL normal PPP ...until the **mixing** concentration reached 1 mL/mL normal PPP”**.

Page 18 of the revised manuscript: “returned to normal levels by 20 min after **refusion**...higher aPTT (38.7 s) and TT (100 s, detection limit) values were observed by 20 min after **refusion**, which resulted from the inhibition of downstream thrombin by **refusion** of heparin in the salvaged blood...FDP and D-dimer levels pre- and post-**refusion**...The blood cell levels and biochemical parameters of rabbits were measured throughout the **refusion**” **have been corrected with “returned to normal levels by 20 min after **reinfusion**...higher aPTT (38.7 s) and TT (100 s, detection limit) values by 20 min after **reinfusion**, likely due to the inhibition of thrombin by the **return** of heparin ...FDP and D-dimer levels pre- and post-**reinfusion**...The blood cell levels and biochemical parameters of rabbits were measured throughout the **refusion**”**.

Page 20 of the revised manuscript: “the **refusion** of sponge-treated blood” **have been corrected with “*in vivo* **reinfusion** of sponge-collected blood”**.

Page 21 of the revised manuscript: “in the patients with **refusion**...which ensures the safe **refusion** of recovered blood without anticoagulant” **have been corrected with “in the patients with **reinfusion**...which ensures the safe **reinfusion** of collected blood without anticoagulant”**.

Page 23 of the revised manuscript: “the efficacy and safety of whole blood **salvage** and **refusion** by the MS@D-HMP...For **refusion**, approximately 15 mL of the collected whole blood” **have been corrected with “the efficacy and safety of whole blood **auto-transfusion** by the MS@D-HMP...For **reinfusion**, approximately 15 mL collected whole blood”**.

(III) The term “salvage”

Page 2 of the revised manuscript: “the whole blood **salvage** in trauma-induced hemorrhage...The transfusion of **salvaged** blood...reduction in the number of **salvaged** red blood cells...the hemostatic response of the **salvaged** blood...the volume of blood to be **salvaged**...large volumes of **salvaged** RBCs” have been corrected with “the whole blood **auto-transfusion** in trauma-induced hemorrhage...The transfusion of **collected** blood...decrease in the number of **collected** red blood cells...the hemostatic response during blood **collection**...the volume of blood to be **collected**...large volumes of **collected** RBCs”.

Page 3 of the revised manuscript: “in the **salvaged** blood...blood **salvage** without the addition of anticoagulants...simultaneous blood collection with instantaneous anticoagulation for blood **salvage**...a rapid and convenient way for whole blood **salvage**” have been corrected with “in **collected** blood...blood **auto-transfusion** without additional anticoagulants...simultaneous blood collection with instantaneous anticoagulation for blood **auto-transfusion**...a rapid and convenient method for whole blood **auto-transfusion**”.

Page 4 of the revised manuscript: “a rapid sorption process facilitating blood **salvage**” have been corrected with “a rapid sorption process that facilitates blood **collection**”.

Page 5 of the revised manuscript: “Design and characterization of the sponges for instantaneous blood **salvage**” have been corrected with “Design and characterization of the sponges for instantaneous blood **auto-transfusion**”.

Page 6 of the revised manuscript: “the plasma **salvaged** by the MS@D-HMP... in a human whole blood **salvage** model... Whole blood without any anticoagulant was **salvaged** by the sponges” have been corrected with “the plasma **after incubation with MS@D-HMP**... in human whole blood **salvage model**... Whole blood was **collected** without anticoagulant using sponge”.

Page 8 of the revised manuscript: “the blood **salvage** model...blood **salvaged** by MS@D-HMP” have been corrected with “the whole blood (without anticoagulant) **after incubation with MS@D-HMP**... blood **after incubation with MS@D-HMP**”.

Page 12 of the revised manuscript: “thrombin generation and amplification in the **salvaged** blood” have been corrected with “thrombin generation and amplification in the **collected** blood”.

Page 16 of the revised manuscript: “Efficacy and safety of blood **salvage** and reinfusion *in vivo*” have been corrected with “**Collection** of rabbit whole blood using the anticoagulant sponges”.

Page 17 of the revised manuscript: “the blood **salvage** performance and reinfusion safety” have been corrected with “blood **collection** and reinfusion safety”.

Page 18 of the revised manuscript: “reinfused with the **salvaged** blood...inhibition of downstream thrombin by reinfusion of heparin in the **salvaged** blood” have been corrected with “reinfused with the **collected** blood...inhibition of thrombin by the **return** of heparin.”

Page 20 of the revised manuscript: “the **salvaged** blood was free from washing...the salvaged blood is **separated** from the coatings” have been corrected with “The **collected** blood using the sponge required no washing...**collected** blood is separated from the coatings”.

Page 20-21 of the revised manuscript: “in the **salvaged** plasma when the sponge was removed...Given our target application for blood **salvage**” have been corrected with “in the **collected** plasma when the

sponge was removed...Given our target application for blood **auto-transfusion**".

Page 23 of the revised manuscript: "the efficacy and safety of whole blood **salvage**" have been corrected with "the efficacy and safety of whole blood **auto-transfusion**".

Comment #4: "Despite many *in vitro* experiments, the mechanism by which the sponges reduce the coagulability of the blood remains unclear. What is the capacity of the sponges to adsorb clotting factors and calcium? This question would be best addressed using purified proteins or calcium and standardized recovery assays. Such assays could be used to determine the binding capacity of the sponges and to determine whether the sponges activate or inactivate the clotting factors."

Response to comment: Thanks for your professional comments. We concluded the mechanisms of MS@D-HMP to reduce coagulability as (i) inhibition of intrinsic pathway mainly through adsorption of FXI to the surface; (ii) chelation of calcium ions; (iii) inhibition of common pathway via inactivation of thrombin. The suggested experiments could no doubt help us shed light on the actual anticoagulant performance, and the underlying anticoagulant mechanisms of sponge. We would answer this question in the following aspects:

(I) The adsorption capacity of sponge to coagulation factors and calcium ions

For the **chelation of calcium ions**, we initially investigated the time-dependent calcium ions adsorption behavior by MS@D-HMP in a solution of calcium ions (2 mM, physiological concentration). MS@D-HMP showed a rapid calcium ions adsorption kinetics with an equilibrium adsorption capacity at 19.6 $\mu\text{g}/\text{mg}$ (**Supplementary Fig. 14a**). Then, the efficacy of calcium ions adsorption was confirmed in plasma environment. Hirudin-anticoagulated plasma with different initial calcium ion concentrations was obtained by addition of stock solution of calcium ions. The resulted plasma was incubated with MS@D-HMP for 10 min, and the adsorption amount was determined. Although high depletion efficiency (85.1 %) was achieved in the plasma environment with an initial calcium ions concentration of 2 mM, the equilibrium adsorption capacity was only 0.60 $\mu\text{g}/\text{mg}$, which was much lower than that in the calcium ions solution (**Supplementary Fig. 14b** and Supplementary Table 6). The decreased equilibrium calcium ions adsorption capacity in plasma might result from the competitive effect of other cations in the plasma and the shorter incubation time. Hence, we recommend to refer the equilibrium binding capacity of calcium ions for the MS@D-HMP measured in plasma conditions.

Supplementary Figure 14. Calcium ions adsorption behaviors of MS@D-HMP in the solution of calcium ions and plasma. a

Calcium ions adsorption amount of MS@D-HMP in calcium ions solution (2 mM) at different incubation time (n = 3 independent samples, mean ± SD). **b** Equilibrium binding capacity and the corresponding depletion efficiency of calcium ions by MS@D-HMP in hirudin-anticoagulated plasma environment (n = 3 biologically independent samples, mean ± SD).

For depletion of FXI, we only discussed the equilibrium binding capacity since the sponge cannot reach the saturated binding capacity under practical conditions. The equilibrium FXI binding capacity and FXI inactivation efficiency by MS@D-HMP were determined using FXI-deficient plasma after addition of different amounts of purified FXI protein. 6 mg sponge was exactly immersed in 100 μ L plasma for 30 min to ensure adequate contact between sponge and plasma. Interestingly, equilibrium FXI binding capacity by the sponge reached 80.6 ng/mg with physiological level of FXI (5 μ g/mL), and it almost increased proportionally with the initial FXI concentration (**Fig. 3e**). FXI inactivation efficiency reached ~ 98 % with physiological FXI level. High FXI inactivation efficiency (85.8 %) remained even when the initial FXI concentration increased to 80 μ g/mL (16 times the physiological level). Such FXI binding and inactivation efficiency that the sponge achieved at ideally high initial FXI concentration should promise a robust anticoagulant ability that was overqualified for our practical application. Although saturated binding capacity might be obtained with further increasing initial FXI concentration and operation time, such ideal conditions are not available in practical use. Hence, the saturation FXI binding capacity may sometimes not be direct indicative of the practical anticoagulant ability, which necessitates the preliminary experiments to ensure a high factor of safety.

Fig. 3 e Equilibrium binding capacity and the corresponding inactivation efficiency of FXI by MS@D-HMP. FXI-deficient plasma (100 μ L) supplemented with different amounts of FXI proteins was incubated with MS@D-HMP (6 mg) for 30 min (n = 3 biologically independent samples, mean ± SD).

For inhibition of thrombin, inactivation was independent of adsorption, but relied on an inhibitor generated in plasma after incubation with MS@D-HMP. Identifying such inhibitor is a crucial step to achieve the precise quantification of the capacity, but can hardly be realized without appropriate high-throughput and *in-situ* technical means. Actually, efforts have been made to exclude the probability of ATIII and HCII in our reply to your **Comment #5**. Understanding the mechanism of thrombin inhibition induced by HMP-modified surface is of interest in our future work.

(II) The contributions of equilibrium adsorption capacity to optimize practical adsorption process

We fully agree with you about the fact that binding capacity (particularly saturated binding capacity

and equilibrium binding capacity) is a good index to evaluate the adsorption performance of adsorbent and then optimize adsorption process. We have mentioned above that the saturated adsorption capacity (determined in single adsorbate/buffer solution, sufficient adsorbate concentration and infinite time) tend not to be reached under practical conditions. Hence, the equilibrium adsorption capacity has been determined. But sometimes such equilibrium adsorption capacity may not be reached when use because the sponge is meant to be disposable. The main reason for single use only is that exposure of bacteria and lipopolysaccharide during long-term use may increase the risk of sepsis in patients. To ensure a high factor of safety, slight overdose of sponge is also preferred to avoid the anticoagulation failure.

The equilibrium binding capacity under different initial adsorbate concentrations has also been compared with the corresponding actual capacity when used in physiological level. We found that it is still difficult to estimate the appropriate amount of sponge under practical conditions on the basis of equilibrium binding capacity alone. Patients' blood may vary with disease states, which cause different pressures on three anticoagulant behaviors of the sponge. For example, recovering blood with introduced tissue factors and thrombin prematurely activated relies on the ability to inhibit thrombin and chelate calcium ions, while activation of intrinsic pathway via contact system during storage needs to be blocked by depletion of FXI and chelation of calcium ions. Considering the inconsistency between the equilibrium binding capacity and practical anticoagulant performance, it is recommended to re-consider the amount of sponge used in combination with the actual capacity pre-determined from the applied conditions.

(III) Other information we can obtain from exploring the adsorption of coagulation factors

The results of equilibrium binding capacity of FXI by sponge at different initial FXI concentration were used to fit the classical adsorption isotherms, Langmuir isotherm model and Freundlich isotherm model, which could help describe the interaction between FXI and sponge. Interestingly, Freundlich isotherm model, rather than Langmuir isotherm model, described the adsorption process better (**Table R1**). Freundlich's adsorption isotherm is empirical while Langmuir's adsorption isotherm is theoretical. The basic assumptions of the Langmuir isotherm model are (i) monolayer adsorption; (ii) the distribution of adsorption sites is homogeneous; (iii) the adsorption energy is constant; (iv) the interaction between adsorbate molecules is negligible.²⁵ Based on the results of two-stage successive incubations coupled with clotting time tests (**Fig. 3f**), we speculate that the deviation from the Langmuir isotherm model was probably not due to multi-layer adsorption. If sponge adsorbed FXI through the multi-layer adsorption, the FXI adsorption behavior should have changed with the development of the tightly-bound protein layer on the surface at different times. However, the sponge still exhibited strong FXI adsorption after great change of the tightly-bound protein layer on the surface. It is more likely that the surface of sponge was not uniform, and the distribution of adsorption sites on the surface was heterogeneous.

Table R1 Fitted Langmuir isotherm model and Freundlich isotherm model and the corresponding parameters for the adsorption of FXI to MS@D-HMP in plasma environment. Langmuir isotherm, and Freundlich isotherm are applied, since they are the most widely used adsorption isotherms to study the adsorption of solute from liquid solutions. The Langmuir model assumes that all adsorption sites are homogeneous, the adsorption process is a dynamic balance, the adsorption occurs in monolayer, and the adsorbed molecules are all independent. It can be expressed by equation R1, where C_e ($\mu\text{g/mL}$) is the concentration of the FXI at equilibrium; q_e ($\mu\text{g/mg}$) is the amount of FXI adsorbed by the unit mass after the adsorption reaches equilibrium; q_{max} ($\mu\text{g/mg}$) is the adsorption capacity; K_L

(mL/μg/mg) is the Langmuir adsorption constant. Freundlich isotherm is an empirical equation used to describe the heterogeneous system and it depicts reversible adsorption. It can be expressed by equation R2, where $1/n$ is the constant which incorporates factors affecting the adsorbed amount at equilibrium; k_F is the Freundlich isotherm constant ((μg/mg)(mL/μg)^{1/n}); the meanings of q_e (μg/mg) and C_e (μg/mL) are the same as those in the Langmuir equation.

	Langmuir isotherm model	Freundlich isotherm model
Equation	$\frac{C_e}{q_e} = \frac{1}{q_{max}k_L} + \frac{C_e}{q_{max}}$ (R1)	$q_e = k_F C_e^{1/n}$ (R2)
Parameter	q_{max} (μg/mg) = 1.094; k_L (mL/μg) = 0.352	$n = 2.191$; k_F ((μg/mg)(mL/μg) ^{1/n}) = 0.261
R ²	0.6486	0.9654

Since the secondary structure of proteins could reconfigure or even be compromised due to the interaction between the surface and the proteins,^{26, 27} the diversity in the adsorption sites might increase the possibility of protein denaturation and activation. Direct activation of FXI on the surface may pose a significant security risk as adsorption of FXI on the surface accounted mainly for the depletion of FXI by sponge. Fortunately, this case has been considered in the fluorometric-based FIXa activity assay (Fig. 4a), and the results confirmed that sponge would not directly induce activation of FXI (Fig. 4b, f). Overall, the point on the heterogeneous adsorption of FXI on the surface remains speculative. Designing surfaces that are more conducive for such mechanistic exploration is of our future interest. Through this, more direct evidence may be provided to support our hypothesis and shed light on the interaction between FXI and HMP-modified surfaces.

The related discussions about calcium ions binding capacity have been added on Page 6 of the revised manuscript: “In a solution of calcium ions (2 mM, physiological concentration), MS@D-HMP showed a rapid calcium ions adsorption kinetics with an equilibrium adsorption capacity at 19.6 μg/mg (Supplementary Fig. 14a). Calcium ions depletion efficiency remained high (85.1 %) in the plasma environment, although a decreased equilibrium adsorption capacity (0.60 μg/mg) was observed due to the competitive effect of other cations and shorter incubation time (Supplementary Fig. 14b and Supplementary Table 6).”

The related discussions about FXI binding capacity have been added on Page 8 of the revised manuscript: “Strong sponge-FXI interaction was presented beforehand through equilibrium FXI binding capacity and FXI inactivation efficiency in plasma with different initial FXI concentrations (Fig. 3e). Of note, the FXI inactivation efficiency reached ~98 % with physiological FXI concentration (5 μg/mL), thus ensuring a high factor of safety. The equilibrium FXI binding capacity by sponge reached 80.6 ng/mg with a physiological FXI concentration, and almost increased proportionally with the initial FXI concentration. High FXI inactivation efficiency (85.8 %) remained even when the initial FXI concentration increased to 80 μg/mL (16 times the physiological concentration), which should promise a robust anticoagulant ability when used for blood auto-transfusion.”

The experimental details about determining calcium ions binding capacity have been added on Page 5-6 of the revised supplementary information: “The calcium ions adsorption kinetics by MS@D-HMP was determined in calcium ions solution. 20 mL calcium ions solution (2 mM) was incubated with 20 mg MS@D-HMP at 37 °C for a certain time (5, 15, 30, 60, 240, 480, 1440 min). The residual concentrations of calcium ions were determined by an atomic absorption spectrometer (Shimadzu

SPCA-626D, Japan), and then the adsorption amounts were calculated.

The calcium ions adsorption by MS@D-HMP was investigated in plasma. Certain amounts of calcium ions were added into 400 μL hirudin anticoagulant plasma. The re-calcified plasma was incubated with 10 mg MS@D-HMP for 10 min. The calcium concentration in the treated plasma was determined by a biochemical analysis instrument (Cabas C311, Roche, Switzerland). Then the adsorption amounts were calculated. This is also a preliminary experiment for the posterior fluorescence-based FIXa activity assay as FIXa activates FIX to FIXa in a calcium-dependent way.”

The experimental details about determining FXI binding capacity have been added on Page 8 of the revised supplementary information: “The equilibrium binding capacity and the corresponding inactivation efficiency of FXI by MS@D-HMP were determined using the purified FXI protein. A certain amount of FXI native protein was mixed with 100 μL FXI-deficient plasma to obtain the final FXI concentrations of 2.5, 5, 10, 20, 40 and 80 $\mu\text{g}/\text{mL}$. The FXI-supplemented plasma was incubated at 37 $^{\circ}\text{C}$ for 30 min, and then incubated with 6 mg MS@D-HMP for another 30 min at 37 $^{\circ}\text{C}$. The residual FXI concentration of the incubated plasma was determined based on a standard curve for FXI concentration versus FXI activity. Gradient-diluted FXI-supplemented plasma was prepared, and the resulted activity was detected using an automatic coagulation analyzer (ACL Elite Pro, werfan) based on similar method as described in **Experimental Section 5.1**. The equilibrium binding capacity and the corresponding inactivation efficiency can be calculated as follows:

$$\text{Equilibrium binding capacity} = \frac{V(C_0 - C_e)}{m} \quad (2)$$

$$\text{Inactivation efficiency} = \frac{C_0 - C_e}{C_0} \times 100 \% \quad (3)$$

where C_0 and C_f are the concentrations of FXI in plasma before and after incubation with MS@D-HMP. V is the volume of the plasma, m is the weight of MS@D-HMP used.”

Comment #5: “The observation that the sponges prolong the activated partial thromboplastin time (aPTT) and the thrombin time (TT), but not the prothrombin time (PT), is puzzling. I can see how the adsorption of clotting factors in the intrinsic pathway could prolong the aPTT, but it is unclear how this phenomenon explains the prolonged TT. Although the adsorption of fibrinogen by the sponges could result in the prolongation of the TT, it should also prolong the PT. Studies using purified proteins may help to sort this out.”

Response to comment: Thanks for your professional comments. We fully agree with you about the possibility that the prolongation of PT might be associated with the prolonged TT as PT is a measure of the integrity of the extrinsic and final common pathways of the coagulation cascade. Actually, apart from the significantly prolonged aPTT and TT, slight prolongation of PT was indeed observed for both human plasma (~ 3 s) and rabbit plasma (~ 7 s) after contacting with MS@D-HMP (30 mg/300 μL human plasma, 0.4 g/mL rabbit plasma). But adsorption of fibrinogen contributed little in this process because the fibrinogen value after contacting with MS@D-HMP is around 1.4 g/L. For hypofibrinogenemia to prolong a TT, the fibrinogen value will usually be $\leq 0.7\text{-}1$ g/L.²⁸ In addition, the PT assay is less sensitive to deficiencies within the final common pathway (factors X, and II and fibrinogen) and more sensitive to deficiencies of factor VII within the extrinsic pathway.²⁹ For example, heparin at low concentration (0.5 IU/mL) prolonged TT to > 240 s, but only prolonged PT by ~ 10 s.²⁴ PT could be significantly prolonged by ~ 60 s when heparin concentration increased to 1 IU/mL. Hence,

the slight prolonged PT may be due to the inhibited thrombin in common pathway rather than adsorption of fibrinogen.

The underlying reasons for the prolonged TT and inhibited common pathway are further investigated. When performing TT tests, fibrinogen in the samples is converted to fibrin with the addition of TT reagent (Test Thrombin Reagent, Siemens) containing 1.5 IU/mL bovine thrombin. The turbidity change is monitored by an automatic blood coagulation analyzer (CA-50, Sysmex Corporation, Japan) to give a TT value once clot formation. Generally, the prolongation of TT value can be ascribed to two causes, the quantitative deficiency of fibrinogen in the plasma, or the presence of certain inhibitors in the plasma capable of inhibiting bovine thrombin, such as heparin.²⁹

For the former possibility, the loss of fibrinogen was indeed observed in the MS@D-HMP-incubated PPP (**Supplementary Fig. 21a**). The decreased fibrinogen was not converted to fibrin (**Fig. 5 h**). When replenishing the fibrinogen in the plasma to normal range with purified human fibrinogen, no reduction in the prolongation of TT value was observed (**Supplementary Fig. 21c**). Hence, the loss of fibrinogen made no contribution to the prolongation of TT value.

Fig. 5 h Concentration of D-dimer in the citrate-anticoagulated plasma after incubation with sponge. Plasma was re-calcified (final Ca^{2+} concentration at 10 mM), and then incubated with MS@D-HMP. Plasma without recalcification were used as normal control. For positive control, re-calcified plasma was incubated for 1 h, and fibrinolysis was initiated by addition of human plasmin (80 $\mu\text{g}/\text{mL}$) ($n = 3$ -4 biologically independent samples, mean \pm SD. One-way ANOVA with Bonferroni post-hoc tests).

For the latter possibility, certain inhibitors may exist in the plasma capable of inhibiting bovine thrombin. Unlike traditional water-soluble anticoagulants such as heparin, MS@D-HMP had been separated from the incubated PPP before performing the TT tests. In addition, the effect of leakage of the anticoagulant coating on the inhibited coagulation was investigated. The as-prepared MS@D-HMP was incubated with PBS buffer for different time intervals (1 day, 3 days, and 7 days). UV-vis spectra were utilized to detect the leakage in the eluate after incubation. PDA and HMP were chosen as the control samples, as they are the representative components of anticoagulant coating. The characteristic

absorption peaks for PDA and HMP solution were both around 200 nm (**Supplementary Figs. 22a, b**). No obvious absorption peak was observed for the eluate after incubation for 1 day, 3 days and 7 days (**Supplementary Fig. 22c**). In addition, the eluate was mixed with PPP, and the clotting times including aPTT, PT and TT were determined. Compared with the control PPP, the addition of eluate had negligible effect on the aPTT, PT and TT values (**Supplementary Figs. 22d-f**). Hence, the leakage of the anticoagulant coating and its effect on coagulation system are negligible. These results proved that the bovine thrombin added later in TT tests had no chance to come into direct contact with MS@D-HMP.

Supplementary Figure 22. Leakage of coating from sponge and its effect on clotting time. a-c UV-vis spectra for gradient-diluted polydopamine (a), HMP (b), and the eluate after incubation for 1 day, 3 days and 7 days (c). d-f Clotting times including aPTT (d), PT (e), and TT (f) of PPP mixed with the eluate after incubation for 1 day, 3 days and 7 days (n = 3 biologically independent samples, mean \pm SD. One-way ANOVA with Bonferroni post-hoc tests).

Hence, we speculated that MS@D-HMP induced the generation of unknown inhibitor which caused inhibition of thrombin. The problem, however, is that there are hundreds of proteins in plasma that are related to coagulation system, and we do not know exactly which specific proteins were involved in this process. Technical means to precisely analyze the transient protein-protein interactions, interactions between proteins and bio-interfaces, and conformational changes of proteins when contacting with bio-interfaces in the complex plasma environment, remain limited. Discovering these particular inhibitors would clearly require the innovation of systematic new techniques, which is beyond the scope of this article.

We have performed some exploration experiments to investigate the effect of two traditional inhibitors, including antithrombin III (ATIII) and heparin cofactor II (HCII), using ATIII-deficient plasma and purified HCII protein (Commercial HCII-deficient plasma is not available because of customs screening due to the Covid-19 pandemic). Clotting times of the ATIII-deficient plasma after contacting with the MS@D-HMP were determined. MS@D-HMP induced a significant prolongation

for aPTT and TT, and inhibited the activity of FXI significantly (**Supplementary Figs. 23a-c**). In contrast, heparin lost its coagulation inhibition ability in the absence of ATIII. This result meant that MS@D-HMP might not inhibit thrombin through ATIII like heparin does. Thrombin inhibition by HCII is possible at high heparin concentration.³⁰ Hence, the effect of the MS@D-HMP on HCII was investigated using purified HCII protein. FIIa cleaves fibrinogen and triggers the spontaneous fibrin polymerization, which caused turbidity change (**Supplementary Fig. 23d**). Neither HCII nor heparin alone significantly inhibited fibrin polymerization induced by thrombin, whereas a significantly inhibited thrombin activity was observed with co-existence of heparin and HCII (**Supplementary Fig. 23e**). In contrast, MS@D-HMP-treated HCII caused no significant inhibition of the thrombin activity. This result indicated that the inhibitory effects of the MS@D-HMP on common pathway might be not dependent on HCII like heparin does. Actually, this result may need further investigation if the following two conditions are present. One is that HCII may be adsorbed by sponge, another is that the effect of sponge on HCII relies on other substances in plasma. The HCII-deficient plasma could exclude the effect of these two possibilities. Unfortunately, such an experiment cannot be carried out in the near future due to uncontrollable forces, but we still hope to make more explorations in our future work. Overall, inhibition of thrombin by MS@D-HMP might not rely on ATIII and HCII like heparin does.

Supplementary Figure 23. Effect of ATIII and HCII on the anticoagulant behaviors of sponge. a-c aPTT values (a), FXI activity (b) and TT values (c) of ATIII-deficient plasma after incubation with MS@D-HMP. Heparin-anticoagulated ATIII-deficient plasma (0.1 IU/100 μ L ATIII-deficient plasma) was used for comparison (n = 3 biologically independent samples, mean \pm SD. One-way ANOVA with Bonferroni post-hoc tests). d Effect of MS@D-HMP on the inhibition of FIIa-mediated fibrin polymerization in the presence of HCII. Turbidity change was monitored by absorbance at 350 nm. (n = 3 biologically independent samples, mean \pm SD). e Calculated residual FIIa activity in the presence of HCII (n = 3 biologically independent samples, mean \pm SD. One-way ANOVA with Bonferroni post-hoc tests).

The related discussion about the association of prolonged PT and TT have been added on Page 5 of the revised manuscript: “MS@D-HMP significantly prolonged aPTT and TT in a concentration-dependent manner, and slightly prolonged PT (~ 3 s at 30 mg/300 μ L PPP)” and on Page 16 of the revised manuscript: “Less prolongation was observed for PT (16.4 and 13.9 s), which may be due to the inhibited thrombin in common pathway.”

The related discussion about the potential degradation of fibrinogen have been added on Page 14 of revised manuscript: “Given the activation of contact system, the possible generation of micro-thrombi and subsequent fibrinolysis in the plasma after incubation with MS@D-HMP was investigated based on the D-dimer level. Compared with normal plasma, no significant increase was observed for the D-dimer level in the re-calcified plasma treated by MS@D-HMP after 6-hour incubation, which meant that micro-thrombi did not form and no fibrin was degraded (Fig. 5h).”

The reasons for the prolonged TT have been reorganized and related discussions have been added on Page 12 of the revised manuscript: “The prolongation of TT value may have two possible causes: the quantitative deficiency of fibrinogen in the plasma, or the presence of certain inhibitors in the plasma that inhibit adscititious thrombin, such as heparin/ATIII complex⁴⁶. The loss of fibrinogen was observed in the MS@D-HMP-incubated plasma (Supplementary Figs. 21a-b). However, the decreased fibrinogen was not converted to fibrin (as discussed in Fig. 5h). When replenishing the fibrinogen in the plasma to normal range with purified human fibrinogen, no reduction in the prolongation of TT value was observed (Supplementary Fig. 21c). Hence, the loss of fibrinogen made no contribution to the prolongation of TT value.

MS@D-HMP had been removed from the incubated PPP before performing TT tests. The leaching of anticoagulant coating and its effect on clotting times were negligible (Supplementary Fig. 22). This meant that the adscititious thrombin in TT tests had no chance to come into direct contact with MS@D-HMP. We speculated that MS@D-HMP might induce the generation of unknown thrombin inhibitor in plasma. The effect of two traditional thrombin inhibitors, such as ATIII and heparin cofactor II (HCII), on sponge-mediated anticoagulation was investigated using ATIII-deficient plasma and purified HCII protein. In the case of ATIII-deficient plasma, MS@D-HMP induced a significant prolongation for aPTT and TT, and inhibited the activity of FXI significantly (Supplementary Figs. 23a-c). However, heparin lost its anticoagulant ability in ATIII-deficient plasma. Regarding HCII-mediated thrombin-initiated fibrin formation, the MS@D-HMP-treated HCII did not cause any significant inhibition of thrombin activity, while thrombin was significantly inhibited with the co-existence of heparin and HCII (Supplementary Figs. 23d-e). These results indicated that the inhibition of thrombin by MS@D-HMP may not rely on ATIII binding or HCII similar to heparin.”

The experimental details about the detection of D-dimer have been added on Page 12 of the revised supplementary information: “To rule out the possibility of generation of micro-fibrin (micro-thrombi)

and fibrinolysis in the plasma after incubation with MS@D-HMP, and then converted into D-dimer, the concentration of D-dimer in the plasma was detected via ELISA method (Human D-dimer ELISA Kit, Cusabio, CSB-E05175h). Citrate-anticoagulated PPP (100 μ L) was re-calcified with stock solution of calcium ions (final Ca^{2+} concentration at 10 mM), and then incubated with 10 mg MS@D-HMP. After incubation at 37 $^{\circ}$ C for 30 min, the plasma was collected (sponge was removed for MS@D-HMP-treated group) and incubated for another 6 h. Plasma without recalcification were used as normal control. For positive control, re-calcified plasma was incubated for 1 h, and fibrinolysis was initiated by addition of human plasmin (80 μ g/mL, Sigma-Aldrich) with 6-hour incubation. Finally, the detection of D-dimer was conducted according to the respective instruction manuals. At least 4 parallel sample groups were applied to get a reliable value, and the results were expressed as mean \pm SD.”

The experimental details about the investigation of leakage of anticoagulant and its effect on clotting times have been added on Page 10 of the revised supplementary information: “To investigate the leakage of the coating material from the sponge surface, 50 mg MS@D-HMP was incubated with 10 mL PBS buffer for different time intervals (1 day, 3 days, and 7 days). The eluate after incubation was collected. UV-vis spectra (UV-1750, Shimadzu) were recorded at the wavelengths from 190 to 400 nm using a 1 cm path length quartz cell. PDA (0.0625-1 mg/mL) and HMP (0.0625-1 mg/mL) were used as comparisons. Then, 20 μ L eluate was mixed with 200 μ L PPP, and clotting times were tested.”

The details about the preparation of heparin-mimetic polymers and polydopamine have been added on Page 4 of the revised supplementary information: “For the synthesis of soluble heparin-mimetic polymers (HMP), 12 g AA, 8 g AMPS, 0.1 g APS and 80 g deionized water were mixed to obtain homogeneous solution. The obtained reaction solution was heated to 70 $^{\circ}$ C for 10 h to initiate the free radical polymerization. After that, the pH of the resulted HMP solution was adjusted to 7 with NaOH solution under magnetic stirring. Finally, the solution was dialyzed against deionized water for 48 h to obtain the HMP, and lyophilized for further use. For the synthesis of polydopamine (PDA), DOPAM (5 mg/mL) and DOPA (1 mg/mL) were dissolved in Tris-buffer solution (pH=8.5, 50 mM), and kept for 12 h with oscillation at room temperature. The PDA particles were separated by centrifugation (20000 rpm) and washing multiple times with deionized water.”

The experimental details about the effect of ATIII and HCII on sponge-mediated anticoagulation have been added on Page 10 of the revised supplementary information: “To investigate the effect of the MS@D-HMP on ATIII, 10 mg MS@D-HMP was incubated with 100 μ L ATIII-deficient plasma (BioMedica Diagnostics Inc.) for 30 min. The incubated plasma was collected to determine the aPTT, TT and FXI activity. For comparison, heparin was added into the ATIII-deficient plasma at a concentration of 0.1 IU/100 μ L PPP (aPTT > 600 s, TT > 240 s for heparin-anticoagulated normal PPP at this concentration). The FXI activities were determined based on activated partial thromboplastin time (aPTT). Briefly, 5 μ L incubated ATIII-deficient plasma was added in a test cup and diluted with 45 μ L buffer solution. After incubating at 37 $^{\circ}$ C for 30 s, the complex solution was mixed with 50 μ L corresponding factor deficient plasma (Coagulation Factor XI deficient plasma; Sysmex; incubated 10 min at 37 $^{\circ}$ C before use). After another 30-second incubation at 37 $^{\circ}$ C, 50 μ L aPTT reagent was added, followed with adding 50 μ L CaCl_2 solution (25 mM) after incubating at 37 $^{\circ}$ C for 3 min, and then the aPTT value was measured in an automatic coagulation analyzer (ACL Elite Pro, werfan). Commercial normal plasma (Sysmex) was used to determine standard curve through the instrument's built-in gradient dilution procedure. Finally, the FXI factor activity was calculated based on the standard curve.

To investigate the effect of the MS@D-HMP on HCII, 6 mg MS@D-HMP was incubated with 60 μL HCII/PBS solution (0.34 μM) for 10 min, and 50 μL incubated solution was collected for HCII+MS@D-HMP group. For comparison, 50 μL PBS solution was incubated for 10 min for blank group. 50 μL HCII/PBS solution (0.34 μM) was incubated for 10 min for HCII group. 50 μL heparin/PBS solution (0.2 IU/ μL) was incubated for 10 min for heparin group. 50 μL HCII/heparin/PBS solution (0.34 μM for HCII, 0.2 IU/ μL for heparin) was incubated for 10 min for HCII+heparin group. These incubated solutions were mixed with 20 μL FIIa (3 IU/mL), and incubated for another 5 min. Finally, fibrinogen solution was added to obtain a final concentration at 2.5 mg/mL. The turbidity change was measured at 350 nm for 30 min in a plate reader (Thermo Scientific, USA). The residual FIIa activity was calculated based on the standard curve using gradient-diluted FIIa solution.”

Comment #6: “What is the effect of soluble HMP versus HMP immobilized on the sponges on blood coagulability? Does the sponge surface induce contact activation? These questions would be best addressed using purified proteins as well as a plasma system.”

Response to comment: Thanks for your good comments. The effect of HMP versus MS@D-HMP on blood coagulability was explored in terms of clotting times and fresh plasma-based correction assay. Firstly, the clotting times of PPP after adding different concentrations of HMP were investigated. It is obvious that HMP showed a concentration-dependent prolongation for aPTT and TT, but less prolongation for PT (**Fig. R2**). TT values exceeded detection limit (240 s) when the concentration of HMP reached 5 $\mu\text{g}/100 \mu\text{L}$ PPP. In comparison, MS@D-HMP achieved significant coagulation inhibition at a relatively higher concentration due to the introduction of the sponge substrate without anticoagulant ability, and poor surface/protein interaction compared with HMP (**Fig. 2a**). TT values reached equilibrium at around 130 s when the concentration of MS@D-HMP exceeded 20 mg/300 μL PPP.

Fig. R2 Clotting times (aPTT, PT and TT) of HMP-incubated PPP versus concentration curves. Certain amounts (0.5, 1, 2.5, 5 and 10 μg) of HMP were added into 100 μL citrate-anticoagulated PPP. After incubation for 5 min, the clotting times including aPTT, PT and TT were detected (n = 3 biologically independent samples, mean \pm SD).

Secondly, the HMP-anticoagulated PPP (10 $\mu\text{g}/100 \mu\text{L}$ PPP) was mixed with fresh citrate-anticoagulated PPP. The HMP-anticoagulated PPP showed a significant prolongation for TT with a low mixing concentration at 0.05 mL HMP-anticoagulated PPP/mL fresh PPP (**Fig. R3**). Similar to heparin-anticoagulated PPP, HMP-anticoagulated PPP can make TT value of mixed PPP exceed detection limit. Mixing MS@D-HMP-treated PPP showed significantly lower prolongation for TT

values with a plateau at around 100 s as MS@D-HMP was separated from the treated PPP.

Fig. R3 Correction of coagulation function by replenishing fresh normal plasma. a-c aPTT (a), PT (b) and TT (c) of different sample plasma after mixing with fresh citrate-anticoagulated normal plasma at different concentrations. Sodium citrate-, heparin-, and HMP-anticoagulated plasma, and plasma incubated with MS@D-HMP were replenished with fresh citrate-anticoagulated normal plasma at different concentrations. Citrate-anticoagulated blood was collected using vacuum tubes (5 mL, Jiangsu Kangjian Inc., China) containing 3.8 % sodium citrate with anticoagulant/blood ratio of 1:9. Heparin-anticoagulated blood was collected using commercial heparin vacuum tubes (5 mL, Jiangsu Kangjian Inc., China), and the final concentration of heparin was around 10 IU/mL blood. For plasma collection, the blood was centrifuged at 4000 rpm for 15 min to obtain PPP. The concentration of HMP in the HMP-anticoagulated plasma was 10 $\mu\text{g}/100 \mu\text{L}$ plasma. For MS@D-HMP-incubated plasma, 10 mg MS@D-HMP was incubated with 100 μL plasma for 30 min, and the plasma was collected (n = 3 biologically independent samples, mean \pm SD).

The results of various experiments, including plasma-based FXIIa chromogenic substrate S-2302 experiment, ELISA-based detection of generated bradykinin in plasma, and fluorometric-based quantitative monitoring of FIXa in plasma, confirmed that MS@D-HMP induced contact activation. The normal plasma treated by MS@D-HMP showed significant proteolytic activity to S-2302 (**Fig. 4d**), while FXII-deficient plasma treated by MS@D-HMP exhibited no obvious proteolytic activity (**Fig. 4e**). This meant that FXIIa generated in the plasma after contacting with MS@D-HMP. The generated FXIIa could activate FXI to FXIa, activate the downstream factors subsequently, and finally cause the formation of FXa (**Fig. 4b**). In addition, the generated FXIIa induced activation of kallikrein/kinin system with the formation of bradykinin (**Fig. 4c**). These plasma-based results well demonstrated the activation of contact system by MS@D-HMP.

We fully agree about the advantage of performing experiments using purified proteins. To investigate whether MS@D-HMP can directly induce activation of FXII, FXIIa chromogenic substrate S-2302 experiment was performed using FXII purified protein. FXII solution treated by MS@D-HMP showed significant proteolytic activity to S-2302 (**Supplementary Fig. 20**), which confirmed the direct activation of FXII to FXIIa by MS@D-HMP.

Supplementary Figure 20. Evaluation of FXIIa activity in the FXII solution after incubation with MS@D-HMP. Hydrolysis of the chromogenic FXIIa substrate S-2302 was detected at an absorbance = 405 nm (n = 3 biologically independent samples, mean ± SD. One-way ANOVA with Bonferroni post-hoc tests)

The related discussions have been added on Page 11 of the revised manuscript: “Activation of FXII by MS@D-HMP was confirmed via chromogenic assays, showing significant proteolytic activity to chromogenic FXIIa substrate (S-2302) both in the sole FXII solution and normal plasma treated by MS@D-HMP (Fig. 4d, Supplementary Fig. 20).”

The experimental details about evaluation of FXII activation in the FXII buffer solution have been added on Page 9 of the revised supplementary information: “To investigate whether MS@D-HMP can directly induce activation of FXII, FXIIa chromogenic substrate S-2302 experiment was performed using purified FXII protein. For the positive control group, 25 μL aPTT reagent (Dade Actin Activated Cephaloplastin Reagent, 210 Siemens) was added into 50 μL FXII/PBS solution (500 nM). For the negative control group, 25 μL PBS was added into 50 μL FXII/PBS solution (500 nM). For the sponge, 50 μL PBS was added into 100 μL FXII/PBS solution (500 nM), and then incubated with 5 mg MS@D-HMP. After incubation for 15 min, 50 μL incubated solution was mixed with 100 μL substrate S-2302 (Chromogenix, 4 mM). After incubation for 30 min, the obtained mixture was transferred to a 96-well plate immediately, and the optical density was detected at 405 nm in a plate reader (Thermo Scientific, USA).”

Comment #7: “The authors speculate that sponges impregnated with HMP-modified microspheres would obviate the need for filtration and washing of blood collected during surgery. However, such sponges could also adsorb bacteria, fat, or tumor cells. How would this potential problem be addressed and would this problem not dampen the enthusiasm for use of such sponges?”

Response to comment: Thanks for your professional comments, and we totally agree with your concerns about the potential contaminants such as bacteria, fat, or tumor cells during auto-transfusion.

We reconsidered two potential application scenarios while incorporating your concern.

The first one is whole blood auto-transfusion in a highly clean and sterile condition. The bacteria at the wound site would be greatly reduced as both sponge and patient's wound site are sterilized in advance. Administration of prophylactic antibiotics can further prevent infection if needed.³¹ This application had been verified in our animal model, and all the rabbits undergoing whole blood auto-transfusion lived healthily without infection. We further evaluated the activation of platelets and leucocytes after blood treatment, since the bacterial invasion accompanies with the activation of the immune cells in most cases. Flow cytometry was performed to evaluate the level of platelet and monocyte activation after incubation with the sponge. Compared with negative control, there was no significant difference for the expression of CD62p in the MS@D-HMP-incubated platelets, and CD11b in the MS@D-HMP-incubated monocytes (**Figs. 5f-g**). These results confirmed no significant activation of platelets and monocytes after incubation with the sponge. Under such condition, wash process can further collect the blood cells trapped by the sponge considering the low activation and surface adhesion of the cells. Whole blood was incubated with the sponge, and then separated into a centrifuge tube. Platelet-poor plasma was used to elute blood cells trapped in the sponge. As shown in **Supplementary Figure 25**, the levels of collected blood cells increased significantly compared with those without washing (RBC 113.9 %, WBC 94.8 % and PLT 78.6 %).

Adsorption of such fat, cancer cells are possible, and any blood salvage will have similar issues. When fat enters the circulation, it may initiate a fat embolism syndrome, which is characterized mainly by pulmonary dysfunction.³² To avoid the fat embolism syndrome when reinfused, the last part of blood should be discarded because of the fat layers on top of the blood.³³ Further removal of fat particles in the collected blood could be achieved with WBC depletion filters, such as BIO R Plus (Fresenius), RC100KLE (Pall Biomedical) and Sepacell (Asahi Medical), although WBCs are also removed.³⁴ In cases of malignancy, the use of traditional cell salvage has been controversial due to the theoretical risk of disseminating the tumor. WBC depletion filters have been used for filtration of malignancy in urologic surgery,^{35, 36} and pulmonary surgery,³⁷ and showed high effectiveness at removing tumor cell contamination. However, traditional cell salvage is still not recommended to be used during surgical procedures involving malignant tumors to prevent the metastasis of cancer cells that may have not been removed during the filtering step and are reinfused.^{38, 39} From this point, whole blood auto-transfusion using sponges is also not recommended in cases of malignancy. Overall, we totally agree with your concerns, especially about the adsorption of fat and tumor cells. We have to acknowledge that whole blood auto-transfusion using sponges alone cannot solve all these problems. Related limitations have been added and discussed in the revised manuscript.

The second application is portable blood salvage in unexpected situations, such as car accidents, battlefield scenarios and so on. The sponge recovers the blood to provide a valuable window of opportunity for subsequent clinical operations. In this case, caution is necessary for whole blood salvage due to the adsorption of environmental contaminants, and thus recovery and washing of blood cells might be a more conservative and high-safety strategy. However, this is still a breakthrough which has never been realized, as there is no suitable and portable device to recover blood cells directly from the patient's wound outside hospital setting.

In conclusion, we believe that the use of anticoagulant sponge for auto-transfusion still has substantial advantages in both scenarios. In the ideal case, the blood cells collected by sponge maintain

good condition, and the reinfusion of whole blood facilitates the faster recovery of coagulation function *in vivo*. In unexpected situations, only the sponge can recover blood cells directly from the patient's wound outside hospital setting, highlighting the distinction between sponge and traditional auto-transfusion technology.

The advantages of sponge in multi-scenario applications have been emphasized on Page 20 of the revised manuscript: “The portability and easy handling ability of the sponge also open up the possibility of multi-scenario applications (such as battleground, car accidents, etc.). In these unexpected situations, only anticoagulant sponge can recover blood cells directly from the patient's wound outside hospital setting, highlighting the distinction between sponge and traditional auto-transfusion technology.”

The limitations have been added in the Outlook on Page 20 of the revised manuscript: “It is important to say, however, contaminants such as bacteria, fat, bowel contents or amniotic fluid should be considered in more complex environments⁵⁵, to avoid risks including sepsis, fat embolism syndrome, renal failure, disseminated intravascular coagulation, and potentially death^{56,57}. Last part of blood with fat layers on the top should be discarded, and further removal of fat particles could be achieved in conjunction with WBC depletion filters⁵⁸. Considering the theoretical risk of tumor disseminating, whole blood auto-transfusion using sponges is still not recommended in cases of malignancy⁵⁹. Independent of this, the results showed that *in vivo* reinfusion of sponge-collected blood in highly sterile environment with prophylactic antibiotics administration may not cause detectable abnormality.”

The related discussion about the numbers of collected blood cells after washing have been added on Page 14 of the revised manuscript: “These reductions were mainly due to the physical interception of porous structures, and a large portion of the missing blood cells could be recovered through the washing procedure (Supplementary Fig. 25).”

The related discussion about platelet and monocyte activation have been added on Page 14 of the revised manuscript: “The real level of platelet activation was quantified by flow cytometry based on the expression of platelet activation marker CD62p. Compared with negative control, there was no significant difference in the expression of CD62p in the PRP treated with MS@D-HMP (Fig. 5f), confirming that MS@D-HMP did not induce significant platelet activation. The monocyte activation was also evaluated based on the expression of monocyte activation marker CD11b. Compared with negative control, there was no significant difference in the expression of CD11b in the MS@D-HMP-incubated monocytes (Fig. 5g), confirming no significant monocyte activation.”

MINOR

Comment #8: “The size of the sponges does not appear to be standardized across experiments nor is this information provided for all the experiments.”

Response to comment: Thanks for your good comment and we apologize for the missing details. In our experiments, we determined the sponge dosage by weighing it since the weight directly related to the amount of HMP coating. We conducted sufficient parallel experiments for reproducibility, and optimized sponge usage for various testing purposes:

Most of the *in vitro* anticoagulant experiments (Figs. 2b, 2d-g, 3b-d, 3f-h, 4g, 5h, Supplementary Figs. 23-24) used a uniform dose of 10 mg MS@D-HMP per 100 μ L biofluid (plasma, whole blood or

protein buffer solution) as MS@D-HMP showed excellent anticoagulant performances at this dose based on the results of clotting times in **Fig. 2a**.

To determine the adsorption capacity of sponge to FXI protein, MS@D-HMP (6 mg) was exactly immersed in the plasma (100 μ L) to ensure the similar interaction of each part of sponge with FXI in plasma (**Fig. 3e**). To perform the blood count assay for the sponge-treated blood, 25 mg MS@D-HMP was incubated with 400 μ L whole blood to ensure exact immersion (**Figs. 5a-b, Supplementary Figs. 25, 26a**).

To determine the adsorption kinetic of sponge to calcium ions, 20 mg MS@D-HMP was incubated with excess solution of calcium ions (20 mL) to ensure the equilibrium adsorption close to the saturation value (**Supplementary Fig. 14a**). To determine the adsorption of sponge to calcium ions in plasma environment, 10 mg MS@D-HMP was incubated with 400 μ L PPP as this experiment was also used to calculate the calcium supplementation dose when performing FIXa activity assay (**Supplementary Fig. 14b**).

To investigate the potential activation of coagulation factors induced by sponge, MS@D-HMP was incubated with excess PPP to exclude the influence of air. For example, 10 mg MS@D-HMP was incubated with 400 μ L PPP (**Figs. 4b, 4f, Supplementary Fig. 19**); 10 mg MS@D-HMP was incubated with 300 μ L PPP/normal saline solution (**Figs. 4d-e**); 5 mg MS@D-HMP was incubated with 150 μ L FXII buffer solution (**Supplementary Fig. 20**).

To evaluate the hemocompatibility, MS@D-HMP was incubated with excess PPP to ensure that sponge was in contact with sufficient blood components. For example, 10 mg MS@D-HMP was incubated with 200 μ L whole blood or PRP (**Figs. 5d-e, i-j, Supplementary Figs. 27**); 10 mg MS@D-HMP was incubated with 200 μ L PPP (**Fig. 4c**); 50 mg MS@D-HMP was incubated with 1 mL diluted RBC suspension (**Fig. 5c, Supplementary Fig. 26b**); 15 mg MS@D-HMP was incubated with 200 μ L PRP or whole blood (**Figs. 5f-g**).

Considering the higher concentration of calcium ions in rabbit whole blood and the hypercoagulable state of blood near the wound (indicated in **Supplementary Fig. 1**), the dose of sponge used in animal experiments (**Figs. 6-7, Supplementary Figs. 28-32**) was determined to be 0.4 g MS@D-HMP per 1 mL rabbit whole blood by preliminary experiments to ensure the efficacy and safety.

We have added the available information as best as possible in the revised figure captions and in revised supplementary information.

Comment #9: *“The figure legends provide inconsistent amounts of detail.”*

Response to comment: Thanks for your good comment. We have carefully checked our figure legends to ensure that each caption can be understood in isolation from the main text. To be specific, each legend begins with a brief title sentence for the whole figure and continues with a short statement of what is depicted in the figure. The experimental conditions and statistical analysis methods are added when appropriate. The related details have been added in the revised manuscript as shown below.

Page 5 in the revised manuscript: **“Fig. 1 Design and characterization of the sponges for instantaneous blood auto-transfusion. a** Schematic diagram of the preparation of the HMP-modified sponges. **The inset SEM images show the rough surface morphology after modification. Scale bars, 10 μ m. b** Pore-size distributions of MS, MS@D and MS@D-HMP. **The experiments were performed**

independently in duplicate, with similar results obtained. **c** Permeating behaviors of water (20 μL droplet) on the surface of MS, MS@D and MS@D-HMP. The experiments were performed independently in duplicate, with similar results obtained. **d** Quantitative evaluation of water (left) and blood (right) sorption into MS@D-HMP. The measured weight of water (left) and blood (right) absorbed into MS@D-HMP per unit contact area (m_s) versus the square root of sorption time ($t^{1/2}$). The experiments were performed independently in duplicate, with similar results obtained. **e** Zeta potentials of MS, MS@D and MS@D-HMP ($n = 6$ independent samples, mean \pm SD. One-way ANOVA with Bonferroni post-hoc tests). **f** Compressive stress-strain curves of MS@D-HMP after five loading-unloading cycles. The experiments were performed independently in duplicate, with similar results obtained.”

Page 7 in the revised manuscript: “**Fig. 2 Coagulation inhibition behaviors of the anticoagulant sponges *in vitro*.** **a** Concentration-dependent prolongation of clotting times (aPTT, PT and TT) for the citrate-anticoagulated PPP after incubation with MS@D-HMP for 30 min ($n = 3$ biologically independent samples, mean \pm SD). **b** aPTT for the citrate-anticoagulated PPP (100 μL) after incubation with MS@D-HMP (10 mg) for different time intervals ($n = 3$ biologically independent samples, mean \pm SD). **c** Concentration-dependent prolongation of PRTs for the citrate-anticoagulated PPP after incubation with MS@D-HMP ($n = 3$ biologically independent samples, mean \pm SD). PPP (300 μL) was re-calcified with CaCl_2 solution (final Ca^{2+} concentration at 12.5 mM), and then incubated with MS@D-HMP. The treated PPP was taken out immediately (within 5 s), and monitored for clotting. **d** Photographs of recalcification of fresh plasma (upper) and MS@D-HMP-treated plasma (lower) in the glass vial. **e** Schematic diagrams and photographs of the whole blood (without anticoagulant) after incubation with MS@D-HMP *in vitro*. **f** TEG traces for pristine whole blood and blood after incubation with MS@D-HMP. **g** Generated concentrations of TAT in the citrate-anticoagulated whole blood after incubation with MS, MS@D, and MS@D-HMP for 30 min. Positive control (recalcified blood, final Ca^{2+} concentration at 10 mM) and negative control (citrate-anticoagulated blood) are also shown ($n = 3$ biologically independent samples, mean \pm SD. One-way ANOVA with Bonferroni post-hoc tests).”

Page 10 in the revised manuscript: “**Fig. 3 Mechanisms of coagulation factor inactivation of the sponges.** **a** Activities of FVIII, FIX, FXI and FXII for the normal citrate-anticoagulated PPP after incubation with MS@D-HMP at different concentrations for 30 min. The activities were determined based on aPTT by mixing incubated plasma with corresponding factor deficient plasma ($n = 5$ biologically independent samples, mean \pm SD. One-way ANOVA with Bonferroni post-hoc tests). **b** Abundance of coagulation-related proteins in the normal plasma and MS@D-HMP-treated plasma. Citrate-anticoagulated PPP (300 μL) was incubated with MS@D-HMP (30 mg) for 30 min. Values are expressed in relative percentages based on total proteins. **c** Abundance of contact system-related proteins in normal PPP and MS@D-HMP-incubated plasma. Values are expressed in relative percentages based on total proteins. **d** Schematic of two-stage successive incubations combined with PBS treatment. In stage 1, tightly- and loosely-bound proteins were adsorbed on the sponge in 1st incubation with fresh plasma. Such sponge without washing was allowed for 2nd incubation with another fresh plasma. Then, the sponge undergoing stage 1 was washed by PBS to remove the loosely-bound proteins, and used in stage 2 with similar procedures as those of stage 1. **e** Equilibrium binding capacity and the corresponding inactivation efficiency of FXI by MS@D-HMP. FXI-deficient plasma (100 μL) supplemented with different amounts of FXI proteins was incubated with MS@D-HMP (6 mg) for 30 min ($n = 3$ biologically independent samples, mean \pm SD). **f** aPTT values for the PPP after

successive incubations with MS@D-HMP in stages 1 and 2 ($n = 8$ biologically independent samples, mean \pm SD. Two-way ANOVA with Bonferroni post-hoc tests). **g** Abundance of coagulation-related proteins in fresh plasma, plasma after 1st incubation with MS@D-HMP in stages 1, and plasma after 1st incubation with MS@D-HMP in stages 2. Values are expressed in relative percentages based on total proteins. **h** Abundance of coagulation-related proteins in the tightly-bound protein layer on the surface of MS@D-HMP in stages 1 and 2. Values are expressed in relative percentages based on total proteins.”

Page 13-14 in the revised manuscript: “**Fig. 4 Evaluation of inhibited coagulation pathways in the sponges-treated plasma.** **a** Schematic of exploring the procoagulant activity of FXIa in the MS@D-HMP-incubated plasma via fluorometric-based FIXa activity assay. **b** Quantitative monitoring of FIXa in the normal **hirudin-anticoagulated** plasma after incubation with MS@D-HMP for 10 min. Actin (aPTT reagent) was used as positive control and citrate was used to block the activation of FIX by FXIa ($n = 4$ biologically independent samples, mean \pm SD. One-way ANOVA with Bonferroni post-hoc tests). **c** Generated concentrations of bradykinin in the **citrate-anticoagulated** plasma after incubation with MS@D-HMP for 20 min. Plasma without treatment and plasma treated with glass power (1 mg/100 μ L plasma) for 20 min were used as negative and positive controls, respectively ($n = 6$ biologically independent samples, mean \pm SD. One-way ANOVA with Bonferroni post-hoc tests). **d-e** Evaluation of FXIIa activity in normal plasma (**d**) and FXII-deficient plasma (**e**) after incubation with MS@D-HMP for 15 min using hydrolysis of chromogenic substrate S-2302 at an absorbance $\lambda = 405$ nm. Positive control (actin, aPTT reagent) was shown in (**d**) ($n = 3-6$ biologically independent samples, mean \pm SD. One-way ANOVA with Bonferroni post-hoc tests for (**d**). Unpaired, two-tailed student’s *t*-test for (**e**)). **f** Quantitative monitoring of FIXa in the FXII-deficient plasma after incubation with MS@D-HMP for 10 min ($n = 3$ biologically independent samples, mean \pm SD. One-way ANOVA with Bonferroni post-hoc tests). **g** Activities of FVIII, FIX and FXI for the FXII-deficient plasma after incubation with MS@D-HMP for 30 min ($n = 3$ biologically independent samples, mean \pm SD. Unpaired, two-tailed student’s *t*-test). **h** Schematic of initiated and inhibited coagulation cascade by traditional negatively charged surface and MS@D-HMP, respectively.”

Page 15-16 in the revised manuscript: “**Fig. 5 Biocompatibility evaluation of the anticoagulant sponges in vitro.** **a** Blood cell count for pristine whole blood and blood after incubation with MS@D-HMP for 30 min ($n = 6$ biologically independent samples, mean \pm SD. Unpaired, two-tailed student’s *t*-test). **b** Differential white blood cell count (DIFF) scatter (forward (FS), side (SS)) charts of pristine whole blood (left) and blood after incubation with MS@D-HMP (right) for 30 min. **c** Hemolysis ratio of the blood after incubation with MS@D-HMP for 3 h. Deionized water was used as positive control ($n = 3$ biologically independent samples, mean \pm SD. Unpaired, two-tailed student’s *t*-test). **d** Quantitative evaluation of adhered platelets on the sponges by LDH assay ($n = 8$ biologically independent samples, mean \pm SD. One-way ANOVA with Bonferroni post-hoc tests). **e** Generation of PF4 in the whole blood after incubation with sponges for 30 min ($n = 3$ biologically independent samples, mean \pm SD. Kruskal-Wallis test with Dunn’s multiple comparisons). **f** Effect of sponges on platelet activation in PRP, measured on the basis of expression of platelet activation marker CD62p. Thrombin receptor activating peptide (TRAP, 0.1 mM) and control PRP were used as positive and negative control, respectively ($n = 5$ biologically independent samples, mean \pm SD. One-way ANOVA with Bonferroni post-hoc tests). **g** Effect of sponges on monocyte activation in whole blood, measured on the basis of expression of monocyte activation marker CD11b. Lipopolysaccharide (10 ng/mL) and

control whole blood were used as positive and negative control, respectively (n = 4 biologically independent samples, mean ± SD. One-way ANOVA with Bonferroni post-hoc tests). **h** Concentration of D-dimer in the citrate-anticoagulated plasma after incubation with sponge. Plasma was re-calcified (final Ca²⁺ concentration at 10 mM), and then incubated with MS@D-HMP. Plasma without recalcification were used as normal control. For positive control, re-calcified plasma was incubated for 1 h, and fibrinolysis was initiated by addition of human plasmin (80 ug/mL) (n = 3-4 biologically independent samples, mean ± SD. One-way ANOVA with Bonferroni post-hoc tests). **i-j** Generation of C3a (**i**) and C5a (**j**) in the whole blood after incubation with sponges for 30 min. Cobra-venom factor⁵¹ (CVF, final concentration at 25 µg/mL) and PBS were used as positive and negative controls, respectively (n = 4-6 biologically independent samples, mean ± SD. One-way ANOVA with Bonferroni post-hoc tests for (**i**), Kruskal-Wallis test with Dunn's multiple comparisons for (**j**)).”

Page 17 in the revised manuscript: “**Fig. 6 Blood collection and anticoagulation using MS@D-HMP in rabbits.** **a** Schematic of whole blood collection and instantaneous anticoagulation using heparin or MS@D-HMP in rabbits. **b** Images of WBCTs for heparin-treated group and MS@D-HMP-treated group. The collected blood (150 µL/well) was incubated in a 96-well polystyrene plate, and the un-clotted blood was washed away by PBS (150 µL) after different time intervals. The collected blood without anticoagulant (before) was shown as comparison. **c** Relative hemoglobin absorbance RHA(t) plot for the collected whole blood in heparin-treated group and MS@D-HMP-treated group at different time intervals. The free RBCs in the un-clotted blood was resuspended in 1 mL deionized water, and hemoglobin concentration was measured at an absorbance $\lambda = 540$ nm (n = 3 biologically independent samples, mean ± SD. Two-way ANOVA with Geisser-Greenhouse correction and Bonferroni post-hoc tests). **d** aPTT, PT and TT values for heparin-treated group and MS@D-HMP-treated group (n = 3 biologically independent samples, mean ± SD. One-way ANOVA with Bonferroni post-hoc tests). **e** Activity of intrinsic coagulation factors measured by titration experiments in the dFVIII, dFIX, dFXI and dFXII PPP for heparin-treated group and MS@D-HMP-treated group (n = 3 biologically independent samples, mean ± SD. One-way ANOVA with Bonferroni post-hoc tests). **f** Change in the serum calcium concentrations after treatment with heparin or MS@D-HMP (n = 3 biologically independent samples, mean ± SD. One-way ANOVA with Bonferroni post-hoc tests). **g-i** Hemolysis ratio (**g**), blood cell levels (**h**) and protein levels (**i**) of the collected blood in heparin-treated group and MS@D-HMP-treated group (n = 3 biologically independent samples, mean ± SD. One-way ANOVA with Bonferroni post-hoc tests for (**g**). Paired, two-tailed student's *t*-test for (**h**), and GLB and TP in (**i**). Wilcoxon matched-pairs signed-tank test for ALB in (**i**)).”

Page 19 in the revised manuscript: “**Fig. 7 Efficacy and safety of MS@D-HMP in a rabbit femoral artery hemorrhage model.** **a** Schematic of the whole blood auto-transfusion and experimental time line showing the sequence of events. **b-e** aPTT (**b**), TT (**c**), FDP (**d**) and D-dimer (**e**) for heparin-treated group and MS@D-HMP-treated group (n = 3 biologically independent samples, mean ± SD. Two-way ANOVA with Geisser-Greenhouse correction and Bonferroni post-hoc tests). **f** Blood cell count for heparin-treated group and MS@D-HMP-treated group (n = 3 biologically independent samples, mean ± SD). **g-h** Analysis of liver function parameters (**g,h**), renal function parameters (**i**), serum complement and immunoglobulins levels (**j**), blood lipids levels (**k**) and serum electrolyte levels (**l**) of the rabbits in heparin-treated group and MS@D-HMP-treated group: **g**: total protein (TP; g per litre); albumin (ALB; g per litre); globulin (GLB; g per litre); and albumin globulin ratio (A/G); **h**: alkaline phosphatase (ALP; units per litre); alanine aminotransferase (ALT; units per litre); aspartate

aminotransferase (AST; units per litre); gamma-glutamyl transferase (GGT; units per litre); direct bilirubin (DBIL; μmol per litre); total bilirubin (TBIL; μmol per litre); **i**: urea (UREA; mmol per litre); creatinine (CREA; μmol per litre); and uric acid (UA; μmol per litre); **j**: complement component 4 (C4; g l^{-1}); complement component 3 (C3; g l^{-1}); immunoglobulin A (IGA; g l^{-1}); immunoglobulin G (IGG; g l^{-1}) and immunoglobulin M (IGM; g l^{-1}). **k**: cholesterol (CHOL; mmol per litre); triglyceride (TG; mmol per litre); high-density lipoprotein (HDL; mmol per litre); low-density lipoprotein (LDL; mmol per litre); **l**: potassium, sodium, chlorine, phosphorus and calcium (quantity in mmol per litre) ($n = 3$ biologically independent samples, mean \pm SD).”

Comment #10: “There is inconsistent use of statistical tests to determine the relevance of the findings.”

Response to comment: Thanks for your good comment and we apologize for the mistakes. We have carefully reviewed our statistical analysis to ensure that the same statistical tests are consistently applied across all relevant analyses in the article. We have verified that the statistical tests used are appropriate for the data and research questions at hand, and have ensured that they are consistently applied in a rigorous and accurate manner throughout the article.

In general, data are expressed as mean \pm SD., indicated by error bars in all graphs. Each experiment was performed independently at least in duplicate and quantified at least in triplicate. Data was statistically analyzed with unpaired, two-tailed student’s *t*-test (Figs. 4e, 4g, 5a, 5c, Supplementary Fig. 10), paired, two-tailed student’s *t*-test (Fig. 6 h-i) or one-way ANOVA followed by Bonferroni post-hoc tests (Figs. 1e, 2g, 4b-d, 4f, 5d, 5f, 5g-i, 6d-g, Supplementary Figs. 13, 15-16, 19-23, 25). If data did not reach the criteria for parametric statistics, non-parametric statistics using Wilcoxon Signed-Rank Test (Fig. 6i) or Kruskal-Wallis test (Figs. 5e, 5j) followed by Dunn’s multiple comparisons if two or three samples, respectively, were tested. In time course analyses, mixed two-way ANOVA with Bonferroni post-hoc tests were used (Figs. 3f, 6c, 7b-e, Supplementary Figs. 31-32). GraphPad Prism v.8.0. was used for statistical analysis. $P < 0.05$ was considered statistically significant.

The related details about statistical analysis have been added on Page 23 of the revised manuscript: “In general, data are expressed as mean \pm SD., indicated by error bars in all graphs. Each experiment was performed independently at least in duplicate and quantified at least in triplicate. Data was statistically analyzed with unpaired, two-tailed student’s *t*-test (Figs. 4e, 4g, 5a, 5c, Supplementary Fig. 10), paired, two-tailed student’s *t*-test (Fig. 6 h-i) or one-way ANOVA followed by Bonferroni post-hoc tests (Figs. 1e, 2g, 4b-d, 4f, 5d, 5f, 5g-i, 6d-g, Supplementary Figs. 13, 15-16, 19-23, 25). If data did not reach the criteria for parametric statistics, non-parametric statistics using Wilcoxon Signed-Rank Test (Fig. 6i) or Kruskal-Wallis test (Figs. 5e, 5j) followed by Dunn’s multiple comparisons if two or three samples, respectively, were tested. In time course analyses, mixed two-way ANOVA with Bonferroni post-hoc tests were used (Figs. 3f, 6c, 7b-e, Supplementary Figs. 31-32). GraphPad Prism v.8.0. was used for statistical analysis. $P < 0.05$ was considered statistically significant.”

References

1. Paluck SJ, Nguyen TH, Maynard HD. Heparin-mimicking polymers: synthesis and biological applications. *Biomacromolecules* **17**, 3417-3440 (2016).
2. Ma L, *et al.* In vitro and in vivo anticoagulant activity of heparin-like biomacromolecules and the mechanism analysis for heparin-mimicking activity. *Int. J. Biol. Macromol.* **122**, 784-792 (2019).

3. Choay J, Petitou M, Lormeau JC, Sinay P, Casu B, Gatti G. Structure-activity relationship in heparin - a synthetic pentasaccharide with high-affinity for anti-thrombin-III and eliciting high anti-factor-Xa activity. *Biochem. Biophys. Res. Commun.* **116**, 492-499 (1983).
4. Vanboeckel CAA, Petitou M. The unique antithrombin-III binding domain of heparin - a lead to new synthetic antithrombotics. *Angew. Chem. Int. Ed.* **32**, 1671-1690 (1993).
5. Petitou M, *et al.* Synthesis of thrombin-inhibiting heparin mimetics without side effects. *Nature* **398**, 417-422 (1999).
6. Grootenhuis PDJ, Westerduin P, Meuleman D, Petitou M, Vanboeckel CAA. Rational design of synthetic heparin analogs with tailor-made coagulation-factor inhibitory activity. *Nat. Struct. Biol.* **2**, 736-739 (1995).
7. Herbert JM, *et al.* SR123781A, a synthetic heparin mimetic. *Thromb. Haemost.* **85**, 852-860 (2001).
8. Avci FY, Karst NA, Linhardt RJ. Synthetic oligosaccharides as heparin-mimetics displaying anticoagulant properties. *Curr. Pharm. Des.* **9**, 2323-2335 (2003).
9. Demir M, *et al.* Anticoagulant and antiprotease profiles of a novel natural heparinomimetic mannopentaose phosphate sulfate (PI-88). *Clin. Appl. Thromb./Hemost.* **7**, 131-140 (2001).
10. Hayakawa Y, Hayashi T, Hayashi T, Niiya K, Sakuragawa N. Selective activation of heparin-cofactor-II by a sulfated polysaccharide isolated from the leaves of artemisia-princeps. *Blood Coagul. Fibrinolysis* **6**, 643-649 (1995).
11. Monien BH, Cheang KI, Desai UR. Mechanism of poly(acrylic acid) acceleration of antithrombin inhibition of thrombin: implications for the design of novel heparin mimics. *J. Med. Chem.* **48**, 5360-5368 (2005).
12. Al Nahain A, Ignjatovic V, Monagle P, Tsanaktsidis J, Ferro V. Heparin mimetics with anticoagulant activity. *Med. Res. Rev.* **38**, 1582-1613 (2018).
13. Coombe DR, Kett WC. Heparin mimetics. *Handb. Exp. Pharmacol.*, 361-383 (2012).
14. Ito Y, Liu L, Imanishi Y. Interaction of poly(sodium vinyl sulfonate) and its surface graft with antithrombin-III. *J. Biomed. Mater. Res.* **25**, 99-115 (1991).
15. Jeske W, Fareed J. Antithrombin III-mediated and heparin-cofactor II-mediated anticoagulant and antiprotease actions of heparin and its synthetic analogs. *Semin. Thromb. Hemost.* **19**, 241-247 (1993).
16. Onishi M, Miyashita Y, Motomura T, Yamashita S, Sakamoto N, Akashi M. Anticoagulant and antiprotease activities of a heparinoid sulfated glucoside-bearing polymer. *J. Biomater. Sci. Polym. Ed.* **9**, 973-984 (1998).
17. Sugidachi A, Asai F, Koike H. Anticoagulant and antiprotease activities of aprosulate sodium, a new synthetic polyanion, in human plasma and purified systems. *Blood Coagul. Fibrinolysis* **5**, 773-779 (1994).
18. Tamada Y, Murata M, Hayashi T, Goto K. Anticoagulant mechanism of sulfonated polyisoprenes. *Biomaterials* **23**, 1375-1382 (2002).
19. Sano M, Tamada Y, Niwa K, Morita T, Yoshino G. Sulfated sericin is a novel anticoagulant influencing the blood coagulation cascade. *J. Biomater. Sci. Polym. Ed.* **20**, 773-783 (2009).
20. Charef S, TaponBretaudiere J, Fischer AM, Pfluger F, Jozefowicz M, Labarre D. Heparin-like functionalized polymer surfaces: discrimination between catalytic and adsorption processes during the course of thrombin inhibition. *Biomaterials* **17**, 903-912 (1996).
21. Fougnot C, Jozefowicz M, Rosenberg RD. Catalysis of the generation of thrombin-antithrombin complex by insoluble anticoagulant polystyrene derivatives. *Biomaterials* **5**, 94-99 (1984).
22. Belattar N. Affinity adsorption of human vitamin K-dependent coagulation factor IX onto heparin-like poly (styrene sodium sulfonate) adsorbent. *Mater. Sci. Eng. C* **27**, 849-854 (2007).
23. Fougnot C, Jozefowicz M, Bara L, Samama M. Interactions of anticoagulant insoluble modified polystyrene resins with plasmatic proteins. *Thromb. Res.* **28**, 37-46 (1982).
24. Song X, *et al.* Transient blood thinning during extracorporeal blood purification via the inactivation of coagulation

- factors by hydrogel microspheres. *Nat. Biomed. Eng.* **5**, 1143-1156 (2021).
25. Wang J, Guo X. Adsorption isotherm models: classification, physical meaning, application and solving method. *Chemosphere* **258**, 127279 (2020).
 26. Lu X, Xu P, Ding H-M, Yu Y-S, Huo D, Ma Y-Q. Tailoring the component of protein corona via simple chemistry. *Nat. Commun.* **10**, 4520 (2019).
 27. Vogler EA. Protein adsorption in three dimensions. *Biomaterials* **33**, 1201-1237 (2012).
 28. Kamal AH, Tefferi A, Pruthi RK. How to interpret and pursue an abnormal prothrombin time, activated partial thromboplastin time, and bleeding time in adults. *Mayo Clin. Proc.* **82**, 864-873 (2007).
 29. Bennett ST, Lehman CM, Rodgers GM. *Laboratory hemostasis: a practical guide for pathologists*. Springer (2014).
 30. Tollesfsen DM, Majerus DW, Blank MK. Heparin cofactor-II - purification and properties of a heparin-dependent inhibitor of thrombin in human-plasma. *J. Biol. Chem.* **257**, 2162-2169 (1982).
 31. Wollinsky KH, Oethinger M, Buchele M, Kluger P, Puhl W, Mehrkens HH. Autotransfusion - bacterial contamination during hip arthroplasty and efficacy of cefuroxime prophylaxis - a randomized controlled study of 40 patients. *Acta Orthop. Scand.* **68**, 225-230 (1997).
 32. Johnson MJ, Lucas GL. Fat embolism syndrome. *Orthopedics* **19**, 41-49 (1996).
 33. Michael B, Manfred F, Detlef F, Martin S, Yves M, Hugo VA. Fat elimination during intraoperative autotransfusion: an in vitro investigation. *Anesth. Analg.* **85**, 959-962 (1997).
 34. Ramirez G, Romero A, Garcia-Vallejo JJ, Munoz M. Detection and removal of fat particles from postoperative salvaged blood in orthopedic surgery. *Transfusion* **42**, 66-75 (2002).
 35. Edelman MJ, Potter P, Mahaffey KG, Frink R, Leidich RB. The potential for reintroduction of tumor cells during intraoperative blood salvage: reduction of risk with use of the RC-400 leukocyte depletion filter. *Urology* **47**, 179-181 (1996).
 36. Wiesel M, Gudemann C, Hoever KH, Staehler G, Martin E. Separation of urologic tumor cells from red blood cells by the use of a cell-saver and membrane filters. *Invest. urol.* **5**, 244-248 (1994).
 37. Perseghin P, Vigano M, Rocco G, DellaPona C, Buscemi A, Rizzi A. Effectiveness of leukocyte filters in reducing tumor cell contamination after intraoperative blood salvage in lung cancer patients. *Vox Sang.* **72**, 221-224 (1997).
 38. Kuppura L, Wee M. Perioperative cell salvage. *CEACCP* **10**, 104-108 (2010).
 39. Guidelines for best practices in intraoperative cell salvage. *AST*, (2018).

REVIEWERS' COMMENTS

Reviewer #1 (Remarks to the Author):

Review to Ms NCOMMS-22-37903-R1

Tao Xi et al., Highly efficient self-anticoagulant sponges for whole blood auto-transfusion and its mechanism of coagulation factor inactivation

In this revised version, the authors reacted carefully to all reviewer comments. New experiments have been performed for the interaction of the described material with calcium ions, antithrombin III, heparin cofactor II, saturation effects, D-dimer formation and more.

The manuscript is slightly reorganized and more clearly structured by subsections. The background and intention of the study are better elaborated.

The reviewer sees no scientific problems with publication of the text. However, the language, especially of new sections, needs careful redactional or professional revision. Frequently, articles are missing.

Reviewer #2 (Remarks to the Author):

The authors have responded adequately to the reviewer comments. The manuscript can be now accepted for publication in Nature Communications in its present form.

Reviewer #3 (Remarks to the Author):

The authors have done extensive additional work to address my concerns.

Point-by-point response to the detailed comments by the reviewers of “*Highly efficient self-anticoagulant sponges for whole blood auto-transfusion and its mechanism of coagulation factor inactivation*” with manuscript ID: NCOMMS-22-37903A.

Reviewer #1 (Remarks to the Author):

Review to Ms NCOMMS-22-37903-R1

Tao Xi et al., Highly efficient self-anticoagulant sponges for whole blood auto-transfusion and its mechanism of coagulation factor inactivation

In this revised version, the authors reacted carefully to all reviewer comments. New experiments have been performed for the interaction of the described material with calcium ions, antithrombin III, heparin cofactor II, saturation effects, D-dimer formation and more.

The manuscript is slightly reorganized and more clearly structured by subsections. The background and intention of the study are better elaborated.

The reviewer sees no scientific problems with publication of the text. However, the language, especially of new sections, needs careful redactional or professional revision. Frequently, articles are missing.

Response to comment: Thank you for the support for this manuscript. We appreciate your comments, and we have asked a native speaker to review our manuscript again to make sure the language and contents are all readable and clear to readers. We believe that the quality of this revised manuscript is improved to meet the requirement.

Reviewer #2 (Remarks to the Author):

The authors have responded adequately to the reviewer comments. The manuscript can be now accepted for publication in Nature Communications in its present form.

Response to comment: Thank you for your support. We really appreciate your professional suggestions and comments on our manuscript.

Reviewer #3 (Remarks to the Author):

The authors have done extensive additional work to address my concerns.

Response to comment: Thank you for re-reading our manuscript and the support.